# Understanding Dynamics of Adam in Zero-Sum Games: An ODE Approach

Yi Feng [*1]   Weiming Ou [*2]   Xiao Wang [*3]

## Abstract

The remarkable success of the Adam in training neural networks has naturally led to the widespread use of its descent-ascent counterpart, Adam-DA, for solving zero-sum games. Despite its popularity in practice, a rigorous theoretical understanding of Adam-DA still lags behind. In this paper, we derive ordinary differential equations (ODEs) that serve as continuous-time limits of the Adam-DA. These ODEs closely approximate the discrete-time dynamics of Adam-DA, providing a tractable analytical framework for understanding its behavior in zero-sum games. Using this ODE approach, we investigate two fundamental aspects of Adam-DA: local convergence and implicit gradient regularization. Our analysis reveals that the roles of the first- and second-order momentum parameters in zero-sum games are exactly the opposite of their well-documented effects in minimization problems. We validate these predictions through GAN experiments across multiple architectures and datasets, demonstrating the practical implications of this reversed momentum effect.

## 1. Introduction

Zero-sum games lie at the core of many modern machine learning tasks, including GANs (Goodfellow et al., 2014a) and adversarial training (Goodfellow et al., 2014b). While a rich body of theory has been developed for solving zero-sum games, particularly for Gradient Descent-Ascent (GDA) and its variants (Daskalakis et al., 2017; Mokhtari et al., 2020; Fasoulakis et al., 2022), in practice Adam Descent-Ascent (Adam-DA), the descent-ascent counterpart of Adam (Kingma & Ba, 2014b), remains the optimizer of choice. For

instance, most GANs studies employ Adam-DA for training (Arjovsky et al., 2017; Zhao et al., 2021; Sauer et al., 2023). Despite this ubiquity, the theoretical understanding of Adam in zero-sum games remains far less developed compared to its extensive studied behavior in minimization.

When studying momentum-based algorithms in games, such as Adam-DA, one might expect that, although some game-specific adaptations are necessary, the principles derived from theoretical analyses in minimization largely remain applicable to zero-sum games. For instance, the recent work of (Huang & Zhang, 2022), (Bot et al., 2023) and (Lotidis et al., 2024) study the benefits of incorporating Nesterov or Heavy-ball momentum, a core component of Adam, in game-solving algorithms. All of these works primarily focus on *positive* momentum parameters, reflecting the standard choice in optimization algorithms for minimization. However, practical experience in GANs training–a prominent application of zero-sum games–suggests that *negative* momentum often improves stability and performance (Gidel et al., 2019). This gap between theory and practice motivates the following questions:

*How can we theoretically understand Adam's behavior in zero-sum games? In particular, in which aspects does Adam behave differently in zero-sum games compared to standard minimization?*

In this work, we try to answer above questions by deriving ordinary differential equations (ODEs) which are the continuous time limit of Adam-DA. The derived ODEs accurately approximate the underlying algorithm. This approach of using ODEs to study optimization algorithms dates back to Polyak's seminal work (Polyak, 1964) for analyzing momentum method in minimization problems and has been further developed in recent years to analyze the dynamics of various algorithms (Su et al., 2016; Wibisono et al., 2016; Shi et al., 2022).

In particular, we study two aspects of Adam in zero-sum games: **local convergence** and **implicit gradient regularization**. These two aspects are natural targets: local convergence characterizes the behavior of the trajectory near an equilibrium, whereas implicit gradient regularization provides a more global perspective by describing how the trajectory interacts with the geometry (e.g., flatness) of the loss landscape. The two viewpoints are complementary and,

---
[*]Authors are listed in alphabetical order. [1]Aarhus University, Aarhus, Denmark. [2]Shanghai University of Finance and Economics, Shanghai, China. [3]MoE Key Laboratory of Interdisciplinary Research of Computation and Economics, Shanghai University of Finance and Economics, Shanghai, China. Correspondence to: Xiao Wang <wangxiao@sufe.edu.cn>.

*Proceedings of the 43ʳᵈ International Conference on Machine Learning*, Seoul, South Korea. PMLR 306, 2026. Copyright 2026 by the author(s).

taken together, offer a more complete picture of Adam's dynamics in zero-sum games.

## 1.1. Summary of Contributions and Limitations

**Contributions on Local Convergence:** We provide quantitative local convergence results for both the ODEs of Adam-DA (Theorem 4.3) and the Adam-DA algorithm (Theorem 4.4). We find that, in typical zero-sum games, a *smaller* first-order momentum improves local convergence over a broader range of step sizes (Corollary 4.5). This contrasts with Adam's behavior in minimization, where achieving a similar effect typically requires a *larger* first-order momentum. As a further corollary, we show that Adam-DA always diverges on bilinear objectives, regardless of parameter choices (Corollary 4.6). This is again opposite to the minimization setting, where Adam can converge to a neighborhood of a local minimum under suitable parameters (Zhang et al., 2022b). These results substantially extend prior separations between game dynamics and minimization, which were previously established mainly for simpler methods without adaptivity and momentum (Bailey & Piliouras, 2018; Bailey et al., 2020).

**Contributions on Implicit Gradient Regularization:** We provide qualitative descriptions of how the parameters of algorithm influence the interaction between the algorithm's trajectories and the flatness of min-max loss landscapes through the analysis of purposed ODEs. We find that a *smaller* first-order momentum $\beta$ and a *larger* second-order momentum $\rho$ guide the trajectories toward flatter regions of the loss landscape. Again, this behavior is exactly the opposite of the implicit gradient regularization effect observed in minimization, where achieving a similar effect requires a *larger* $\beta$ and a *smaller* $\rho$ (Cattaneo et al., 2024). We further validate these predictions empirically on GAN training across common architectures and datasets, a representative class of min–max games captured by our theory.

**Limitations.** A limitation of this work is that we primarily study Adam's dynamics in a deterministic setting. Nevertheless, our findings are informative for the stochastic regime. In particular, the experiments in Section 5 use stochastic Adam for GAN training, and the observed behavior is consistent with our theoretical predictions. We also note that, in minimization, deterministic analyses of Adam have been an important step and have yielded key insights, including the edge-of-stability phenomenon (Cohen et al., 2021; 2025) and implicit regularization effects (Wang et al., 2021; 2022; Zhang et al., 2024; Xie & Li, 2024). Since Adam in zero-sum games is substantially less understood than in minimization, we view the deterministic analysis here as a natural first step toward a complete theory.

## 1.2. Related Works

**ODEs in Optimization.** ODEs method was introduced to study momentum in minimization problems, starting with the seminal work of (Polyak, 1964). Recently, inspired by optimization algorithms that combine momentum and adaptivity such as Adam, ODEs or their stochastic generalization has also been applied to study adaptive methods in minimization, e.g., (Gadat & Gavra, 2022; Ma et al., 2022; Compagnoni et al., 2025). For zero-sum games, (Suh et al., 2023) employed this methodology to study the anchor acceleration methods. (Compagnoni et al., 2024b) developed stochastic differential equations for algorithms like Extra-gradient in zero-sum games under a stochastic setting. Two recent works, (Rosca et al., 2021) and (Feng et al., 2025) provide continuous-time analyses of vanilla Gradient Descent-Ascent and Heavy-ball momentum in zero-sum games, respectively. The current work generalizes these results to the Adam algorithm, which is more complex and remains poorly understood in game-theoretic settings. A more detailed comparison is provided in Appendix A.

**Local Convergence in zero-sum Games.** Local convergence of learning algorithms in zero-sum games has received great attention in recent years. (Liang & Stokes, 2019) analyzed the local convergence of the Gradient Descent-Ascent (GDA) algorithm and its variants, showing the importance of the interaction between players on the dynamics of the algorithms. (Fiez & Ratliff, 2021) provide a local convergence analysis of GDA under finite timescale separation. (Li et al., 2022a) studied the local convergence to a Stackelberg Equilibrium for GDA-based learning algorithms. (Zhang et al., 2022a) studied the local convergence of alternating GDA methods and demonstrated its near-optimal property. Recently, Wang & Chizat (2024) proved the surprising fact that partial curvature generically suffices for the local convergence of GDA.

**Implicit Gradient Regularization.** The implicit gradient regularization (IGR) effect was first developed for minimization problems. Barrett & Dherin (2021) studied IGR for the gradient descent algorithm. It was later extended to momentum methods by Ghosh et al. (2023), which shows that large momentum parameters usually make algorithms find flatter minima. Recently, Cattaneo et al. (2024) has extended these results to Adam.[1] For zero-sum games, (Rosca et al., 2021) first derived IGR for GDA algorithms, and (Feng et al., 2025) extended this approach to momentum methods. Inspired by these findings, several recent algorithms that explicitly incorporate gradient regularization to enhance performance have been proposed (Zhang et al., 2023a;b).

---

[1]The detailed discussion on Adam is in Appendix A.

## 2. Preliminaries

**Notations.** For matrix $\mathcal{M}$, $\mathrm{Sp}(\mathcal{M})$ denotes the set of its eigenvalues in $\mathbb{C}$. For $\lambda \in \mathrm{Sp}(\mathcal{M})$, $\Re(\lambda)$ and $\Im(\lambda)$ represent the real and imaginary parts of $\lambda$. The notation $\mathcal{M} \preccurlyeq \mathbf{0}$ or $\mathcal{M} \succcurlyeq \mathbf{0}$ means that $\mathcal{M}$ is a negative or positive semi-definite matrix. We use $\mathrm{EigVec}(\mathcal{M})$ to denote the eigenspace of $\mathcal{M}$, and $\mathrm{Ker}(\mathcal{M})$ to represent its kernel space, i.e., $\mathrm{Ker}(\mathcal{M}) = \{ \boldsymbol{z} \in \mathbb{C}^d \mid \mathcal{M}\boldsymbol{z} = \mathbf{0} \}$. $\mathcal{I}_d$ denotes the $d$-dimension identity matrix. $\mathrm{Diag}\{\boldsymbol{v}\}$ denotes a diagonal matrix with diagonal elements $\{\boldsymbol{v}_i\}_{i=1}^d$. Without specialization, we use the component-wise multiplication and division of vectors, as well as component-wise addition.

**Zero-Sum Games.** A zero-sum game with smooth loss function $f(\boldsymbol{x}, \boldsymbol{y})$ can be formulated as

$$\min_{\boldsymbol{x} \in \mathbb{R}^{d_1}} \max_{\boldsymbol{y} \in \mathbb{R}^{d_2}} f(\boldsymbol{x}, \boldsymbol{y}) \qquad \text{(Zero-Sum Games)}$$

If a pair of strategies $(\boldsymbol{x}^*, \boldsymbol{y}^*)$ satisfies $\forall \boldsymbol{x} \in \mathcal{U}$, $f(\boldsymbol{x}, \boldsymbol{y}^*) \geq f(\boldsymbol{x}^*, \boldsymbol{y}^*)$ and $\forall \boldsymbol{y} \in \mathcal{V}$, $f(\boldsymbol{x}^*, \boldsymbol{y}^*) \geq f(\boldsymbol{x}^*, \boldsymbol{y})$ for some $\boldsymbol{x}^*$'s neighborhood $\mathcal{U} \subseteq \mathbb{R}^{d_1}$ and $\boldsymbol{y}^*$'s neighborhood $\mathcal{V} \subseteq \mathbb{R}^{d_2}$, then $(\boldsymbol{x}^*, \boldsymbol{y}^*)$ is called a *local Nash equilibrium*. This is one of the most widely used solution concepts in zero-sum games, and is the focus of this work.

**Adam in Zero-Sum Games.** In zero-sum games, the $x$-player (resp. $y$-player) aims to minimize (resp. maximize) the objective function $f(\boldsymbol{x}, \boldsymbol{y})$. Accordingly, the Adam algorithm must be adapted, which result in the following Adam Descent-Ascent (Adam-DA) algorithm. In particular, the x-player updates according to

$$\tilde{\boldsymbol{v}}_{n+1} = \rho \tilde{\boldsymbol{v}}_n + (1-\rho)(\nabla_x f(\boldsymbol{x}_n, \boldsymbol{y}_n))^2,$$
$$\tilde{\boldsymbol{m}}_{n+1} = \beta \tilde{\boldsymbol{m}}_n + (1-\beta)\nabla_x f(\boldsymbol{x}_n, \boldsymbol{y}_n),$$
$$\boldsymbol{x}_{n+1} = \boldsymbol{x}_n - h \frac{\tilde{\boldsymbol{m}}_{n+1}/(1-\beta^{n+1})}{\sqrt{\tilde{\boldsymbol{v}}_{n+1}/(1-\rho^{n+1})} + \epsilon},$$

and the y-players updates according to

$$\hat{\boldsymbol{v}}_{n+1} = \rho \hat{\boldsymbol{v}}_n + (1-\rho)(\nabla_y f(\boldsymbol{x}_n, \boldsymbol{y}_n))^2,$$
$$\hat{\boldsymbol{m}}_{n+1} = \beta \hat{\boldsymbol{m}}_n + (1-\beta)\nabla_y f(\boldsymbol{x}_n, \boldsymbol{y}_n),$$
$$\boldsymbol{y}_{n+1} = \boldsymbol{y}_n + h \frac{\hat{\boldsymbol{m}}_{n+1}/(1-\beta^{n+1})}{\sqrt{\hat{\boldsymbol{v}}_{n+1}/(1-\rho^{n+1})} + \epsilon} \qquad \text{(Adam-DA)}$$

Here $h > 0$ is the step size, $\epsilon > 0$ is the numerical stability parameter, $\beta \in (-1, 1)$ is the first-order momentum factor and $\rho \in (0, 1)$ is the second-order momentum factor.

**Local Behaviors of Dynamical Systems.** For a system of differential equations $\dot{\boldsymbol{x}}(t) = g(x)$ where $g : \mathbb{R}^d \to \mathbb{R}^d$ is a differentiable function, let $\tilde{\boldsymbol{x}} \in \mathbb{R}^d$ satisfy $g(\tilde{\boldsymbol{x}}) = 0$. Then the local behavior of the system near $\tilde{\boldsymbol{x}}$ is determined by the eigenvalues of Jacobian $\mathcal{J}_g(\tilde{\boldsymbol{x}}) = \left( \frac{\partial g_i}{\partial x_j}(\tilde{\boldsymbol{x}}) \right)_{i,j}$:

**Proposition 2.1.** *(Khalil & Grizzle, 2002) Suppose that g is continuously differentiable. If $\alpha = \max_{\lambda \in \mathrm{Sp}(\mathcal{J}_g)} \Re(\lambda) < 0$, then there exist constants $\delta > 0$ and $C > 0$ such that for all initial conditions satisfying $\|\boldsymbol{x}(0) - \tilde{\boldsymbol{x}}\| \leq \delta$, we have $\|\boldsymbol{x}(t) - \tilde{\boldsymbol{x}}\| \leq C e^{t\alpha}, \forall t > 0$.*

## 3. Continuous-Time Model

In this section, we present our continuous-time models for Adam-DA. First, we establish an error bound between the trajectories of discrete-time algorithms and continuous-time models. Then, we compare our model with SignGDA-flow, the min-max adaptation of the continuous-time model proposed by (Ma et al., 2022) for Adam in minimization.

The continuous-time model we purposed for Adam-DA in zero-sum games is the following:

$$\dot{\boldsymbol{x}}(t) = -\mu_\epsilon(\boldsymbol{x}, \boldsymbol{y}) \Bigg( \nabla_x f(\boldsymbol{x}, \boldsymbol{y}) + \frac{h}{2} \mathcal{M}^\mu_{\beta, \rho, \epsilon}(\boldsymbol{x}, \boldsymbol{y}) \cdot$$
$$\nabla_x \Big( \|\nabla_x f(\boldsymbol{x}, \boldsymbol{y})\|_{1,\epsilon} - \|\nabla_y f(\boldsymbol{x}, \boldsymbol{y})\|_{1,\epsilon} \Big) \Bigg),$$

$$\dot{\boldsymbol{y}}(t) = \nu_\epsilon(\boldsymbol{x}, \boldsymbol{y}) \Bigg( \nabla_y f(\boldsymbol{x}, \boldsymbol{y}) + \frac{h}{2} \mathcal{M}^\nu_{\beta, \rho, \epsilon}(\boldsymbol{x}, \boldsymbol{y}) \cdot$$
$$\nabla_y \Big( \|\nabla_x f(\boldsymbol{x}, \boldsymbol{y})\|_{1,\epsilon} - \|\nabla_y f(\boldsymbol{x}, \boldsymbol{y})\|_{1,\epsilon} \Big) \Bigg).$$
$$\text{(Continuous Adam-DA)}$$

Here the perturbed $\ell_1$ norm[2] is defined as $\|\boldsymbol{v}\|_{1,\epsilon} := \sum_{i=1}^d \sqrt{\boldsymbol{v}_i^2 + \epsilon}$. Other terms are defined as:

- $\mu_\epsilon(\mathbf{x}, \mathbf{y}) := \mathrm{Diag}\left\{ \left( \|\partial_{x_j} f(\mathbf{x}, \mathbf{y})\|_{1,\epsilon}^{-1} \right)_{j=1}^{d_1} \right\}$

- $\mathcal{M}^\mu_{\beta, \rho, \epsilon} := \mathcal{K}(\beta, \rho)\mathcal{I}_{d_1} + \frac{\epsilon(1+\rho)}{1-\rho} \mu_\epsilon^2(\mathbf{x}, \mathbf{y})$

- $\nu_\epsilon(\mathbf{x}, \mathbf{y}) := \mathrm{Diag}\left\{ \left( \|\partial_{y_i} f(\mathbf{x}, \mathbf{y})\|_{1,\epsilon}^{-1} \right)_{i=1}^{d_2} \right\}$

- $\mathcal{M}^\nu_{\beta, \rho, \epsilon} := \mathcal{K}(\beta, \rho)\mathcal{I}_{d_2} + \frac{\epsilon(1+\rho)}{1-\rho} \nu_\epsilon^2(\mathbf{x}, \mathbf{y})$

where $\mathcal{K}(\beta, \rho) = (1+\beta)/(1-\beta) - (1+\rho)/(1-\rho)$.

**Theorem 3.1.** *Let $f(\boldsymbol{x}, \boldsymbol{y})$ be a smooth function with bounded derivatives up to fourth order. Then for any given finite time horizon, the solution trajectories $(\boldsymbol{x}(t), \boldsymbol{y}(t))$ of Continuous Adam-DA is locally $\mathcal{O}(h^3)$-close to the trajectories of Adam-DA after $\max\{ \frac{2\log h}{\log |\beta|}, \frac{2\log h}{\log \rho} \}$ steps.*

*Remark* 3.1. The bounded derivatives condition in Theorem 3.1 is necessary to prove rigorous results about the errors

---

[2]Strictly speaking, $\|\cdot\|_{1,\epsilon}$ is not a norm. The terminology used here follows (Cattaneo et al., 2024).

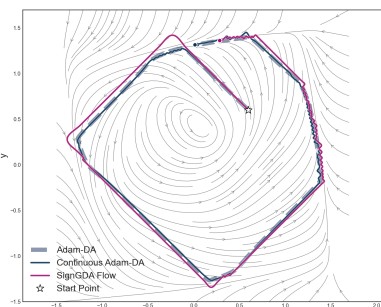 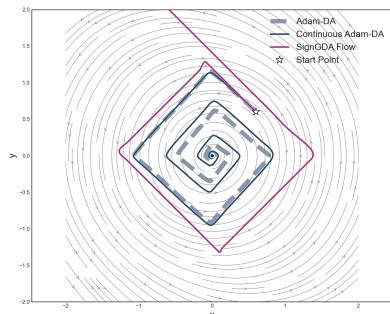 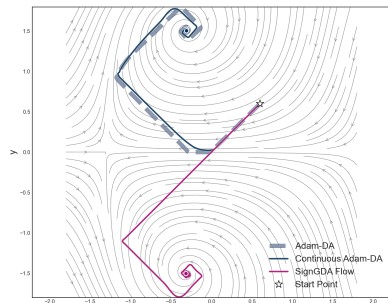

*Figure 1.* Trajectories of *Adam-DA, Continuous Adam-DA, and SignGDA-flow* on three test functions from (*Compagnoni et al., 2024b*). *Continuous Adam-DA* closely approximates *Adam-DA*. Especially in *1(b)* and *1(c)*, where *SignGDA-flow* either diverges or approaches to a different equilibrium, while the trajectories of the other two methods remain similar. More details are provided in Appendix *B.1*.

between ODEs and algorithms. It is widely used in related literature (Rosca et al., 2021; Ghosh et al., 2023; Compagnoni et al., 2024a). Indeed, as remarked by (Cattaneo et al., 2024), such assumptions are explicitly or implicitly present in all previous work on the backward error analysis of gradient-based machine learning algorithms.

The proof of Theorem 3.1 relies on the backward error analysis from the numerical analysis literature (Haier et al., 2006), and is inspired by (Cattaneo et al., 2024), which derived ODEs for the Adam algorithm in the minimization setting to study its implicit regularization effect. However, our equations differ significantly due to the presence of two interacting players inherent to zero-sum games. Detailed proof of Theorem 3.1 is provided in Appendix B.2.

Theorem 3.1 shows that Continuous Adam-DA approximates Adam-DA with a $\mathcal{O}(h^3)$-local error. To highlight this advantage, we compare it with SignGDA-flow, a min-max adaptation of the ODEs proposed by (Ma et al., 2022), which approximates Adam with a $\mathcal{O}(h^2)$-local error:

$$\dot{\boldsymbol{x}}(t) = -\mu_\epsilon(\boldsymbol{x}, \boldsymbol{y})\nabla_x f(\boldsymbol{x}, \boldsymbol{y}),$$
$$\dot{\boldsymbol{y}}(t) = \nu_\epsilon(\boldsymbol{x}, \boldsymbol{y})\nabla_y f(\boldsymbol{x}, \boldsymbol{y}) \qquad \text{(SignGDA-flow)}$$

The name of SignGDA-flow comes from the fact that when $\epsilon \approx 0$, SignGDA-flow depends only on the signs of the partial derivatives. In Figure 1, we present numerical examples from (Compagnoni et al., 2024b) to compare these three methods. It can be observed that Continuous Adam-DA better approximates Adam-DA, thus highlighting the benefits of $\mathcal{O}(h^3)$-local error.

## 4. Local Convergence

Unlike minimization algorithms, which almost always converge to a local minimum (Lee et al., 2019), algorithms for zero-sum games often fail to converge and may exhibit complex behaviors, such as cycles and chaos (Bailey et al., 2020; Cheung & Piliouras, 2020). However, recent studies

show that if the initial strategy pair is sufficiently close to a local equilibrium, convergence can still be achieved (Li et al., 2022a; Wang & Chizat, 2024; Zhang, 2025), which is known as *local convergence*. Building upon these findings, we are motivated to investigate the following question:

*How do the parameters in Continuous Adam-DA and Adam-DA influence their local convergence?*

In this section, we answer the above question. In Section 4.1, we describe the Jacobian structure of Continuous Adam-DA at a local equilibrium, which is our main tool to study local convergence. The main results are presented in Section 4.2. Most proofs are deferred to Appendix C.

### 4.1. Jacobian Structure of Adam-DA

From dynamical system theory, the Jacobian matrix of a dynamical system completely determines its local behavior around an equilibrium. We first introduce the Jacobian of the following ODE for Gradient Decent-Ascent (GDA), which serves as a fundamental tool for analyzing Continuous Adam-DA and Adam-DA:

$$\dot{\boldsymbol{x}}(t) = -\nabla_x f(\boldsymbol{x}, \boldsymbol{y}),$$
$$\dot{\boldsymbol{y}}(t) = \nabla_y f(\boldsymbol{x}, \boldsymbol{y}) \qquad \text{(Continuous GDA)}$$

Continuous GDA is the ODE for GDA (Wang & Chizat, 2024). The Jacobian of Continuous GDA at a local Nash equilibrium $(\boldsymbol{x}^*, \boldsymbol{y}^*)$ is defined by

$$\mathcal{J} = \begin{bmatrix} -\nabla_x^2 f(\boldsymbol{x}^*, \boldsymbol{y}^*) & -\nabla_{xy} f(\boldsymbol{x}^*, \boldsymbol{y}^*) \\ \nabla_{yx} f(\boldsymbol{x}^*, \boldsymbol{y}^*) & \nabla_y^2 f(\boldsymbol{x}^*, \boldsymbol{y}^*) \end{bmatrix} \quad \text{(Jacobian)}$$

Moreover, by the definition of local Nash equilibrium, we have $\nabla_x^2 f(\boldsymbol{x}^*, \boldsymbol{y}^*) \succcurlyeq \boldsymbol{0}$, $\nabla_y^2 f(\boldsymbol{x}^*, \boldsymbol{y}^*) \preccurlyeq \boldsymbol{0}$.

It may seem that Continuous GDA is an over-simplified version of Continuous Adam-DA. Surprisingly, their Jacobian are closely related, as shown in the following proposition:

**Proposition 4.1.** *Let $\mathcal{J}_{\mathrm{Adam}}$ be the Jacobian of Continuous Adam-DA at the local equilibrium $(\boldsymbol{x}^*, \boldsymbol{y}^*)$, then $\mathcal{J}_{Adam}$ is given by a quadratic polynomial of $\mathcal{J}$:*

$$\mathcal{J}_{\mathrm{Adam}} = \frac{1}{\sqrt{\epsilon}}\left(\mathcal{I} - \frac{h(1+\beta)}{2\sqrt{\epsilon}(1-\beta)}\mathcal{J}\right)\mathcal{J}.$$

This simple yet powerful relationship allows us to provide a detailed analysis of the local behavior of Continuous Adam-DA in the following section.

Finally, we recall the work of (Letcher et al., 2019) introduced a decomposition of $\mathcal{J}$ as a summation of its symmetry part and anti-symmetry part $\mathcal{J} = \mathcal{S} + \mathcal{A}$, where

$$\mathcal{S} = \begin{bmatrix} -\nabla_x^2 f(\boldsymbol{x}^*, \boldsymbol{y}^*) & \mathbf{0} \\ \mathbf{0}^\top & \nabla_y^2 f(\boldsymbol{x}^*, \boldsymbol{y}^*) \end{bmatrix}$$

and

$$\mathcal{A} = \begin{bmatrix} \mathbf{0} & -\nabla_{xy} f(\boldsymbol{x}^*, \boldsymbol{y}^*) \\ \nabla_{yx} f(\boldsymbol{x}^*, \boldsymbol{y}^*) & \mathbf{0}^\top \end{bmatrix}.$$

In this decomposition, the symmetric part $\mathcal{S}$ describes the part of the dynamics where players independently optimize their own loss functions. The anti-symmetric part $\mathcal{A}$ describes the part where players interact adversarially. In typical zero-sum games, the magnitude of $\mathcal{A}$ should dominate $\mathcal{S}$; otherwise, the game reduces to a minimization problem where both players minimize their losses independently.

## 4.2. Local Convergence of Adam in Zero-Sum Games

In this section, we present our findings regarding the local convergence. We offer results for both Continuous Adam-DA and Adam-DA, demonstrating that they display similar behavior. First, we present the following assumption in the literature that characterizes the class of zero-sum games that we are interested in within this work.

**Assumption 4.2.** *(Wang & Chizat, 2024) There exist at least one eigenvalue $\lambda \in \mathrm{Sp}(\mathcal{J})$ such that $|\Im(\lambda)| > |\Re(\lambda)|$, we denote the set of all such eigenvalues as $\widetilde{\mathrm{Sp}}(\mathcal{J})$. We also assume $\mathrm{EigVec}(\mathcal{A}) \cap \mathrm{Ker}(\mathcal{S}) = \{\boldsymbol{0}\}$.*

Note that $\mathrm{EigVec}(\mathcal{A}) \cap \mathrm{Ker}(\mathcal{S}) = \{\boldsymbol{0}\}$ holds **generically** in the following sense: for any fixed $\mathcal{S} \neq \mathbf{0}$, if $\mathcal{A}$ is sampled independently from an absolutely continuous distribution, then this condition holds with probability one. Thus it is a mild assumption.

The eigenvalue part of Assumption 4.2 is satisfied for zero-sum games in which the $\mathcal{A}$ in (Jacobian) dominates $\mathcal{S}$ in magnitude. This regime corresponds to a strong coupling between the two players' local dynamics near the

equilibrium. Note that these are also the most representative zero-sum games in the literature (Liang & Stokes, 2019; Wang & Chizat, 2024), in contrast to games that can essentially be treated as minimization problems, e.g., $\min_{\boldsymbol{x}} \max_{\boldsymbol{y}} f(\boldsymbol{x}) + g(\boldsymbol{y})$, where players do not interact.

**Theorem 4.3.** *Under Assumption 4.2, let $\beta \in (-1, 1)$ and $\rho \in (0, 1)$. Then Continuous Adam-DA achieves local convergence to $(\boldsymbol{x}^*, \boldsymbol{y}^*)$ with an exponential rate iff the step-size $h$ satisfies:*

$$0 < h < \min_{\lambda \in \widetilde{\mathrm{Sp}}(\mathcal{J})} \frac{2\sqrt{\epsilon}(1-\beta)|\Re(\lambda)|}{(1+\beta)(\Im(\lambda)^2 - \Re(\lambda)^2)}$$

Under Assumption 4.2, $\Im(\lambda)^2 - \Re(\lambda)^2 > 0$, so the upper bound is positive. By Proposition 2.1, the local convergence of Continuous Adam-DA is governed by the eigenvalues of $\mathcal{J}_{\mathrm{Adam}}$, and the convergence rate equals the largest real part of these eigenvalues. The polynomial relationship between $\mathcal{J}_{\mathrm{Adam}}$ and $\mathcal{J}$ in Proposition 4.1 is key to the proof. The full proof is in Appendix C.2.

It is interesting to ask how well the above theorem predicts the behavior of the original discrete-time Adam-DA. We establish the following local convergence result for Adam-DA, and the detailed proof is provided in Appendix C.3:

**Theorem 4.4.** *Under Assumption 4.2, let $\beta \in (-1, 1)$ and $\rho \in (0, 1)$. Then Adam-DA achieves local convergence to the local equilibrium $(\boldsymbol{x}^*, \boldsymbol{y}^*)$ with an exponential rate iff the step-size $h$ satisfies:*

$$0 < h < \min_{\lambda \in \mathrm{Sp}(\mathcal{J})} \frac{2\sqrt{\epsilon}(1-\beta^2)|\Re(\lambda)|}{(1+\beta^2)|\lambda|^2 + 2\beta(\Im(\lambda)^2 - \Re(\lambda)^2)} \tag{1}$$

By comparing Theorem 4.3 and Theorem 4.4, we observe that they share several common properties. One particularly interesting property is the following corollary, which shows that in both theorems there exists a trade-off between the parameters $h$ and $\beta$.

**Corollary 4.5.** *The upper bound in Theorem 4.3 is a decreasing function of $\beta \in (-1, 1)$. Similarly, the upper bound in Theorem 4.4 is also a decreasing function of $\beta$ if the minimizer of (1) is achieved in $\widetilde{\mathrm{Sp}}(\mathcal{J})$ and $\beta$ satisfies*

$$\max_{\lambda \in \widetilde{\mathrm{Sp}}(\mathcal{J})} \frac{(|\Re(\lambda)| - |\Im(\lambda)|)}{(|\Re(\lambda)| + |\Im(\lambda)|)} < \beta < 1, \tag{2}$$

*Therefore, within these regions, using a **smaller** $\beta$ allows Continuous Adam-DA and Adam-DA to achieve local convergence across a broader range of step sizes $h$.*

The interval in (2) can be very close to $(-1, 1)$ when $|\Im(\lambda)|$ is much larger than $|\Re(\lambda)|$, as is the case in typical games such as bilinear games and GANs. Corollary 4.5 reflects a

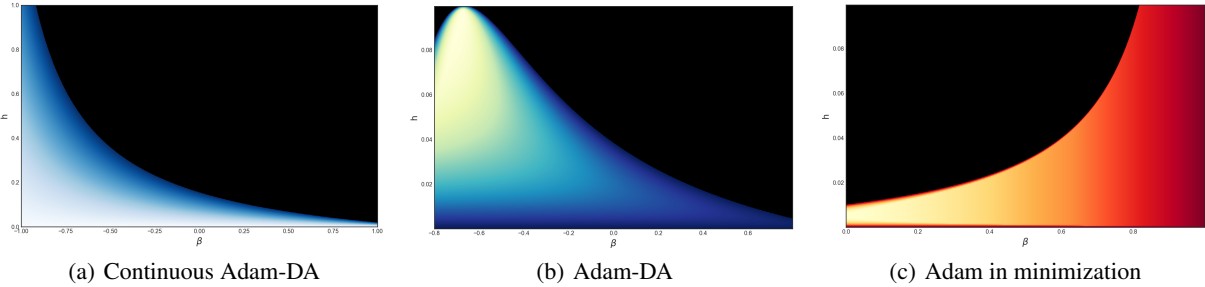

(a) Continuous Adam-DA            (b) Adam-DA            (c) Adam in minimization

*Figure 2. Numerical experiments on quadratic test functions for the local convergence of Continuous Adam-DA, Adam-DA in zero-sum games, and Adam in minimization. The **black** regions indicate divergence, while lighter regions indicate faster convergence. The X-axis is β and Y-axis is h. **Smaller** β values allow Continuous Adam-DA and Adam-DA to converge over a wider range of step sizes, while **larger** β allow larger h in minimization. These results support Corollary 4.5 and highlight fundamental differences in the role of momentum β between min-max and minimization problems. The non-decreasing behavior of β for extremely small β in 2(b) aligns with Corollary 4.5.*

fundamental difference in the role of $\beta$ between zero-sum games and minimization. In minimization problems, it is known that a larger $\beta$ can enable Adam to achieve local convergence with a wider range of step sizes (O'Donoghue & Candes, 2015; Bock & Weiß, 2019). However, as shown by Corollary 4.5, to achieve a similar effect, we need to use a smaller $\beta$ in zero-sum games. We illustrate this point in Figure 2. More experiments are presented in Appendix C.4.

Finally, Theorem 4.3 and 4.4 imply Adam-DA always diverge in the important class of bilinear games:

**Corollary 4.6.** *For bilinear games $f(\boldsymbol{x}, \boldsymbol{y}) = \boldsymbol{x}^\top A \boldsymbol{y}$, Continuous Adam-DA and Adam-DA always diverge regardless of the choice of parameters $\beta$, $\rho$, $\epsilon$ and $h$.*

Corollary 4.6 extends the previous results on the divergence of GDA and MWU in the bilinear zero-sum games (Bailey & Piliouras, 2018; Bailey et al., 2020) to the Adam algorithm. This corollary reflects another difference between Adam-DA in zero-sum games and Adam in minimization. In minimization, Adam can always converge to a neighborhood of a local minimum with carefully chosen parameters (Zhang et al., 2022b). However, Corollary 4.6 shows that Adam-DA always diverges in the fundamental class of bilinear games.

## 5. Implicit Gradient Regularization

Implicit gradient regularization (IGR) refers to the terms involving the gradients of loss functions that emerge from ODEs of the discrete-time algorithms. One example is that in minimization, Barrett & Dherin (2021) derived the following $\mathcal{O}(h^3)$-local error ODEs for the Gradient Descent algorithm (GD) $\boldsymbol{x}_{t+1} = \boldsymbol{x}_t - h\nabla_x f(\boldsymbol{x}_t)$:

$$\dot{\boldsymbol{x}}(t) = -\nabla_x f(\boldsymbol{x}) - (h/4)\nabla_x \|\nabla_x f(\boldsymbol{x})\|_2^2, \quad (3)$$

The IGR term in (3) is $-(h/4)\nabla_x \|\nabla_x f(\boldsymbol{x})\|_2^2$, which reveals that besides minimizing function value $f(\boldsymbol{x})$, GD also implicitly minimizing $\|\nabla_x f(\boldsymbol{x})\|_2^2$, thereby guiding the op-

timization trajectories towards flatter region of the loss landscapes. Here we use the terminology *flatter* regions to refer to regions in the loss landscape with smaller gradient norm, following (Barrett & Dherin, 2021). Recently, (Cattaneo et al., 2024) generalize this approach to the Adam algorithm in minimization. They found in minimization problems, the trajectories of Adam explore flatter regions of the loss landscape with a **larger** $\beta$ and a **smaller** $\rho$. Motivated by these findings, we ask the question:

*How the parameters $\beta$ and $\rho$ affect the interaction between the trajectories of Adam-DA and the flatter regions of loss landscapes in zero-sum games?*

As in Section 4, here we restrict our attention to typical zero-sum games that are distinct from pure minimization problems. We focus on adversarial interaction-dominated games where $\nabla_{xy} f(\mathbf{x}, \mathbf{y})$—which characterizes the adversarial interaction between players' dynamics in (Jacobian)—dominates the terms $\nabla_x^2 f(\mathbf{x}, \mathbf{y})$ and $\nabla_y^2 f(\mathbf{x}, \mathbf{y})$ in magnitude along the training trajectory. This property can be assessed by examining the Jacobian spectrum along the algorithm's iterates: in interaction-dominated games, the eigenvalues typically exhibit large imaginary parts, reflecting the antisymmetric structure induced by the coupling between the two players (Liang & Stokes, 2019). In the literature, we found that the GAN training problem most closely satisfies this interaction-dominated condition, where eigenvalues of the Jacobian with large imaginary parts are identified as a main cause of GAN training instability (Nagarajan & Kolter, 2017).

Our goal in this section is to argue the following thesis, which highlights the distinct behavior of IGR with Adam-DA in zero-sum games compared to Adam in minimization (Cattaneo et al., 2024):

**_Thesis._** In zero-sum games where players' interactions dominate the dynamics, *smaller* $\beta$ and *larger* $\rho$ guides trajectories of Adam-DA towards flatter regions of loss land-

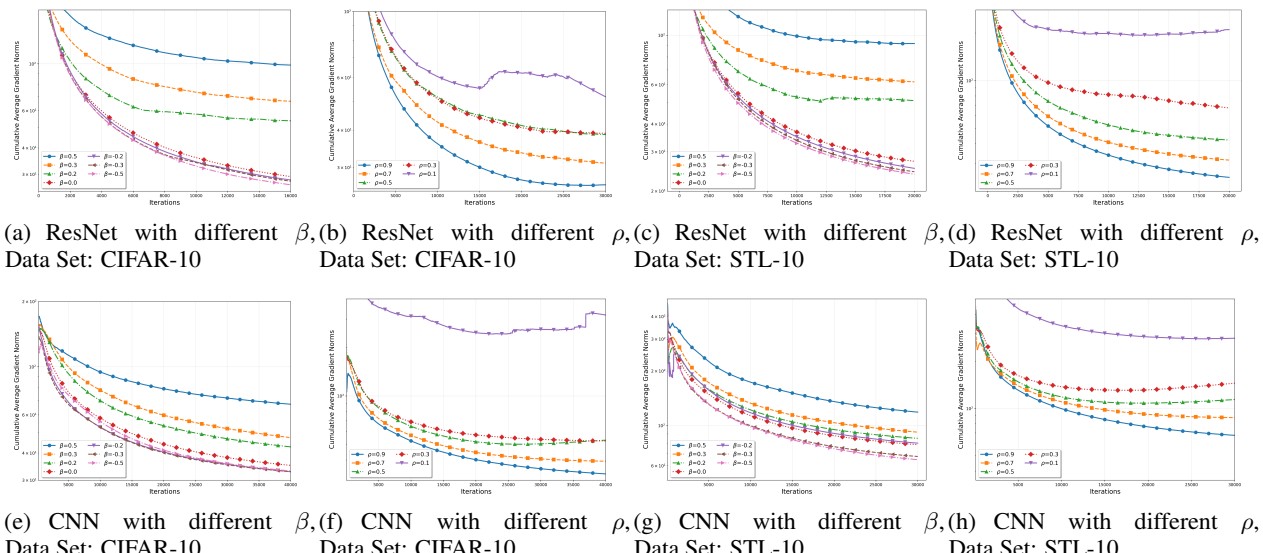

(a) ResNet with different $\beta$, Data Set: CIFAR-10 (b) ResNet with different $\rho$, Data Set: CIFAR-10 (c) ResNet with different $\beta$, Data Set: STL-10 (d) ResNet with different $\rho$, Data Set: STL-10

(e) CNN with different $\beta$, Data Set: CIFAR-10 (f) CNN with different $\rho$, Data Set: CIFAR-10 (g) CNN with different $\beta$, Data Set: STL-10 (h) CNN with different $\rho$, Data Set: STL-10

*Figure 3. The $\ell_1$ norm of gradients of Adam-DA with varying $\beta$ and $\rho$ during GANs training. Datasets: CIFAR-10 and STL-10. Architectures: ResNet and CNN. As shown in 3(a), 3(c), 3(e), and 3(g), smaller $\beta$ values result in smaller gradient norms. According to 3(b), 3(d), 3(f), and 3(h), larger $\rho$ values also lead to smaller gradient norms. Both findings support the thesis.*

scapes in terms of $\ell_1$ norm, i.e., regions with lower values of $\|\nabla_x f\|_1 + \|\nabla_y f\|_1$.

We present observations from Continuous Adam-DA that motivate our thesis in Section 5.1, and verify it experimentally in Section 5.2.

### 5.1. Observations from ODEs

The interesting scenario in understanding implicit gradient regularization occurs when $\epsilon$ is much smaller than the gradient norms[3]. In this scenario, we take $\epsilon \to 0$, and Continuous Adam-DA reduces to the following equations:

$$\dot{\boldsymbol{x}}(t) = \mu_\epsilon(\boldsymbol{x}, \boldsymbol{y}) \cdot \Big[ -\nabla_x f(\boldsymbol{x}, \boldsymbol{y})$$
$$+ \frac{h}{2} \mathcal{K}(\beta, \rho) \nabla_x \Big( \|\nabla_y f(\boldsymbol{x}, \boldsymbol{y})\|_{1,\epsilon} - \|\nabla_x f(\boldsymbol{x}, \boldsymbol{y})\|_{1,\epsilon} \Big) \Big],$$
$$\dot{\boldsymbol{y}}(t) = \nu_\epsilon(\boldsymbol{x}, \boldsymbol{y}) \cdot \Big[ \nabla_y f(\boldsymbol{x}, \boldsymbol{y})$$
$$+ \frac{h}{2} \mathcal{K}(\beta, \rho) \nabla_y \Big( \|\nabla_x f(\boldsymbol{x}, \boldsymbol{y})\|_{1,\epsilon} - \|\nabla_y f(\boldsymbol{x}, \boldsymbol{y})\|_{1,\epsilon} \Big) \Big].$$

where $\mathcal{K}(\beta, \rho) = (1+\beta)/(1-\beta) - (1+\rho)/(1-\rho)$ as in Section 3. Here we continue to use $\ell_{1,\epsilon}$ and $\mu_\epsilon, \nu_\epsilon$ to avoid the non-differentiable problem of the $\ell_1$ norm and the zero denominator problem.

We consider terms about $\|\nabla_x f(\boldsymbol{x}, \boldsymbol{y})\|_{1,\epsilon}$. There are two factors affecting its evolution in above equations: The $x$-player's equation include a gradient *descent* term on

$\mathcal{K}(\beta, \rho) \|\nabla_x f(\boldsymbol{x}, \boldsymbol{y})\|_{1,\epsilon}$, which aims to *minimize* it:

$$-\frac{h}{2} \mathcal{K}(\beta, \rho) \nabla_x \|\nabla_x f(\boldsymbol{x}, \boldsymbol{y})\|_{1,\epsilon} =$$
$$-\frac{h}{2} \mathcal{K}(\beta, \rho) \left( \nabla_x^2 f(\boldsymbol{x}, \boldsymbol{y}) \cdot \mu_\epsilon(\boldsymbol{x}, \boldsymbol{y}) \cdot \nabla_x f(\boldsymbol{x}, \boldsymbol{y}) \right), \quad (4)$$

while $y$-player's equation include a gradient *ascent* term to *maximize* it:

$$\frac{h}{2} \mathcal{K}(\beta, \rho) \nabla_y \|\nabla_x f(\boldsymbol{x}, \boldsymbol{y})\|_{1,\epsilon} =$$
$$\frac{h}{2} \mathcal{K}(\beta, \rho) \left( \nabla_{yx} f(\boldsymbol{x}, \boldsymbol{y}) \cdot \mu_\epsilon(\boldsymbol{x}, \boldsymbol{y}) \cdot \nabla_x f(\boldsymbol{x}, \boldsymbol{y}) \right). \quad (5)$$

The proof of (4) and (5) are provided in Appendix C.1. By comparing (4) and (5), we can predict that the evolution of $\|\nabla_x f(\boldsymbol{x}, \boldsymbol{y})\|_{1,\epsilon}$ in the competition between the two players depends on the relative sizes of $\nabla_x^2 f(\boldsymbol{x}, \boldsymbol{y})$ and $\nabla_{yx} f(\boldsymbol{x}, \boldsymbol{y})$. Moreover, under the condition that the players' interactions dominate the overall dynamics, we have $\nabla_{yx} f(\boldsymbol{x}, \boldsymbol{y}) \gg \nabla_x^2 f(\boldsymbol{x}, \boldsymbol{y})$, i.e., the matrix $\nabla_{yx} f(\boldsymbol{x}, \boldsymbol{y})$ dominates $\nabla_x^2 f(\boldsymbol{x}, \boldsymbol{y})$ in magnitude. This indicates that the $x$-player's impact on the evolution of $\|\nabla_x f(\boldsymbol{x}, \boldsymbol{y})\|_{1,\epsilon}$ is negligible compared to the $y$-player's, so we can focus on (5) to understand how various parameters affect its evolution. From (5), the IGR causes the $y$-player to perform gradient ascent on $\|\nabla_x f(\boldsymbol{x}, \boldsymbol{y})\|_{1,\epsilon}$, effectively trying to maximize it. To encourage the trajectories of Adam-DA to explore flatter regions of the loss landscapes, this effect should be reduced. This can be achieved by selecting a smaller $\beta$ and a larger $\rho$, which lower $\mathcal{K}(\beta, \rho)$; adjusting these parameters accordingly helps minimize its value.

---

[3]The default value of $\epsilon$ in PyTorch is $1 \times 10^{-8}$. In practice, gradient norms are much larger than this.

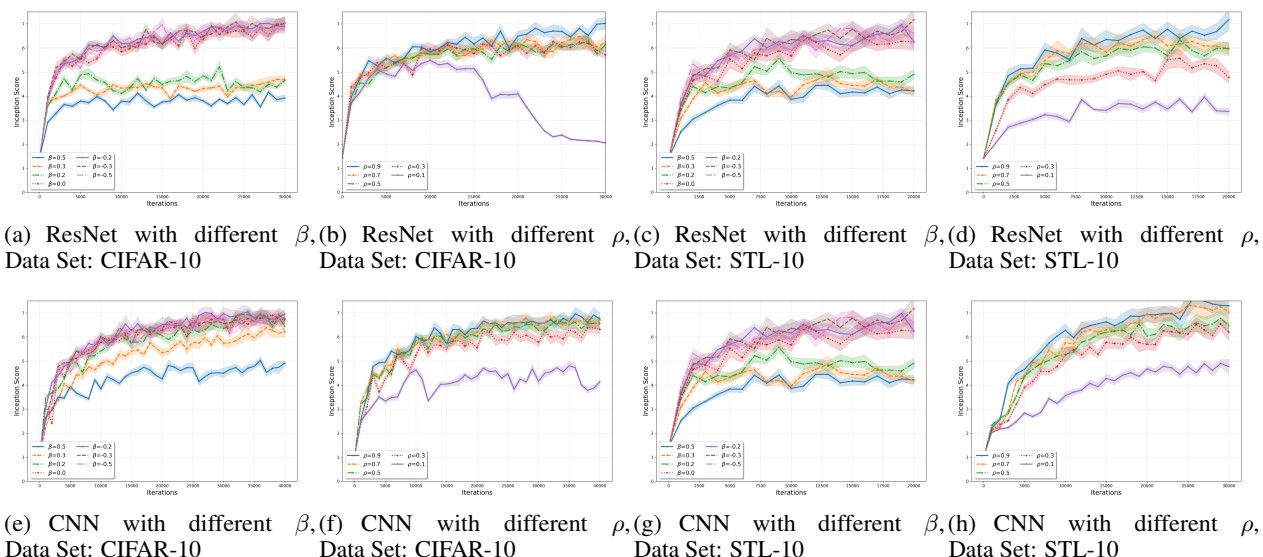

(a) ResNet with different $\beta$, Data Set: CIFAR-10 (b) ResNet with different $\rho$, Data Set: CIFAR-10 (c) ResNet with different $\beta$, Data Set: STL-10 (d) ResNet with different $\rho$, Data Set: STL-10

(e) CNN with different $\beta$, Data Set: CIFAR-10 (f) CNN with different $\rho$, Data Set: CIFAR-10 (g) CNN with different $\beta$, Data Set: STL-10 (h) CNN with different $\rho$, Data Set: STL-10

*Figure 4. Inception Score for the corresponding experimental settings in Figure 3. A high score implies better training performance. Training algorithms with lower $\ell_1$ gradient norms also have better performance.*

The above argument also applies to the evolution of $\|\nabla_y f(\boldsymbol{x}, \boldsymbol{y})\|_{1,\epsilon}$, due to the symmetry between the two players. In particular, a smaller $\beta$ and a larger $\rho$ are expected to result in lower values of $\|\nabla_y f(\boldsymbol{x}, \boldsymbol{y})\|_{1,\epsilon}$ along the trajectories. Thus, we obtain a justification of the thesis proposed above through Continuous Adam-DA.

### 5.2. Experiments

This section presents experimental results on GAN training to support our thesis. GAN training is a zero-sum game with strong adversarial interactions, aligning well with our thesis assumptions. Our goal here is **not** to outperform existing methods, but to validate the IGR effect of Adam-DA. The code for the experiments is publicly available online.[4]

**Experimental Setting.** Our setup generally follows the improved Wasserstein GANs framework (Gulrajani et al., 2017). Data sets include CIFAR-10 ($32 \times 32$ resolution) and STL-10 ($64 \times 64$ resolution). Network architectures are ResNet and CNN. We train GANs using the Adam-DA with different $\beta$ and $\rho$. Both generator and discriminator use the learning rate $2 \times 10^{-4}$. The batch size is $64$. Future details are presented in Appendix D. We use the following cumulative average gradient norms for visualization:

$$\text{AvgS}_{\beta,\rho}(t) = \frac{1}{t} \sum_{s=1}^{t} \left( \|\nabla_x f(\boldsymbol{x}_s, \boldsymbol{y}_s)\|_1 + \|\nabla_y f(\boldsymbol{x}_s, \boldsymbol{y}_s)\|_1 \right).$$

Smaller $\text{AvgS}_{\beta,\rho}$ indicate that the algorithms' trajectories are exploring flatter regions of loss landscapes.

[4] https://github.com/FY123123/adam-zero-sum-games

**Experimental Results.** In Figure 3, we show how the cumulative average gradient norms change during training. To evaluate the impact of $\beta$, we fix a value of $\rho = 0.9$ and plot the evolution of these norms for different $\beta$ values in Figures 3(a), 3(c), 3(e), and 3(g). We observe that smaller $\beta$ values lead to smaller gradient norms, supporting our thesis about the role of $\beta$. Similarly, to assess the effect of $\rho$, we fix $\beta = 0$ and plot the norms under different $\rho$ values in Figures 3(b), 3(d), 3(f), and 3(h). Larger $\rho$ values result in smaller gradient norms, which supports our thesis regarding $\rho$. The experimental results are consistency across different architectures and data sets.

In Figure 4, we present the Inception Scores (IS) of various GANs trained by Adam-DA. IS is a widely used metric for evaluating the performance of GAN training, with higher scores indicating better performance (Salimans et al., 2016). We find models that exhibit lower cumulative average gradient norms tend to achieve higher IS. This is analogous to the phenomena observed in minimization problems, where flatter regions tend to generalize better (Foret et al., 2021).

## 6. Conclusions

In this paper, we develop an ODE-based framework for analyzing the dynamics of Adam-DA in zero-sum games. We focus on how the algorithm's parameters shape two key aspects of the dynamics, namely local convergence and implicit gradient regularization. For both aspects, we show that these parameters influence Adam-DA in a way that is exactly opposite to Adam in minimization problems. This provides a theoretical explanation for empirical observations

such as the benefits of negative momentum in GAN training (Gidel et al., 2019), and it extends prior separation results between game dynamics and minimization dynamics from simpler methods without adaptivity or momentum (Bailey & Piliouras, 2018; Bailey et al., 2020) to the Adam-DA algorithm. An interesting direction in the future is to study dynamics of Adam in more complex and practical settings, such as the multi-player general-sum games, where the roles of momentum and adaptivity are far from well-understood.

## Acknowledgements

This work was supported by Danish Data Science Academy, which is funded by the Novo Nordisk Foundation (NNF21SA0069429). The research of Xiao Wang is supported by National Key R&D Program of China (2023YFA1009500), the Fundamental Research Funds for the Central Universities, Shanghai Sci-tech Co-research Program (25HB2703800). The research of Weiming Ou is supported by the Fundamental Research Funds for the Central Universities. The authors thank Chang He and Ping Li for their feedback on early drafts of this work. They also thank the anonymous reviewer of ICML 2026 for their valuable suggestions.

## Impact Statement

This paper presents work whose goal is to advance the field of machine learning. There are many potential societal consequences of our work, none of which we feel must be specifically highlighted here.

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

## A. More discussion on related works

**Comparison with (Rosca et al., 2021) and (Feng et al., 2025).** We make a detailed comparison with (Rosca et al., 2021) and (Feng et al., 2025), which studied the dynamics of Gradient Descent-Ascent method and Heavy Ball method in zero-sum games through the lens of ODEs in the following. The work of (Rosca et al., 2021) investigated implicit regularization of GDA in two-player general-sum games. However, their GDA algorithm involves neither adaptive stepsizes nor momentum parameters. Next, the work of (Feng et al., 2025) studied relatively simple Heavy-ball momentum method, which involves only a non-adaptive momentum parameter. Adam's adaptive structure substantially increases the complexity of both the ODEs and the discrete-time analysis. For instance, deriving an appropriate $\mathcal{O}(h^3)$ local error equation is significantly more involved than in the Heavy-ball case, which in turn makes the Jacobian analysis considerably more difficult. Our analysis yield new insights that cannot be inferred from previous works of (Rosca et al., 2021; Feng et al., 2025). In particular:

- Discrete-time vs. continuous-time behavior: We give a local convergence result for the continuous-time model and also analyze the original discrete-time Adam-DA algorithms. The discrete-time results show that the continuous-time model matches the algorithms well in usual settings. But when the momentum $\beta$ is very small, this match becomes weak, which highlight the potential failure of continuous-time models that do not appear in previous works.

- The role of $\rho$ : Heavy-ball momentum is non-adaptive, so the influence of $\rho$ is entirely absent in (Feng et al., 2025). In contrast, we show that in min–max games, larger $\rho$ amplifies Adam's implicit regularization effect—in sharp contrast to the minimization setting, where (Cattaneo et al., 2024) shows that smaller $\rho$ strengthens implicit regularization. This is a fundamentally new phenomenon specific to the min–max setting.

- The role of $\epsilon$ : Traditionally, $\epsilon$ is viewed merely as a term preventing division by zero and is assumed to be extremely small. In fact, using large $\epsilon$ is often considered inconsistent with the design philosophy of adaptive gradient methods (Zhang et al., 2022b). However, Theorems 4.3 and 4.4 demonstrate that a large $\epsilon$ can enhance local convergence in min–max games—again revealing behavior entirely absent in the Heavy-ball analysis.

**More Related Works in Game Dynamics.** Besides converging to local nash equilibrium, other properties of zero-sum games has been researched. Early rigorous characterizations of local optimality and stability in this setting were developed by (Jin et al., 2020), who formalized notions of local min–max optimality in nonconvex–nonconcave landscapes. (Fiez et al., 2021) investigated the non-asymptotic convergence rates to $\epsilon$-critical points in several classes of zero-sum games. At the same time, (Hsieh et al., 2021) exposed limitations of common min–max algorithms by showing possible convergence to spurious, non-critical sets, highlighting the delicate geometry of min–max flows. Parallel to these algorithmic and discrete-time analyses, recent work has explored adaptive time-scale strategies: (Li et al., 2022b) proposed Tiada, a time-scale adaptive method for nonconvex minimax problems, and (Yang et al., 2022) developed nested adaptive schemes that achieve parameter-agnostic convergence—complementary approaches that adaptively tune per-variable step sizes and thereby mitigate the need for manual time-scale separation.

**Related work on Adam.** The evolution of adaptive optimization began with AdaGrad (Duchi et al., 2011), which normalizes updates via the sum of squared gradients to optimize for sparse data. While theoretically sound, AdaGrad's monotonic accumulation leads to premature learning rate decay in non-convex regimes. To address this, RMSProp (Tieleman, 2012) introduced an exponential moving average (EMA) to curb the rapid decline. The seminal Adam optimizer (Kingma & Ba, 2014a) later integrated this adaptive scaling with momentum-based acceleration, tracking both first and second gradient moments to achieve superior stability in training deep neural networks.

Although Adam dominates empirically, theoretical convergence concerns led to the development of AMSGrad (Reddi et al., 2018), which imposes a non-decreasing constraint on the denominator. In terms of analysis, early studies relied on restrictive assumptions like bounded gradients (Zaheer et al., 2018; Chen et al., 2018; Zou et al., 2019; Défossez et al., 2020). Contemporary research, however, has pivoted toward high-probability guarantees under looser conditions: (Li et al., 2023) allows for sub-Gaussian noise, (Hong & Lin, 2024) accommodates affine noise, and (Wang et al., 2023) has proven that Adam meets fundamental lower bounds. Extensions to finite-sum settings have also been established (Zhang et al., 2022b; Wang et al., 2024).

## B. Additional Materials for Section 3

### B.1. Future Details of Figure 1

The test functions in Figure 1 comes from (Compagnoni et al., 2024b):

- (Figure in the left:) $f_1(x, y) = x(y - 0.45) + \phi(x) - \phi(y)$, $\phi(z) = \frac{1}{4}z^2 - \frac{1}{2}z^4 + \frac{1}{6}z^6$.

  Initial point: $(x_0, y_0) = (0.6, 0.6)$. $\beta = 0$, $\rho = 0.5$, $\epsilon = 10^{-6}$. Step size = 0.007.

- (Figure in the middle:)$f_2(x, y) = xy - \frac{1}{10}(\frac{1}{2}y^2 - \frac{1}{4}y^4)$

  Initial point: $(x_0, y_0) = (0.6, 0.6)$. $\beta = -0.3$, $\rho = 0.9$, $\epsilon = 10^{-3}$. Step size = 0.002.

- (Figure in the right:)$f_3(x, y) = \frac{1}{10}x^2 - \frac{1}{10}y^2 + \sin(x)\cos(y)$

  Initial point: $(x_0, y_0) = (0.6, 0.6)$. $\beta = 0.3$, $\rho = 0.5$, $\epsilon = 10^{-4}$. Step size = 0.005.

In the following we also present 30 random initial conditions, and draw the distance between the trajectories of Continuous Adam-DA, SignGDA-flow with Adam-DA. It can be observed that Continuous Adam-DA can better approximate Adam-DA than SignGDA-flow.

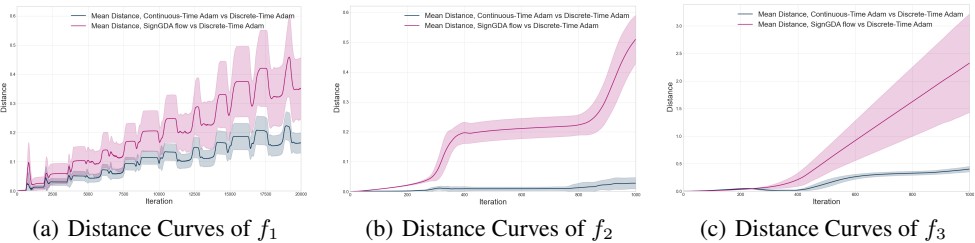

| (a) Distance Curves of $f_1$ | (b) Distance Curves of $f_2$ | (c) Distance Curves of $f_3$ |

*Figure 5. Figures 5(a), 5(b), and 5(c) show the distances of two continuous-time models between Adam, with results averaged over 30 random initial conditions.*

In the following Figure 6, we present additional experiments with different choice of parameters. The initial points are are chosen to be $(0.3, -0.3)$ in figures of trajectories. For the left figure, $\beta = 0.5, \rho = 0.8, \epsilon = 10^{-6}, h = 0.007$. For the middle figure, $\beta = 0.9, \rho = 0.8, \epsilon = 10^{-3}, h = 0.002$. For the right figure, $\beta = -0.1, \rho = 0.6, \epsilon = 10^{-4}, h = 0.005$

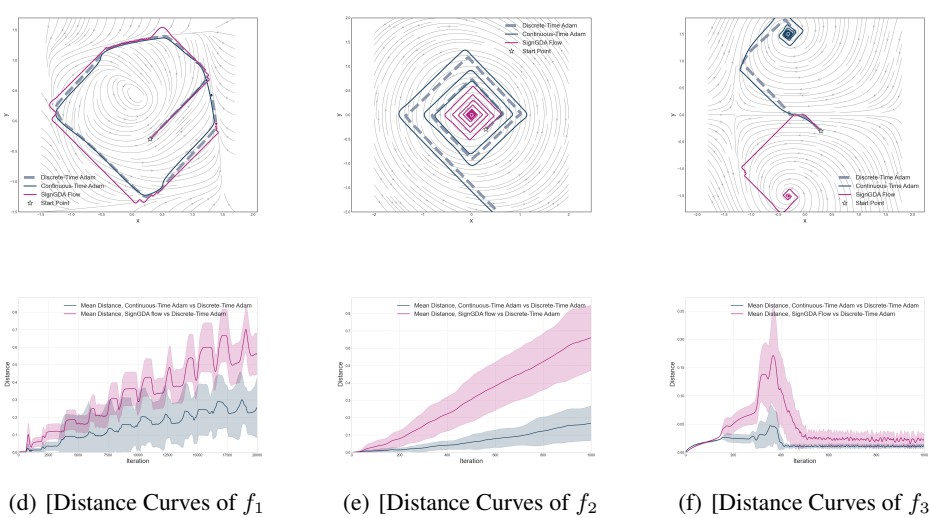

| (d) [Distance Curves of $f_1$ | (e) [Distance Curves of $f_2$ | (f) [Distance Curves of $f_3$ |

*Figure 6. Additional experiments with different parameters.*

### B.2. Proof of Theorem 3.1

**Notation Statement.** In Appendix B.2, component index and iteration index are used in many quantities. To avoid ambiguity, we use superscripts as coordinate indices (e.g., $\boldsymbol{x}^{(j)}$) and subscripts as iteration indices (e.g., $A_n(\boldsymbol{x}, \boldsymbol{y})$). The derivation of ODEs in Appendix B.2 is written by coordinate-wise, which is the same as the matrix form in Theorem 3.1.

Before providing the formal proof of Theorem 3.1, we provide the sketched proof of Theorem 3.1 consists of several steps:

- Step 1: In Lemma B.2, we derive a family of ODEs that each ODE corresponds to a time region $[t_n, t_{n+1})$ with length $h$ ($h$ is the stepsize of Adam-DA). The solutions of these ODEs achieves good approximation of Adam-DA with local error $\mathcal{O}(h^3)$. It should be pointed out that these ODEs may be not differentiable at time nodes and they are dependent on $n$, which hinder the analysis.

- Step 2: In Lemma B.4, we prove that the family of ODEs in Step 1 will converge to the single equation Continuous Adam-DA with an exponential rate. Due to the exponential rate, after a short time region, we can bound the local error between the family of ODEs and Continuous Adam-DA by $\mathcal{O}(h^3)$. The single ODE Continuous Adam-DA is differentiable everywhere and independent of $n$, which can simplify the analysis of the parameters' behaviors in Zero-Sum Games.

Define

$$N_n^{\boldsymbol{x}^{(j)}}(\boldsymbol{x}(t_n), \boldsymbol{y}(t_n)) = \frac{\partial f(\boldsymbol{x}(t_n), \boldsymbol{y}(t_n))}{\partial \boldsymbol{x}^{(j)}}, \ D_n^{\boldsymbol{x}^{(j)}}(\boldsymbol{x}(t_n), \boldsymbol{y}(t_n)) = \sqrt{\left(\frac{\partial f(\boldsymbol{x}(t_n), \boldsymbol{y}(t_n))}{\partial \boldsymbol{x}^{(j)}}\right)^2 + \epsilon},$$

$$N_n^{\boldsymbol{y}^{(i)}}(\boldsymbol{x}(t_n), \boldsymbol{y}(t_n)) = \frac{\partial f(\boldsymbol{x}(t_n), \boldsymbol{y}(t_n))}{\partial \boldsymbol{y}^{(i)}}, \ D_n^{\boldsymbol{y}^{(i)}}(\boldsymbol{x}(t_n), \boldsymbol{y}(t_n)) = \sqrt{\left(\frac{\partial f(\boldsymbol{x}(t_n), \boldsymbol{y}(t_n))}{\partial \boldsymbol{y}^{(i)}}\right)^2 + \epsilon},$$

$$Q_n^{\boldsymbol{x}^{(j)}}(\boldsymbol{x}(t_n), \boldsymbol{y}(t_n))$$
$$= \left(\frac{\beta}{1-\beta} - \frac{(n+1)\beta^{n+1}}{1-\beta^{n+1}}\right) \left[\sum_{s=1}^{d_1} \frac{\partial^2 f(\boldsymbol{x}(t_n), \boldsymbol{y}(t_n))}{\partial \boldsymbol{x}^{(j)} \partial \boldsymbol{x}^{(s)}} \frac{N_n^{\boldsymbol{x}^{(s)}}}{D_n^{\boldsymbol{x}^{(s)}}} - \sum_{s=1}^{d_2} \frac{\partial^2 f(\boldsymbol{x}(t_n), \boldsymbol{y}(t_n))}{\partial \boldsymbol{x}^{(j)} \partial \boldsymbol{y}^{(s)}} \frac{N_n^{\boldsymbol{y}^{(s)}}}{D_n^{\boldsymbol{y}^{(s)}}}\right],$$

$$Q_n^{\boldsymbol{y}^{(i)}}(\boldsymbol{x}(t_n), \boldsymbol{y}(t_n))$$
$$= \left(\frac{\beta}{1-\beta} - \frac{(n+1)\beta^{n+1}}{1-\beta^{n+1}}\right) \left[\sum_{s=1}^{d_1} \frac{\partial^2 f(\boldsymbol{x}(t_n), \boldsymbol{y}(t_n))}{\partial \boldsymbol{y}^{(i)} \partial \boldsymbol{x}^{(s)}} \frac{N_n^{\boldsymbol{x}^{(s)}}}{D_n^{\boldsymbol{x}^{(s)}}} - \sum_{s=1}^{d_2} \frac{\partial^2 f(\boldsymbol{x}(t_n), \boldsymbol{y}(t_n))}{\partial \boldsymbol{y}^{(i)} \partial \boldsymbol{y}^{(s)}} \frac{N_n^{\boldsymbol{y}^{(s)}}}{D_n^{\boldsymbol{y}^{(s)}}}\right], \tag{6}$$

$$P_n^{\boldsymbol{x}^{(j)}}(\boldsymbol{x}(t_n), \boldsymbol{y}(t_n)) = \left(\frac{\rho}{1-\rho} - \frac{(n+1)\rho^{n+1}}{1-\rho^{n+1}}\right) \left[\sum_{s=1}^{d_1} \frac{\partial^2 f(\boldsymbol{x}(t_n), \boldsymbol{y}(t_n))}{\partial \boldsymbol{x}^{(j)} \partial \boldsymbol{x}^{(s)}} \frac{\partial f(\boldsymbol{x}(t_n), \boldsymbol{y}(t_n))}{\partial \boldsymbol{x}^{(j)}} \frac{N_n^{\boldsymbol{x}^{(s)}}}{D_n^{\boldsymbol{x}^{(s)}}}\right.$$
$$\left. - \sum_{s=1}^{d_2} \frac{\partial^2 f(\boldsymbol{x}(t_n), \boldsymbol{y}(t_n))}{\partial \boldsymbol{x}^{(j)} \partial \boldsymbol{y}^{(s)}} \frac{\partial f(\boldsymbol{x}(t_n), \boldsymbol{y}(t_n))}{\partial \boldsymbol{x}^{(j)}} \frac{N_n^{\boldsymbol{y}^{(s)}}}{D_n^{\boldsymbol{y}^{(s)}}}\right],$$

$$P_n^{\boldsymbol{y}^{(i)}}(\boldsymbol{x}(t_n), \boldsymbol{y}(t_n)) = \left(\frac{\rho}{1-\rho} - \frac{(n+1)\rho^{n+1}}{1-\rho^{n+1}}\right) \left[\sum_{s=1}^{d_1} \frac{\partial^2 f(\boldsymbol{x}(t_n), \boldsymbol{y}(t_n))}{\partial \boldsymbol{y}^{(i)} \partial \boldsymbol{x}^{(s)}} \frac{\partial f(\boldsymbol{x}(t_n), \boldsymbol{y}(t_n))}{\partial \boldsymbol{y}^{(i)}} \frac{N_n^{\boldsymbol{x}^{(s)}}}{D_n^{\boldsymbol{x}^{(s)}}}\right.$$
$$\left. - \sum_{s=1}^{d_2} \frac{\partial^2 f(\boldsymbol{x}(t_n), \boldsymbol{y}(t_n))}{\partial \boldsymbol{y}^{(i)} \partial \boldsymbol{y}^{(s)}} \frac{\partial f(\boldsymbol{x}(t_n), \boldsymbol{y}(t_n))}{\partial \boldsymbol{y}^{(i)}} \frac{N_n^{\boldsymbol{y}^{(s)}}}{D_n^{\boldsymbol{y}^{(s)}}}\right].$$

To simplify the notation, we denote $N_n^{\boldsymbol{x}^{(j)}}(\boldsymbol{x}(t_n), \boldsymbol{y}(t_n))$ by $N_n^{\boldsymbol{x}^{(j)}}$ in the following. Similarly, the same convention applies

to $D_n^{\boldsymbol{x}^{(j)}}$, $N_n^{\boldsymbol{y}^{(i)}}$, $D_n^{\boldsymbol{y}^{(i)}}$, $Q_n^{\boldsymbol{x}^{(j)}}$, $P_n^{\boldsymbol{x}^{(j)}}$, $Q_n^{\boldsymbol{y}^{(i)}}$ and $P_n^{\boldsymbol{y}^{(i)}}$.

We proceed to present the following lemma. We will use Lemma B.1 to safely truncate the Taylor-expansion to get an $\mathcal{O}(h^3)$ error.

**Lemma B.1.** *Suppose that $f(\boldsymbol{x},\boldsymbol{y})$ has bounded derivatives up to the fourth order. Then for any bounded $h$, we can get $\dot{\boldsymbol{x}}^{(j)}(t)$, $\ddot{\boldsymbol{x}}^{(j)}(t)$, $\dddot{\boldsymbol{x}}^{(j)}(t)$ and $\dot{\boldsymbol{y}}^{(i)}(t)$, $\ddot{\boldsymbol{y}}^{(i)}(t)$, $\dddot{\boldsymbol{y}}^{(i)}(t)$ are bounded independent of $h$ for all $j = 1, 2, \cdots, d_1$ and $i = 1, 2, \cdots, d_2$.*

*Proof.* The proof follows directly by the definition of $\dot{\boldsymbol{x}}^{(j)}(t)$, $\ddot{\boldsymbol{x}}^{(j)}(t)$, $\dddot{\boldsymbol{x}}^{(j)}(t)$ and $\dot{\boldsymbol{y}}^{(i)}(t)$, $\ddot{\boldsymbol{y}}^{(i)}(t)$, $\dddot{\boldsymbol{y}}^{(i)}(t)$ together with bounded derivatives of $f(x)$ up to the fourth order. $\qquad\square$

**Lemma B.2.** *Let $t_n = nh$ where $h$ is the stepsize of Adam-DA. Then in time region $[t_n, t_{n+1})$, the solutions of the following ODEs:*

$$
\frac{d\boldsymbol{x}^{(j)}(t)}{dt}
$$

$$
= \frac{-\frac{\partial f(\boldsymbol{x},\boldsymbol{y})}{\partial \boldsymbol{x}^{(j)}}}{\sqrt{\left(\frac{\partial f(\boldsymbol{x},\boldsymbol{y})}{\partial \boldsymbol{x}^{(j)}}\right)^2 + \epsilon}} - \frac{h}{2\sqrt{\left(\frac{\partial f(\boldsymbol{x},\boldsymbol{y})}{\partial \boldsymbol{x}^{(j)}}\right)^2 + \epsilon}} \times \left(\frac{\partial}{\partial \boldsymbol{x}^{(j)}}\|\nabla_x f(\boldsymbol{x},\boldsymbol{y})\|_{1,\epsilon} - \frac{\partial}{\partial \boldsymbol{x}^{(j)}}\|\nabla_y f(\boldsymbol{x},\boldsymbol{y})\|_{1,\epsilon}\right)
$$

$$
\times \left[\frac{1+\beta}{1-\beta} - \frac{1+\rho}{1-\rho} + \frac{\epsilon}{\left(\frac{\partial f(\boldsymbol{x},\boldsymbol{y})}{\partial \boldsymbol{x}^{(j)}}\right)^2 + \epsilon}\left(\frac{1+\rho}{1-\rho} - \frac{(n+1)\rho^{n+1}}{1-\rho^{n+1}}\right) - \frac{(n+1)\beta^{n+1}}{1-\beta^{n+1}} + \frac{(n+1)\rho^{n+1}}{1-\rho^{n+1}}\right],
$$

$$
\frac{d\boldsymbol{y}^{(i)}(t)}{dt}
$$

$$
= \frac{\frac{\partial f(\boldsymbol{x},\boldsymbol{y})}{\partial \boldsymbol{y}^{(i)}}}{\sqrt{\left(\frac{\partial f(\boldsymbol{x},\boldsymbol{y})}{\partial \boldsymbol{y}^{(i)}}\right)^2 + \epsilon}} + \frac{h}{2\sqrt{\left(\frac{\partial f(\boldsymbol{x},\boldsymbol{y})}{\partial \boldsymbol{y}^{(i)}}\right)^2 + \epsilon}} \times \left(\frac{\partial}{\partial \boldsymbol{y}^{(i)}}\|\nabla_x f(\boldsymbol{x},\boldsymbol{y})\|_{1,\epsilon} - \frac{\partial}{\partial \boldsymbol{y}^{(i)}}\|\nabla_y f(\boldsymbol{x},\boldsymbol{y})\|_{1,\epsilon}\right)
$$

$$
\times \left[\frac{1+\beta}{1-\beta} - \frac{1+\rho}{1-\rho} + \frac{\epsilon}{\left(\frac{\partial f(\boldsymbol{x},\boldsymbol{y})}{\partial \boldsymbol{y}^{(i)}}\right)^2 + \epsilon}\left(\frac{1+\rho}{1-\rho} - \frac{(n+1)\rho^{n+1}}{1-\rho^{n+1}}\right) - \frac{(n+1)\beta^{n+1}}{1-\beta^{n+1}} + \frac{(n+1)\rho^{n+1}}{1-\rho^{n+1}}\right].
$$

*can approximate the trajectories of Adam-DA with an $\mathcal{O}(h^3)$-local error.*

*Proof.* We rewrite Adam-DA as

$$
\boldsymbol{x}_{n+1}^{(j)} = \boldsymbol{x}_n^{(j)} - h\frac{\sum_{k=0}^n \frac{\beta^{n-k}(1-\beta)}{1-\beta^{n+1}}\frac{\partial f(\boldsymbol{x}_k,\boldsymbol{y}_k)}{\partial \boldsymbol{x}^{(j)}}}{\sqrt{\sum_{k=0}^n \frac{\rho^{n-k}(1-\rho)}{1-\rho^{n+1}}\left(\frac{\partial f(\boldsymbol{x}_k,\boldsymbol{y}_k)}{\partial \boldsymbol{x}^{(j)}}\right)^2 + \epsilon}}, \quad j = 1, \cdots, d_1,
$$

$$
\boldsymbol{y}_{n+1}^{(i)} = \boldsymbol{y}_n^{(i)} + h\frac{\sum_{k=0}^n \frac{\beta^{n-k}(1-\beta)}{1-\beta^{n+1}}\frac{\partial f(\boldsymbol{x}_k,\boldsymbol{y}_k)}{\partial \boldsymbol{y}^{(i)}}}{\sqrt{\sum_{k=0}^n \frac{\rho^{n-k}(1-\rho)}{1-\rho^{n+1}}\left(\frac{\partial f(\boldsymbol{x}_k,\boldsymbol{y}_k)}{\partial \boldsymbol{y}^{(i)}}\right)^2 + \epsilon}}, \quad i = 1, \cdots, d_2.
$$

In each time interval $[t_n, t_{n+1})$, we define

- $\nabla_x G_n(\boldsymbol{x},\boldsymbol{y}) = \left(\frac{\partial G_n(\boldsymbol{x},\boldsymbol{y})}{\partial \boldsymbol{x}^{(1)}}, \cdots, \frac{\partial G_n(\boldsymbol{x},\boldsymbol{y})}{\partial \boldsymbol{x}^{(j)}}, \cdots, \frac{\partial G_n(\boldsymbol{x},\boldsymbol{y})}{\partial \boldsymbol{x}^{(d_1)}}\right)$

- $\nabla_y F_n(\boldsymbol{x},\boldsymbol{y}) = \left(\frac{\partial F_n(\boldsymbol{x},\boldsymbol{y})}{\partial \boldsymbol{y}^{(1)}}, \cdots, \frac{\partial F_n(\boldsymbol{x},\boldsymbol{y})}{\partial \boldsymbol{y}^{(i)}}, \cdots, \frac{\partial F_n(\boldsymbol{x},\boldsymbol{y})}{\partial \boldsymbol{y}^{(d_2)}}\right)$

- $A_n(\boldsymbol{x}, \boldsymbol{y}) = \left( A_n^{(1)}(\boldsymbol{x}, \boldsymbol{y}), \cdots, A_n^{(j)}(\boldsymbol{x}, \boldsymbol{y}), \cdots, A_n^{(d_1)}(\boldsymbol{x}, \boldsymbol{y}) \right)$

- $B_n(\boldsymbol{x}, \boldsymbol{y}) = \left( B_n^{(1)}(\boldsymbol{x}, \boldsymbol{y}), \cdots, B_n^{(i)}(\boldsymbol{x}, \boldsymbol{y}), \cdots, B_n^{(d_2)}(\boldsymbol{x}, \boldsymbol{y}) \right)$

where the terms $G_n(\boldsymbol{x}, \boldsymbol{y})$, $F_n(\boldsymbol{x}, \boldsymbol{y})$, $A_n^{(j)}(\boldsymbol{x}, \boldsymbol{y})$ and $B_n^{(i)}(\boldsymbol{x}, \boldsymbol{y})$ are assumed to satisfy the following equations in the time interval $[t_n, t_{n+1}]$:

$$\frac{d\boldsymbol{x}^{(j)}(t)}{dt} = -\frac{\partial G_n(\boldsymbol{x}, \boldsymbol{y})}{\partial \boldsymbol{x}^{(j)}} + hA_n^{(j)}(\boldsymbol{x}, \boldsymbol{y}), \tag{7}$$

$$\frac{d\boldsymbol{y}^{(i)}(t)}{dt} = \frac{\partial F_n(\boldsymbol{x}, \boldsymbol{y})}{\partial \boldsymbol{y}^{(i)}} + hB_n^{(i)}(\boldsymbol{x}, \boldsymbol{y}). \tag{8}$$

Our goal is to find $G_n$, $F_n$, $A_n$ and $B_n$ such that

$$\boldsymbol{x}^{(j)}(t_{n+1}) - \boldsymbol{x}^{(j)}(t_n) = \frac{-h \sum_{k=0}^{n} \frac{\beta^{n-k}(1-\beta)}{1-\beta^{n+1}} \frac{\partial f(\boldsymbol{x}(t_k), \boldsymbol{y}(t_k))}{\partial \boldsymbol{x}^{(j)}}}{\sqrt{\sum_{k=0}^{n} \frac{\rho^{n-k}(1-\rho)}{1-\rho^{n+1}} \left( \frac{\partial f(\boldsymbol{x}(t_k), \boldsymbol{y}(t_k))}{\partial \boldsymbol{x}^{(j)}} \right)^2 + \epsilon}} + \mathcal{O}(h^3), \tag{9}$$

$$\boldsymbol{y}^{(i)}(t_{n+1}) - \boldsymbol{y}^{(i)}(t_n) = \frac{h \sum_{k=0}^{n} \frac{\beta^{n-k}(1-\beta)}{1-\beta^{n+1}} \frac{\partial f(\boldsymbol{x}(t_k), \boldsymbol{y}(t_k))}{\partial \boldsymbol{y}^{(i)}}}{\sqrt{\sum_{k=0}^{n} \frac{\rho^{n-k}(1-\rho)}{1-\rho^{n+1}} \left( \frac{\partial f(\boldsymbol{x}(t_k), \boldsymbol{y}(t_k))}{\partial \boldsymbol{y}^{(i)}} \right)^2 + \epsilon}} + \mathcal{O}(h^3). \tag{10}$$

We firstly do the Taylor expansions of $\frac{\partial f(\boldsymbol{x}(t_k), \boldsymbol{y}(t_k))}{\partial \boldsymbol{x}^{(j)}}$ as follows:

$$\frac{\partial f(\boldsymbol{x}(t_k), \boldsymbol{y}(t_k))}{\partial \boldsymbol{x}^{(j)}} = \frac{\partial f(\boldsymbol{x}(t_n), \boldsymbol{y}(t_n))}{\partial \boldsymbol{x}^{(j)}} + \sum_{s=1}^{d_1} \frac{\partial^2 f(\boldsymbol{x}(t_n), \boldsymbol{y}(t_n))}{\partial \boldsymbol{x}^{(j)} \partial \boldsymbol{x}^{(s)}} \left( \boldsymbol{x}^{(s)}(t_k) - \boldsymbol{x}^{(s)}(t_n) \right)$$

$$+ \sum_{s=1}^{d_2} \frac{\partial^2 f(\boldsymbol{x}(t_n), \boldsymbol{y}(t_n))}{\partial \boldsymbol{x}^{(j)} \partial \boldsymbol{y}^{(s)}} \left( \boldsymbol{y}^{(s)}(t_k) - \boldsymbol{y}^{(s)}(t_n) \right) + \mathcal{O}\left( h^2 \right)$$

$$= \frac{\partial f(\boldsymbol{x}(t_n), \boldsymbol{y}(t_n))}{\partial \boldsymbol{x}^{(j)}} - \sum_{s=1}^{d_1} \frac{\partial^2 f(\boldsymbol{x}(t_n), \boldsymbol{y}(t_n))}{\partial \boldsymbol{x}^{(j)} \partial \boldsymbol{x}^{(s)}} \dot{\boldsymbol{x}}^{(s)}(t_n^+) (n-k) h$$

$$- \sum_{s=1}^{d_2} \frac{\partial^2 f(\boldsymbol{x}(t_n), \boldsymbol{y}(t_n))}{\partial \boldsymbol{x}^{(j)} \partial \boldsymbol{y}^{(s)}} \dot{\boldsymbol{y}}^{(s)}(t_n^+) (n-k) h + \mathcal{O}\left( h^2 \right), \tag{11}$$

$$= \frac{\partial f(\boldsymbol{x}(t_n), \boldsymbol{y}(t_n))}{\partial \boldsymbol{x}^{(j)}} + \sum_{s=1}^{d_1} \frac{\partial^2 f(\boldsymbol{x}(t_n), \boldsymbol{y}(t_n))}{\partial \boldsymbol{x}^{(j)} \partial \boldsymbol{x}^{(s)}} \frac{\partial G_n(\boldsymbol{x}(t_n), \boldsymbol{y}(t_n))}{\partial \boldsymbol{x}^{(s)}} (n-k) h$$

$$- \sum_{s=1}^{d_2} \frac{\partial^2 f(\boldsymbol{x}(t_n), \boldsymbol{y}(t_n))}{\partial \boldsymbol{x}^{(j)} \partial \boldsymbol{y}^{(s)}} \frac{\partial F_n(\boldsymbol{x}(t_n), \boldsymbol{y}(t_n))}{\partial \boldsymbol{y}^{(s)}} (n-k) h + \mathcal{O}\left( h^2 \right),$$

then we obtain that

$$\sum_{k=0}^{n} \frac{\beta^{n-k}(1-\beta)}{1-\beta^{n+1}} \frac{\partial f(\boldsymbol{x}(t_k), \boldsymbol{y}(t_k))}{\partial \boldsymbol{x}^{(j)}} = \sum_{k=0}^{n} \frac{\beta^{n-k}(1-\beta)}{1-\beta^{n+1}} \frac{\partial f(\boldsymbol{x}(t_n), \boldsymbol{y}(t_n))}{\partial \boldsymbol{x}^{(j)}}$$

$$+ h \sum_{k=0}^{n} \frac{\beta^{n-k}(1-\beta)(n-k)}{1-\beta^{n+1}} \sum_{s=1}^{d_1} \frac{\partial^2 f(\boldsymbol{x}(t_n), \boldsymbol{y}(t_n))}{\partial \boldsymbol{x}^{(j)} \partial \boldsymbol{x}^{(s)}} \frac{\partial G_n(\boldsymbol{x}(t_n), \boldsymbol{y}(t_n))}{\partial \boldsymbol{x}^{(s)}} \tag{12}$$

$$- h \sum_{k=0}^{n} \frac{\beta^{n-k}(1-\beta)(n-k)}{1-\beta^{n+1}} \sum_{s=1}^{d_2} \frac{\partial^2 f(\boldsymbol{x}(t_n), \boldsymbol{y}(t_n))}{\partial \boldsymbol{x}^{(j)} \partial \boldsymbol{y}^{(s)}} \frac{\partial F_n(\boldsymbol{x}(t_n), \boldsymbol{y}(t_n))}{\partial \boldsymbol{y}^{(s)}} + \mathcal{O}\left(h^2\right),$$

Executing square operation in the both sides of (11), we can obtain

$$\left(\frac{\partial f(\boldsymbol{x}(t_k), \boldsymbol{y}(t_k))}{\partial \boldsymbol{x}^{(j)}}\right)^2 = \left(\frac{\partial f(\boldsymbol{x}(t_n), \boldsymbol{y}(t_n))}{\partial \boldsymbol{x}^{(j)}}\right)^2$$

$$+ 2h \sum_{s=1}^{d_1} (n-k) \frac{\partial^2 f(\boldsymbol{x}(t_n), \boldsymbol{y}(t_n))}{\partial \boldsymbol{x}^{(j)} \partial \boldsymbol{x}^{(s)}} \frac{\partial f(\boldsymbol{x}(t_n), \boldsymbol{y}(t_n))}{\partial \boldsymbol{x}^{(j)}} \frac{\partial G_n(\boldsymbol{x}(t_n), \boldsymbol{y}(t_n))}{\partial \boldsymbol{x}^{(s)}} \tag{13}$$

$$- 2h \sum_{s=1}^{d_2} (n-k) \frac{\partial^2 f(\boldsymbol{x}(t_n), \boldsymbol{y}(t_n))}{\partial \boldsymbol{x}^{(j)} \partial \boldsymbol{y}^{(s)}} \frac{\partial f(\boldsymbol{x}(t_n), \boldsymbol{y}(t_n))}{\partial \boldsymbol{x}^{(j)}} \frac{\partial F_n(\boldsymbol{x}(t_n), \boldsymbol{y}(t_n))}{\partial \boldsymbol{y}^{(s)}} + \mathcal{O}\left(h^2\right).$$

Then we can get

$$\sum_{k=0}^{n} \frac{\rho^{n-k}(1-\rho)}{1-\rho^{n+1}} \left(\frac{\partial f(\boldsymbol{x}(t_k), \boldsymbol{y}(t_k))}{\partial \boldsymbol{x}^{(j)}}\right)^2$$

$$= \sum_{k=0}^{n} \frac{\rho^{n-k}(1-\rho)}{1-\rho^{n+1}} \left(\frac{\partial f(\boldsymbol{x}(t_n), \boldsymbol{y}(t_n))}{\partial \boldsymbol{x}^{(j)}}\right)^2$$

$$+ 2h \sum_{k=0}^{n} \frac{\rho^{n-k}(1-\rho)(n-k)}{1-\rho^{n+1}} \sum_{s=1}^{d_1} \frac{\partial^2 f(\boldsymbol{x}(t_n), \boldsymbol{y}(t_n))}{\partial \boldsymbol{x}^{(j)} \partial \boldsymbol{x}^{(s)}} \frac{\partial f(\boldsymbol{x}(t_n), \boldsymbol{y}(t_n))}{\partial \boldsymbol{x}^{(j)}} \frac{\partial G_n(\boldsymbol{x}(t_n), \boldsymbol{y}(t_n))}{\partial \boldsymbol{x}^{(s)}} \tag{14}$$

$$- 2h \sum_{k=0}^{n} \frac{\rho^{n-k}(1-\rho)(n-k)}{1-\rho^{n+1}} \sum_{s=1}^{d_2} \frac{\partial^2 f(\boldsymbol{x}(t_n), \boldsymbol{y}(t_n))}{\partial \boldsymbol{x}^{(j)} \partial \boldsymbol{y}^{(s)}} \frac{\partial f(\boldsymbol{x}(t_n), \boldsymbol{y}(t_n))}{\partial \boldsymbol{x}^{(j)}} \frac{\partial F_n(\boldsymbol{x}(t_n), \boldsymbol{y}(t_n))}{\partial \boldsymbol{y}^{(s)}}$$

$$+ \mathcal{O}\left(h^2\right).$$

Then we use the fact that $\left(\sum_{k=0}^{n} a_k h^k\right)^{-\frac{1}{2}} = \frac{1}{\sqrt{a_0}} - \frac{a_1}{2(\sqrt{a_0})^3} h + O(h^2)$ to get

$$\frac{1}{\sqrt{\sum_{k=0}^{n} \frac{\rho^{n-k}(1-\rho)}{1-\rho^{n+1}} \left(\frac{\partial f(\boldsymbol{x}(t_k), \boldsymbol{y}(t_k))}{\partial \boldsymbol{x}^{(j)}}\right)^2 + \epsilon}} = \frac{1}{D_n^{\boldsymbol{x}^{(j)}}} + \frac{h S_n^{\boldsymbol{x}^{(j)}}}{\left(D_n^{\boldsymbol{x}^{(j)}}\right)^3} + \mathcal{O}\left(h^2\right), \tag{15}$$

where

$$S_n^{\boldsymbol{x}^{(j)}} = \sum_{k=0}^{n} \frac{\rho^{n-k}(1-\rho)(n-k)}{1-\rho^{n+1}} \left( \sum_{s=1}^{d_1} \frac{\partial^2 f(\boldsymbol{x}(t_n), \boldsymbol{y}(t_n))}{\partial \boldsymbol{x}^{(j)} \partial \boldsymbol{x}^{(s)}} \frac{\partial f(\boldsymbol{x}(t_n), \boldsymbol{y}(t_n))}{\partial \boldsymbol{x}^{(j)}} \frac{\partial G_n(\boldsymbol{x}(t_n), \boldsymbol{y}(t_n))}{\partial \boldsymbol{x}^{(s)}} \right.$$

$$\left. - \sum_{s=1}^{d_2} \frac{\partial^2 f(\boldsymbol{x}(t_k), \boldsymbol{y}(t_k))}{\partial \boldsymbol{x}^{(j)} \partial \boldsymbol{y}^{(s)}} \frac{\partial f(\boldsymbol{x}(t_k), \boldsymbol{y}(t_k))}{\partial \boldsymbol{x}^{(j)}} \frac{\partial F_n(\boldsymbol{x}(t_n), \boldsymbol{y}(t_n))}{\partial \boldsymbol{y}^{(s)}} \right),$$

and we have used the equation

$$\sum_{k=0}^{n} \frac{\rho^{n-k}(1-\rho)}{1-\rho^{n+1}} \left( \frac{\partial f(\boldsymbol{x}(t_n), \boldsymbol{y}(t_n))}{\partial \boldsymbol{x}^{(j)}} \right)^2 = \left( \frac{\partial f(\boldsymbol{x}(t_n), \boldsymbol{y}(t_n))}{\partial \boldsymbol{x}^{(j)}} \right)^2.$$

Firstly, we solve $\frac{\partial G_n(\boldsymbol{x}, \boldsymbol{y})}{\partial \boldsymbol{x}^{(j)}}$ and $\frac{\partial F_n(\boldsymbol{x}, \boldsymbol{y})}{\partial \boldsymbol{y}^{(i)}}$. Substituting (12) and (15) into (9), we can get

$$\boldsymbol{x}^{(j)}(t_{n+1}) - \boldsymbol{x}^{(j)}(t_n) = -h \frac{N_n^{\boldsymbol{x}^{(j)}}}{D_n^{\boldsymbol{x}^{(j)}}} + \mathcal{O}(h^2) \tag{16}$$

Since $\dot{\boldsymbol{x}}^{(j)}(t_n^+)$, $\ddot{\boldsymbol{x}}^{(j)}(t_n^+)$ and $\dddot{\boldsymbol{x}}^{(j)}(t_n^+)$ are bounded independent of $h$ (by Lemma B.1), we can obtain the Taylor expansion at time $t_n$ (recall that $t_n = nh$)

$$\boldsymbol{x}^{(j)}(t_{n+1}) - \boldsymbol{x}^{(j)}(t_n) = \boldsymbol{x}^{(j)}(t_n + h) - \boldsymbol{x}^{(j)}(t_n)$$

$$= h\dot{\boldsymbol{x}}^{(j)}(t_n^+) + \frac{h^2}{2}\ddot{\boldsymbol{x}}^{(j)}(t_n^+) + \mathcal{O}(h^3), \tag{17}$$

where $\dot{\boldsymbol{x}}^{(j)}(t_n^+)$ (resp. $\ddot{\boldsymbol{x}}^{(j)}(t_n^+)$) is the right first-order derivative (resp. the right second-order derivative) at time $t_n$.

Applying the chain rule for (7), we have

$$\ddot{\boldsymbol{x}}^{(j)}(t_n^+)$$

$$= -\sum_{s=1}^{d_1} \frac{\partial^2 G_n(\boldsymbol{x}(t_n), \boldsymbol{y}(t_n))}{\partial \boldsymbol{x}^{(s)} \partial \boldsymbol{x}^{(j)}} \dot{\boldsymbol{x}}^{(s)}(t_n^+) - \sum_{s=1}^{d_2} \frac{\partial^2 G_n(\boldsymbol{x}(t_n), \boldsymbol{y}(t_n))}{\partial \boldsymbol{x}^{(j)} \partial \boldsymbol{y}^{(s)}} \dot{\boldsymbol{y}}^{(s)}(t_n^+) + \mathcal{O}(h)$$

$$= \sum_{s=1}^{d_1} \frac{\partial^2 G_n(\boldsymbol{x}(t_n), \boldsymbol{y}(t_n))}{\partial \boldsymbol{x}^{(s)} \partial \boldsymbol{x}^{(j)}} \frac{\partial G_n(\boldsymbol{x}(t_n), \boldsymbol{y}(t_n))}{\partial \boldsymbol{x}^{(s)}} \tag{18}$$

$$- \sum_{s=1}^{d_2} \frac{\partial^2 G_n(\boldsymbol{x}(t_n), \boldsymbol{y}(t_n))}{\partial \boldsymbol{x}^{(j)} \partial \boldsymbol{y}^{(s)}} \frac{\partial F_n(\boldsymbol{x}(t_n), \boldsymbol{y}(t_n))}{\partial \boldsymbol{y}^{(s)}} + \mathcal{O}(h).$$

Also note that from (7), we have

$$\dot{\boldsymbol{x}}^{(j)}(t_n^+) = -\frac{\partial G_n(\boldsymbol{x}(t_n), \boldsymbol{y}(t_n))}{\partial \boldsymbol{x}^{(j)}} + hA_n^{(j)}(\boldsymbol{x}(t_n), \boldsymbol{y}(t_n)), \tag{19}$$

Substituting (18) and (19) into (17), we can get

$$\boldsymbol{x}^{(j)}(t_{n+1}) - \boldsymbol{x}^{(j)}(t_n) = -h\frac{\partial G_n(\boldsymbol{x}(t_n), \boldsymbol{y}(t_n))}{\partial \boldsymbol{x}^{(j)}} + h^2 A_n^{(j)}(\boldsymbol{x}(t_n), \boldsymbol{y}(t_n))$$

$$+ \frac{h^2}{2}\sum_{s=1}^{d_1}\frac{\partial^2 G_n(\boldsymbol{x}(t_n), \boldsymbol{y}(t_n))}{\partial \boldsymbol{x}^{(s)}\partial \boldsymbol{x}^{(j)}}\frac{\partial G_n(\boldsymbol{x}(t_n), \boldsymbol{y}(t_n))}{\partial \boldsymbol{x}^{(s)}} \tag{20}$$

$$- \frac{h^2}{2}\sum_{s=1}^{d_2}\frac{\partial^2 G_n(\boldsymbol{x}(t_n), \boldsymbol{y}(t_n))}{\partial \boldsymbol{x}^{(j)}\partial \boldsymbol{y}^{(s)}}\frac{\partial F_n(\boldsymbol{x}(t_n), \boldsymbol{y}(t_n))}{\partial \boldsymbol{y}^{(s)}} + \mathcal{O}(h^3).$$

Comparing the coefficients of term $h$ in (16) and (20), we can get

$$\frac{\partial G_n(\boldsymbol{x}(t_n), \boldsymbol{y}(t_n))}{\partial \boldsymbol{x}^{(j)}} = \frac{N_n^{\boldsymbol{x}^{(j)}}}{D_n^{\boldsymbol{x}^{(j)}}}. \tag{21}$$

Repeating the similar argument for $y$-player's equation, we can get

$$\frac{\partial F_n(\boldsymbol{x}(t_n), \boldsymbol{y}(t_n))}{\partial \boldsymbol{y}^{(i)}} = \frac{N_n^{\boldsymbol{y}^{(i)}}}{D_n^{\boldsymbol{y}^{(i)}}}. \tag{22}$$

With (21) and (22) in hand, we can substitute (12) and (15) into (9) to get

$$\boldsymbol{x}^{(j)}(t_{n+1}) - \boldsymbol{x}^{(j)}(t_n) = -h\frac{N_n^{\boldsymbol{x}^{(j)}}}{D_n^{\boldsymbol{x}^{(j)}}} - h^2\left(\frac{Q_n^{\boldsymbol{x}^{(j)}}}{D_n^{\boldsymbol{x}^{(j)}}} - \frac{P_n^{\boldsymbol{x}^{(j)}}N_n^{\boldsymbol{x}^{(j)}}}{\left(D_n^{\boldsymbol{x}^{(j)}}\right)^3}\right) + \mathcal{O}(h^3) \tag{23}$$

Comparing the coefficients of term $h^2$ in (23) and (20), we can get

$$A_n^{(j)}(\boldsymbol{x}(t_n), \boldsymbol{y}(t_n))$$

$$= \frac{1}{2}\sum_{s=1}^{d_2}\frac{\partial^2 G_n(\boldsymbol{x}(t_n), \boldsymbol{y}(t_n))}{\partial \boldsymbol{x}^{(j)}\partial \boldsymbol{y}^{(s)}}\frac{\partial F_n(\boldsymbol{x}(t_n), \boldsymbol{y}(t_n))}{\partial \boldsymbol{y}^{(s)}}$$

$$- \frac{1}{2}\sum_{s=1}^{d_1}\frac{\partial^2 G_n(\boldsymbol{x}(t_n), \boldsymbol{y}(t_n))}{\partial \boldsymbol{x}^{(s)}\partial \boldsymbol{x}^{(j)}}\frac{\partial G_n(\boldsymbol{x}(t_n), \boldsymbol{y}(t_n))}{\partial \boldsymbol{x}^{(s)}} - \left(\frac{Q_n^{\boldsymbol{x}^{(j)}}}{D_n^{\boldsymbol{x}^{(j)}}} - \frac{P_n^{\boldsymbol{x}^{(j)}}N_n^{\boldsymbol{x}^{(j)}}}{\left(D_n^{\boldsymbol{x}^{(j)}}\right)^3}\right) \tag{24}$$

$$= -\left(\sum_{s=1}^{d_1}\frac{\frac{\partial N_n^{\boldsymbol{x}^{(j)}}}{\partial \boldsymbol{x}^{(s)}}D_n^{\boldsymbol{x}^{(j)}} - \frac{\partial D_n^{\boldsymbol{x}^{(j)}}}{\partial \boldsymbol{x}^{(s)}}N_n^{\boldsymbol{x}^{(j)}}}{2\left(D_n^{\boldsymbol{x}^{(j)}}\right)^2}\frac{N_n^{\boldsymbol{x}^{(s)}}}{D_n^{\boldsymbol{x}^{(s)}}} - \sum_{s=1}^{d_2}\frac{\frac{\partial N_n^{\boldsymbol{x}^{(j)}}}{\partial \boldsymbol{y}^{(s)}}D_n^{\boldsymbol{x}^{(j)}} - \frac{\partial D_n^{\boldsymbol{x}^{(j)}}}{\partial \boldsymbol{y}^{(s)}}N_n^{\boldsymbol{x}^{(j)}}}{2\left(D_n^{\boldsymbol{x}^{(j)}}\right)^2}\frac{N_n^{\boldsymbol{y}^{(s)}}}{D_n^{\boldsymbol{y}^{(s)}}}\right)$$

$$- \left(\frac{Q_n^{\boldsymbol{x}^{(j)}}}{D_n^{\boldsymbol{x}^{(j)}}} - \frac{P_n^{\boldsymbol{x}^{(j)}}N_n^{\boldsymbol{x}^{(j)}}}{\left(D_n^{\boldsymbol{x}^{(j)}}\right)^3}\right).$$

Substituting $D_n^{\boldsymbol{x}^{(j)}}$, $N_n^{\boldsymbol{x}^{(j)}}$, $P_n^{\boldsymbol{x}^{(j)}}$ and $Q_n^{\boldsymbol{x}^{(j)}}$ in (6) into (24), we can get the desired equation.

Repeating the above similar argument for $y$-player's equation, we can get

$$\frac{\partial F_n(\boldsymbol{x}, \boldsymbol{y})}{\partial \boldsymbol{y}^{(i)}} = \frac{N_n^{\boldsymbol{y}^{(i)}}}{D_n^{\boldsymbol{y}^{(i)}}}, \tag{25}$$

$$B_n^{(i)}(\boldsymbol{x}(t_n), \boldsymbol{y}(t_n)) = \sum_{s=1}^{d_1} \frac{\frac{\partial N_n^{\boldsymbol{y}^{(i)}}}{\partial \boldsymbol{x}^{(s)}} D_n^{\boldsymbol{y}^{(i)}} - \frac{\partial D_n^{\boldsymbol{y}^{(i)}}}{\partial \boldsymbol{x}^{(s)}} N_n^{\boldsymbol{y}^{(i)}}}{2 \left(D_n^{\boldsymbol{y}^{(i)}}\right)^2} \frac{N_n^{\boldsymbol{x}^{(s)}}}{D_n^{\boldsymbol{x}^{(s)}}}$$

$$- \sum_{s=1}^{d_2} \frac{\frac{\partial N_n^{\boldsymbol{y}^{(i)}}}{\partial \boldsymbol{y}^{(s)}} D_n^{\boldsymbol{y}^{(i)}} - \frac{\partial D_n^{\boldsymbol{y}^{(i)}}}{\partial \boldsymbol{y}^{(s)}} N_n^{\boldsymbol{y}^{(i)}}}{2 \left(D_n^{\boldsymbol{y}^{(i)}}\right)^2} \frac{N_n^{\boldsymbol{y}^{(s)}}}{D_n^{\boldsymbol{y}^{(s)}}} + \left( \frac{Q_n^{\boldsymbol{y}^{(i)}}}{D_n^{\boldsymbol{y}^{(i)}}} - \frac{P_n^{\boldsymbol{y}^{(i)}} N_n^{\boldsymbol{y}^{(i)}}}{\left(D_n^{\boldsymbol{y}^{(i)}}\right)^3} \right). \tag{26}$$

Substituting $D_n^{\boldsymbol{y}^{(i)}}$, $N_n^{\boldsymbol{y}^{(i)}}$, $P_n^{\boldsymbol{y}^{(i)}}$ and $Q_n^{\boldsymbol{y}^{(i)}}$ in (6) into (25) and (26), we can get the desired equation. $\qquad\square$

**Corollary B.3.** *When $n$ tends to infinity, the ODEs become*

$$\frac{d\boldsymbol{x}^{(j)}(t)}{dt} = -\frac{\frac{\partial f(\boldsymbol{x}, \boldsymbol{y})}{\partial \boldsymbol{x}^{(j)}}}{\sqrt{\left(\frac{\partial f(\boldsymbol{x}, \boldsymbol{y})}{\partial \boldsymbol{x}^{(j)}}\right)^2 + \epsilon}} - \frac{h}{2\sqrt{\left(\frac{\partial f(\boldsymbol{x}, \boldsymbol{y})}{\partial \boldsymbol{x}^{(j)}}\right)^2 + \epsilon}} \times$$

$$\left( \frac{1+\beta}{1-\beta} - \frac{1+\rho}{1-\rho} + \frac{\epsilon}{\left(\frac{\partial f(\boldsymbol{x}, \boldsymbol{y})}{\partial \boldsymbol{x}^{(j)}}\right)^2 + \epsilon} \frac{1+\rho}{1-\rho} \right) \left( \frac{\partial}{\partial \boldsymbol{x}^{(j)}} \|\nabla_x f(\boldsymbol{x}, \boldsymbol{y})\|_{1,\epsilon} - \frac{\partial}{\partial \boldsymbol{x}^{(j)}} \|\nabla_y f(\boldsymbol{x}, \boldsymbol{y})\|_{1,\epsilon} \right),$$

$$\frac{d\boldsymbol{y}^{(i)}(t)}{dt} = \frac{\frac{\partial f(\boldsymbol{x}, \boldsymbol{y})}{\partial \boldsymbol{y}^{(i)}}}{\sqrt{\left(\frac{\partial f(\boldsymbol{x}, \boldsymbol{y})}{\partial \boldsymbol{y}^{(i)}}\right)^2 + \epsilon}} + \frac{h}{2\sqrt{\left(\frac{\partial f(\boldsymbol{x}, \boldsymbol{y})}{\partial \boldsymbol{y}^{(i)}}\right)^2 + \epsilon}} \times$$

$$\left( \frac{1+\beta}{1-\beta} - \frac{1+\rho}{1-\rho} + \frac{\epsilon}{\left(\frac{\partial f(\boldsymbol{x}, \boldsymbol{y})}{\partial \boldsymbol{y}^{(i)}}\right)^2 + \epsilon} \frac{1+\rho}{1-\rho} \right) \left( \frac{\partial}{\partial \boldsymbol{y}^{(i)}} \|\nabla_x f(\boldsymbol{x}, \boldsymbol{y})\|_{1,\epsilon} - \frac{\partial}{\partial \boldsymbol{y}^{(i)}} \|\nabla_y f(\boldsymbol{x}, \boldsymbol{y})\|_{1,\epsilon} \right).$$

**Remark.** Recall that the derivation of ODEs in Appendix B.2 is written by coordinate-wise and we transform it into matrix form in Section 3.

**Lemma B.4.** *Suppose that $f(\boldsymbol{x})$ has bounded derivatives up to the fourth order. Let $T_0$ be a fixed time interval. For all $t \in [0, T_0]$. Let $(\tilde{\boldsymbol{x}}(t), \tilde{\boldsymbol{y}}(t))$ be the solution trajectory of ODEs defined in Lemma B.2 and $(\boldsymbol{x}(t), \boldsymbol{y}(t))$ be the solution trajectory of ODEs for Continuous Adam-DA defined in Corollary B.3. Suppose that at time $t_n = nh$, we have*

$$(\tilde{\boldsymbol{x}}(t), \tilde{\boldsymbol{y}}(t)) = (\boldsymbol{x}(t), \boldsymbol{y}(t)).$$

*If we select $n > \max\{\frac{2\log h}{\log|\beta|}, \frac{2\log h}{\log \rho}\}$, then at time $t_{n+1} = (n+1)h$, we have*

$$\| (\tilde{\boldsymbol{x}}(t_{n+1}), \tilde{\boldsymbol{y}}(t_{n+1})) - (\boldsymbol{x}(t_{n+1}), \boldsymbol{y}(t_{n+1})) \| = \mathcal{O}(h^3).$$

*Proof.* We only need to prove

$$|\tilde{\boldsymbol{x}}^{(j)}(t_{n+1}) - \boldsymbol{x}^{(j)}(t_{n+1})| = \mathcal{O}(h^3), \quad j = 1, 2, \cdots, d_1,$$
$$|\tilde{\boldsymbol{y}}^{(i)}(t_{n+1}) - \boldsymbol{y}^{(i)}(t_{n+1})| = \mathcal{O}(h^3), \quad i = 1, 2, \cdots, d_2.$$

Firstly, we prove $|\tilde{\boldsymbol{x}}^{(j)}(t_{n+1}) - \boldsymbol{x}^{(j)}(t_{n+1})| = \mathcal{O}(h^3)$. By the Taylor expansion, we have

$$\tilde{\boldsymbol{x}}^{(j)}(t_{n+1}) = \tilde{\boldsymbol{x}}^{(j)}(nh+h) = \tilde{\boldsymbol{x}}^{(j)}(nh) + h\dot{\tilde{\boldsymbol{x}}}^{(j)}(nh^+) + \frac{h^2}{2}\ddot{\tilde{\boldsymbol{x}}}^{(j)}(nh^+) + \mathcal{O}(h^3),$$

$$\boldsymbol{x}^{(j)}(t_{n+1}) = \boldsymbol{x}^{(j)}(nh+h) = \boldsymbol{x}^{(j)}(nh) + h\dot{\boldsymbol{x}}^{(j)}(nh) + \frac{h^2}{2}\ddot{\boldsymbol{x}}^{(j)}(nh) + \mathcal{O}(h^3),$$

Recall that $(\boldsymbol{x}(t_n), \boldsymbol{y}(t_n)) = (\tilde{\boldsymbol{x}}(t_n), \tilde{\boldsymbol{y}}(t_n))$, then we can get

$$|\tilde{\boldsymbol{x}}^{(j)}(t_{n+1}) - \boldsymbol{x}^{(j)}(t_{n+1})| = h|\dot{\tilde{\boldsymbol{x}}}^{(j)}(nh^+) - \dot{\boldsymbol{x}}^{(j)}(nh)| + \frac{h^2}{2}|\ddot{\tilde{\boldsymbol{x}}}^{(j)}(nh^+) - \ddot{\boldsymbol{x}}^{(j)}(nh)| + \mathcal{O}(h^3).$$

By Lemma B.2 and Corollary B.3, we have

$$|h\dot{\tilde{\boldsymbol{x}}}^{(j)}(nh^+) - h\dot{\boldsymbol{x}}^{(j)}(nh)|$$

$$= h^2 \left| \frac{\epsilon}{\left(\frac{\partial f(\boldsymbol{x}(t),\boldsymbol{y}(t))}{\partial \boldsymbol{y}^{(i)}}\right)^2 + \epsilon} \left(-\frac{(n+1)\rho^{n+1}}{1-\rho^{n+1}}\right) - \frac{(n+1)\beta^{n+1}}{1-\beta^{n+1}} + \frac{(n+1)\rho^{n+1}}{1-\rho^{n+1}} \right|$$

$$\times \left| \frac{\partial}{\partial \boldsymbol{x}^{(j)}} \|\nabla_x f(x(t),y(t))\|_{1,\epsilon} - \frac{\partial}{\partial \boldsymbol{x}^{(j)}} \|\nabla_y f(x(t),y(t))\|_{1,\epsilon} \right|$$

$$\leq h^2 \left( \left| \frac{\epsilon}{\left(\frac{\partial f(x(t),y(t))}{\partial \boldsymbol{y}^{(i)}}\right)^2 + \epsilon} \left(-\frac{(n+1)\rho^{n+1}}{1-\rho^{n+1}}\right) \right| + \left| -\frac{(n+1)\beta^{n+1}}{1-\beta^{n+1}} \right| + \left| \frac{(n+1)\rho^{n+1}}{1-\rho^{n+1}} \right| \right)$$

$$\times \left| \frac{\partial}{\partial \boldsymbol{x}^{(j)}} \|\nabla_x f(x(t),y(t))\|_{1,\epsilon} - \frac{\partial}{\partial \boldsymbol{x}^{(j)}} \|\nabla_y f(x(t),y(t))\|_{1,\epsilon} \right|$$

$$\leq h^2 \left( \left| \frac{(n+1)\rho^{n+1}}{1-\rho} \right| + \left| \frac{(n+1)\beta^{n+1}}{1-\beta} \right| + \left| \frac{(n+1)\rho^{n+1}}{1-\rho} \right| \right)$$

$$\times \left| \frac{\partial}{\partial \boldsymbol{x}^{(j)}} \|\nabla_x f(x(t),y(t))\|_{1,\epsilon} - \frac{\partial}{\partial \boldsymbol{x}^{(j)}} \|\nabla_y f(x(t),y(t))\|_{1,\epsilon} \right|$$

Actually, the term

$$\left| \frac{\partial}{\partial \boldsymbol{x}^{(j)}} \|\nabla_x f(\boldsymbol{x}(t),\boldsymbol{y}(t))\|_{1,\epsilon} - \frac{\partial}{\partial \boldsymbol{x}^{(j)}} \|\nabla_y f(\boldsymbol{x}(t),\boldsymbol{y}(t))\|_{1,\epsilon} \right|$$

is bounded since the first and second derivatives of $f$ are bounded. It suffices to prove

$$\left| \frac{(n+1)\rho^{n+1}}{1-\rho} \right| + \left| \frac{(n+1)\beta^{n+1}}{1-\beta} \right| + \left| \frac{(n+1)\rho^{n+1}}{1-\rho} \right| = \mathcal{O}(h).$$

If we select $n > \max\{\frac{2\log h}{\log|\beta|}, \frac{2\log h}{\log\rho}\}$, i.e., $|\beta|^n < h^2$ and $|\rho|^n < h^2$, then we have

$$|(n+1)\beta^{n+1}| < \frac{T_0}{h}|\beta|^{n+1} < T_0|\beta|h, \quad |(n+1)\rho^{n+1}| < \frac{T_0}{h}|\rho|^{n+1} < T_0|\rho|h.$$

Repeating the similar argument, we can prove

$$h^2|\ddot{\tilde{\boldsymbol{x}}}^{(j)}(nh^+) - \ddot{\boldsymbol{x}}^{(j)}(nh)| = \mathcal{O}(h^3)$$

if $n > \max\{\frac{2\log h}{\log|\beta|}, \frac{2\log h}{\log\rho}\}$. We can also prove $|\tilde{\boldsymbol{y}}^{(i)}(t_{n+1}) - \boldsymbol{y}^{(i)}(t_{n+1})| = \mathcal{O}(h^3)$ through similarly arguments. □

The proof of Theorem 3.1 follows from Lemma B.2, Corollary B.3 and Lemma B.4.

## C. Additional Materials for Section 4.

### C.1. Proof of Proposition 4.1.

**Lemma C.1.** *We have*

- $\nabla_x \|\nabla_x f(\boldsymbol{x}, \boldsymbol{y})\|_{1,\epsilon} = \nabla_x^2 f(\boldsymbol{x}, \boldsymbol{y}) \cdot \mu_\epsilon(\boldsymbol{x}, \boldsymbol{y}) \cdot \nabla_x f(\boldsymbol{x}, \boldsymbol{y})$

- $\nabla_y \|\nabla_x f(\boldsymbol{x}, \boldsymbol{y})\|_{1,\epsilon} = \nabla_{yx} f(\boldsymbol{x}, \boldsymbol{y}) \cdot \mu_\epsilon(x, y) \cdot \nabla_x f(\boldsymbol{x}, \boldsymbol{y})$

- $\nabla_y \|\nabla_y f(\boldsymbol{x}, \boldsymbol{y})\|_{1,\epsilon} = \nabla_y^2 f(\boldsymbol{x}, \boldsymbol{y}) \cdot \nu_\epsilon(x, y) \cdot \nabla_y f(\boldsymbol{x}, \boldsymbol{y})$

- $\nabla_x \|\nabla_y f(\boldsymbol{x}, \boldsymbol{y})\|_{1,\epsilon} = \nabla_{xy} f(\boldsymbol{x}, \boldsymbol{y}) \cdot \nu_\epsilon(x, y) \cdot \nabla_y f(\boldsymbol{x}, \boldsymbol{y})$

*Proof.* Here we only calculate $\nabla_x \|\nabla_x f(\boldsymbol{x}, \boldsymbol{y})\|_{1,\epsilon}$, other items are similar.

Let $g(\boldsymbol{x}, \boldsymbol{y}) = \nabla_x f(\boldsymbol{x}, \boldsymbol{y})$, then we have

$$\nabla_x \|\nabla_x f(\boldsymbol{x}, \boldsymbol{y})\|_{1,\epsilon} = \nabla_x \|g(\boldsymbol{x}, \boldsymbol{y})\|_{1,\epsilon} = \nabla_x g(\boldsymbol{x}, \boldsymbol{y}) \cdot \nabla_g \|g(\boldsymbol{x}, \boldsymbol{y})\|_{1,\epsilon} = \nabla_x^2 f(\boldsymbol{x}, \boldsymbol{y}) \cdot \nabla_g \|g(\boldsymbol{x}, \boldsymbol{y})\|_{1,\epsilon}$$

For the term $\nabla_g \|g(\boldsymbol{x}, \boldsymbol{y})\|_{1,\epsilon}$, we have

$$\nabla_g \|g(\boldsymbol{x}, \boldsymbol{y})\|_{1,\epsilon} = \nabla_g \left( \sum_{i=1}^{n} \sqrt{g_i^2 + \epsilon} \right) = \left( \frac{g_i}{\sqrt{g_i^2 + \epsilon}} \right)_i = \mu_\epsilon(\boldsymbol{x}, \boldsymbol{y}) \cdot \nabla_x f(\boldsymbol{x}, \boldsymbol{y})$$

Thus combine above, we get

$$\nabla_x \|\nabla_x f(\boldsymbol{x}, \boldsymbol{y})\|_{1,\epsilon} = \nabla_x^2 f(\boldsymbol{x}, \boldsymbol{y}) \cdot \mu_\epsilon(\boldsymbol{x}, \boldsymbol{y}) \cdot \nabla_x f(\boldsymbol{x}, \boldsymbol{y})$$

$\square$

**Lemma C.2.** *For Continuous Adam-DA, let $(\boldsymbol{x}^*, \boldsymbol{y}^*)$ be a local Nash equilibrium. Denote the constant*

$$\gamma := \frac{h(1 + \beta)}{2\sqrt{\epsilon}(1 - \beta)},$$

*then we have*

$$\nabla_x \left( \frac{d\boldsymbol{x}}{dt} \right) \Big|_{(\boldsymbol{x}^*, \boldsymbol{y}^*)} = -\frac{1}{\sqrt{\epsilon}} \left( \nabla_x^2 f(\boldsymbol{x}^*, \boldsymbol{y}^*) + \gamma \left( \nabla_x^2 f \boldsymbol{x}^*, \boldsymbol{y}^*) \cdot \nabla_x^2 f(\boldsymbol{x}^*, \boldsymbol{y}^*) - \nabla_{xy} f(\boldsymbol{x}^*, \boldsymbol{y}^*) \cdot \nabla_{yx} f(\boldsymbol{x}^*, \boldsymbol{y}^*) \right) \right)$$

$$\nabla_y \left( \frac{d\boldsymbol{x}}{dt} \right) \Big|_{(\boldsymbol{x}^*, \boldsymbol{y}^*)} = -\frac{1}{\sqrt{\epsilon}} \left( \nabla_{xy} f(\boldsymbol{x}^*, \boldsymbol{y}^*) + \gamma \left( \nabla_x^2 f(\boldsymbol{x}^*, \boldsymbol{y}^*) \cdot \nabla_{xy} f(\boldsymbol{x}^*, \boldsymbol{y}^*) - \nabla_{xy} f(\boldsymbol{x}^*, \boldsymbol{y}^*) \cdot \nabla_y^2 f(\boldsymbol{x}^*, \boldsymbol{y}^*) \right) \right)$$

$$\nabla_x \left( \frac{d\boldsymbol{y}}{dt} \right) \Big|_{(\boldsymbol{x}^*, \boldsymbol{y}^*)} = \frac{1}{\sqrt{\epsilon}} \left( \nabla_{yx} f(\boldsymbol{x}^*, \boldsymbol{y}^*) + \gamma \left( \nabla_{yx} f(\boldsymbol{x}^*, \boldsymbol{y}^*) \cdot \nabla_x^2 f(\boldsymbol{x}^*, \boldsymbol{y}^*) - \nabla_y^2 f(\boldsymbol{x}^*, \boldsymbol{y}^*) \cdot \nabla_{yx} f(\boldsymbol{x}^*, \boldsymbol{y}^*) \right) \right)$$

$$\nabla_y \left( \frac{d\boldsymbol{y}}{dt} \right) \Big|_{(\boldsymbol{x}^*, \boldsymbol{y}^*)} = \frac{1}{\sqrt{\epsilon}} \left( \nabla_y^2 f(\boldsymbol{x}^*, \boldsymbol{y}^*) + \gamma \left( \nabla_{yx} f(\boldsymbol{x}^*, \boldsymbol{y}^*) \cdot \nabla_{xy} f(\boldsymbol{x}^*, \boldsymbol{y}^*) - \nabla_y^2 f(\boldsymbol{x}^*, \boldsymbol{y}^*) \cdot \nabla_y^2 f(\boldsymbol{x}^*, \boldsymbol{y}^*) \right) \right)$$

*Proof.* Here we only present the detailed proof for $\nabla_x \left( \frac{d\boldsymbol{x}}{dt} \right) \Big|_{(\boldsymbol{x}^*, \boldsymbol{y}^*)}$, other terms can be proved through a similar calculation. Recall that in Continuous Adam-DA, we have

$$\dot{\boldsymbol{x}}(t) = -\mu_\epsilon(\boldsymbol{x}, \boldsymbol{y}) \left( \nabla_x f(\boldsymbol{x}, \boldsymbol{y}) + \frac{h}{2} \mathcal{M}_{\beta,\rho,\epsilon}^\mu(\boldsymbol{x}, \boldsymbol{y}) \cdot \nabla_x \left( \|\nabla_x f(\boldsymbol{x}, \boldsymbol{y})\|_{1,\epsilon} - \|\nabla_y f(\boldsymbol{x}, \boldsymbol{y})\|_{1,\epsilon} \right) \right),$$

thus by the product rule, we have

$$\nabla_x \left( \frac{d\boldsymbol{x}}{dt} \right) \Big|_{(\boldsymbol{x}^*, \boldsymbol{y}^*)} = -\nabla_x \mu_\epsilon(\boldsymbol{x}^*, \boldsymbol{y}^*) \cdot \left( \nabla_x f(\boldsymbol{x}^*, \boldsymbol{y}^*) + \frac{h}{2} \mathcal{M}_{\beta,\rho,\epsilon}^\mu(\boldsymbol{x}^*, \boldsymbol{y}^*) \cdot \nabla_x \left( \|\nabla_x f(\boldsymbol{x}^*, \boldsymbol{y}^*)\|_{1,\epsilon} - \|\nabla_y f(\boldsymbol{x}^*, \boldsymbol{y}^*)\|_{1,\epsilon} \right) \right)$$

$$- \mu_\epsilon(\boldsymbol{x}^*, \boldsymbol{y}^*) \cdot \nabla_x \left( \nabla_x f(\boldsymbol{x}^*, \boldsymbol{y}^*) + \frac{h}{2} \mathcal{M}_{\beta,\rho,\epsilon}^\mu(\boldsymbol{x}^*, \boldsymbol{y}^*) \cdot \nabla_x \left( \|\nabla_x f(\boldsymbol{x}^*, \boldsymbol{y}^*)\|_{1,\epsilon} - \|\nabla_y f(\boldsymbol{x}^*, \boldsymbol{y}^*)\|_{1,\epsilon} \right) \right) \qquad (27)$$

By the definition of local Nash equilibrium, we have

$$\nabla_x f(\boldsymbol{x}^*, \boldsymbol{y}^*) = \boldsymbol{0}, \quad \nabla_y f(\boldsymbol{x}^*, \boldsymbol{y}^*) = \boldsymbol{0}$$

and

$$\mu_\epsilon(\boldsymbol{x}^*, \boldsymbol{y}^*) := \begin{bmatrix} \frac{1}{\sqrt{\epsilon}} & & & \\ & \frac{1}{\sqrt{\epsilon}} & & \\ & & \ddots & \\ & & & \frac{1}{\sqrt{\epsilon}} \end{bmatrix}_{d_1 \times d_1}, \quad \mathcal{M}^\mu_{\beta,\rho,\epsilon}(\boldsymbol{x}^*, \boldsymbol{y}^*) = \mathcal{K}(\beta,\rho)\mathcal{I}_{d_1} + \frac{\epsilon(1+\rho)}{1-\rho}\mu_\epsilon^2(\boldsymbol{x}^*, \boldsymbol{y}^*).$$

Moreover, future simplification gives

$$\begin{aligned}
\mathcal{M}^\mu_{\beta,\rho,\epsilon}(\boldsymbol{x}^*, \boldsymbol{y}^*) &= \mathcal{K}(\beta,\rho)\mathcal{I}_{d_1} + \frac{\epsilon(1+\rho)}{1-\rho}\mu_\epsilon^2(\boldsymbol{x}^*, \boldsymbol{y}^*) \\
&= \mathcal{K}(\beta,\rho)\mathcal{I}_{d_1} + \frac{1+\rho}{1-\rho}\mathcal{I}_{d_1} \\
&= \frac{1+\beta}{1-\beta}\mathcal{I}_{d_1}, \quad \text{since} \ \ \mathcal{K}(\beta,\rho) = (1+\beta)/(1-\beta) + (1+\rho)/(1-\rho).
\end{aligned}$$

Thus the first term on the right hand side of (27) is $\boldsymbol{0}$, and we have

$$\begin{aligned}
&\nabla_x \left(\frac{d\boldsymbol{x}}{dt}\right)\Big|_{(\boldsymbol{x}^*, \boldsymbol{y}^*)} \\
&= -\mu_\epsilon(\boldsymbol{x}^*, \boldsymbol{y}^*) \cdot \nabla_x \left(\nabla_x f(\boldsymbol{x}^*, \boldsymbol{y}^*) + \frac{h}{2}\mathcal{M}^\mu_{\beta,\rho,\epsilon}(\boldsymbol{x}^*, \boldsymbol{y}^*) \cdot \nabla_x \left(\|\nabla_x f(\boldsymbol{x}^*, \boldsymbol{y}^*)\|_{1,\epsilon} - \|\nabla_y f(\boldsymbol{x}^*, \boldsymbol{y}^*)\|_{1,\epsilon}\right)\right) \\
&= -\frac{1}{\sqrt{\epsilon}} \cdot \left(\nabla_x^2 f(\boldsymbol{x}^*, \boldsymbol{y}^*) + \frac{h(1+\beta)}{2(1-\beta)} \cdot \nabla_x^2\left(\|\nabla_x f(\boldsymbol{x}^*, \boldsymbol{y}^*)\|_{1,\epsilon} - \|\nabla_y f(\boldsymbol{x}^*, \boldsymbol{y}^*)\|_{1,\epsilon}\right)\right)
\end{aligned} \tag{28}$$

Recall from Lemma C.1, we have

$$\begin{aligned}
&\nabla_x^2 \|\nabla_x f(\boldsymbol{x}, \boldsymbol{y})\|_{1,\epsilon} \\
&\quad = \nabla_x \left(\nabla_x^2 f(\boldsymbol{x}, \boldsymbol{y}) \cdot \mu_\epsilon(\boldsymbol{x}, \boldsymbol{y})\right) \cdot \nabla_x f(\boldsymbol{x}, \boldsymbol{y}) + \nabla_x^2 f(\boldsymbol{x}, \boldsymbol{y}) \cdot \mu_\epsilon(\boldsymbol{x}, \boldsymbol{y}) \cdot \nabla_x^2 f(\boldsymbol{x}, \boldsymbol{y})
\end{aligned}$$

and

$$\begin{aligned}
&\nabla_x^2 \|\nabla_y f(\boldsymbol{x}, \boldsymbol{y})\|_{1,\epsilon} \\
&\quad = \nabla_x \left(\nabla_{xy} f(\boldsymbol{x}, \boldsymbol{y}) \cdot \nu_\epsilon(\boldsymbol{x}, \boldsymbol{y})\right) \cdot \nabla_y f(\boldsymbol{x}, \boldsymbol{y}) + \nabla_{xy} f(\boldsymbol{x}, \boldsymbol{y}) \cdot \mu_\epsilon(\boldsymbol{x}, \boldsymbol{y}) \cdot \nabla_{yx} f(\boldsymbol{x}, \boldsymbol{y})
\end{aligned}$$

Take $(\boldsymbol{x}^*, \boldsymbol{y}^*)$ into above two equalities, and use the fact that $\nabla_y f(\boldsymbol{x}^*, \boldsymbol{y}^*), \nabla_y f(\boldsymbol{x}^*, \boldsymbol{y}^*) = \boldsymbol{0}$, we get

$$\begin{aligned}
\nabla_x^2 \|\nabla_x f(\boldsymbol{x}, \boldsymbol{y})\|_{1,\epsilon} &= \nabla_x^2 f(\boldsymbol{x}^*, \boldsymbol{y}^*) \cdot \mu_\epsilon(\boldsymbol{x}^*, \boldsymbol{y}^*) \cdot \nabla_x^2 f(\boldsymbol{x}^*, \boldsymbol{y}^*) \\
&= \frac{1}{\sqrt{\epsilon}}\nabla_x^2 f(\boldsymbol{x}^*, \boldsymbol{y}^*) \cdot \nabla_x^2 f(\boldsymbol{x}^*, \boldsymbol{y}^*)
\end{aligned}$$

and

$$\begin{aligned}
\nabla_x^2 \|\nabla_y f(\boldsymbol{x}, \boldsymbol{y})\|_{1,\epsilon} &= \nabla_{xy} f(\boldsymbol{x}^*, \boldsymbol{y}^*) \cdot \mu_\epsilon(\boldsymbol{x}^*, \boldsymbol{y}^*) \cdot \nabla_{yx} f(\boldsymbol{x}^*, \boldsymbol{y}^*) \\
&= \frac{1}{\sqrt{\epsilon}}\nabla_{xy} f(\boldsymbol{x}^*, \boldsymbol{y}^*) \cdot \nabla_{yx} f(\boldsymbol{x}^*, \boldsymbol{y}^*)
\end{aligned}$$

Take above two equalities into (28), we get

$$\nabla_x \left( \frac{d\boldsymbol{x}}{dt} \right) |_{(\boldsymbol{x}^*, \boldsymbol{y}^*)} = -\frac{1}{\sqrt{\epsilon}} \left( \nabla_x^2 f(\boldsymbol{x}^*, \boldsymbol{y}^*) + \gamma \left( \nabla_x^2 f(\boldsymbol{x}^*, \boldsymbol{y}^*) \cdot \nabla_x^2 f(\boldsymbol{x}^*, \boldsymbol{y}^*) - \nabla_{xy} f(\boldsymbol{x}^*, \boldsymbol{y}^*) \cdot \nabla_{yx} f(\boldsymbol{x}^*, \boldsymbol{y}^*) \right) \right),$$

this completes the proof for the terms $\nabla_x \left( \frac{d\boldsymbol{x}}{dt} \right) |_{(\boldsymbol{x}^*, \boldsymbol{y}^*)}$. $\qquad\square$

Now we are ready to proof Proposition 4.1.

*Proof of Proposition 4.1.* By definition, the Jacobian $\mathcal{J}_{\text{Adam}}$ of Continuous Adam-DA at $(\boldsymbol{x}^*, \boldsymbol{y}^*)$ is written as

$$\mathcal{J}_{\text{Adam}} = \begin{bmatrix} \nabla_x \left( \frac{d\boldsymbol{x}}{dt} \right) |_{(\boldsymbol{x}^*, \boldsymbol{y}^*)} & \nabla_y \left( \frac{d\boldsymbol{x}}{dt} \right) |_{(\boldsymbol{x}^*, \boldsymbol{y}^*)} \\ \nabla_x \left( \frac{d\boldsymbol{y}}{dt} \right) |_{(\boldsymbol{x}^*, \boldsymbol{y}^*)} & \nabla_y \left( \frac{d\boldsymbol{y}}{dt} \right) |_{(\boldsymbol{x}^*, \boldsymbol{y}^*)} \end{bmatrix}$$

Take the terms of $\nabla_x \left( \frac{d\boldsymbol{x}}{dt} \right) |_{(\boldsymbol{x}^*, \boldsymbol{y}^*)}, \nabla_y \left( \frac{d\boldsymbol{x}}{dt} \right) |_{(\boldsymbol{x}^*, \boldsymbol{y}^*)}, \nabla_x \left( \frac{d\boldsymbol{y}}{dt} \right) |_{(\boldsymbol{x}^*, \boldsymbol{y}^*)}$ and $\nabla_y \left( \frac{d\boldsymbol{y}}{dt} \right) |_{(\boldsymbol{x}^*, \boldsymbol{y}^*)}$ in Lemma C.2 into above, we get

$$\mathcal{J}_{\text{Adam}} = \frac{1}{\sqrt{\epsilon}} \left( \mathcal{I}_{d_1} - \frac{h(1+\beta)}{2\sqrt{\epsilon}(1-\beta)} \mathcal{J} \right) \mathcal{J}, \; \mathcal{J} = \begin{bmatrix} -\nabla_x^2 f(\boldsymbol{x}^*, \boldsymbol{y}^*) & -\nabla_{xy} f(\boldsymbol{x}^*, \boldsymbol{y}^*) \\ \nabla_{yx} f(\boldsymbol{x}^*, \boldsymbol{y}^*) & \nabla_y^2 f(\boldsymbol{x}^*, \boldsymbol{y}^*) \end{bmatrix}.$$

$\qquad\square$

## C.2. Proof of Theorem 4.3.

We first introduce the following lemma, which describe the eigenvalues of matrix polynomial

**Lemma C.3.** *(Graham, 2018) If $p(x)$ is a polynomial and $A \in \mathbb{R}^{n \times n}$, then every eigenvalue of $p(A)$ can be represented by $p(\lambda)$, where $\lambda \in \text{Sp}(\mathcal{J})$.*

In the following, we state a corollary of Assumption 4.2, which is proved in (Wang & Chizat, 2024).

**Lemma C.4.** *[Theorem 2.1 in Wang & Chizat (2024)] Under Assumption 4.2, we have $\Re(\lambda) < 0$ for every $\lambda \in \text{Sp}(\mathcal{J})$.*

*Proof of Theorem 4.3.* According to Proposition 4.1 and Lemma C.3, every eigenvalue of $\mathcal{J}_{\text{Adam}}$ can be represented by a quadratic polynomial

$$\frac{1}{\sqrt{\epsilon}} \left( 1 - \frac{h(1+\beta)}{2\sqrt{\epsilon}(1-\beta)} \lambda \right) \lambda, \tag{29}$$

where $\lambda \in \text{Sp}(\mathcal{J})$ is an eigenvalue of Jacobian, and any number represented by (29) is an eigenvalue of $\mathcal{J}_{\text{Adam}}$.

Thus, to ensure the local convergence of Continuous Adam-DA, we need

$$\Re \left( \left[ 1 - \frac{h(1+\beta)}{2\sqrt{\epsilon}(1-\beta)} \lambda \right] \cdot \lambda \right)$$
$$= \Re(\lambda) - \frac{h(1+\beta)}{2\sqrt{\epsilon}(1-\beta)} \left( \Re(\lambda)^2 - \Im(\lambda)^2 \right) < 0, \; \forall \lambda \in \text{Sp}(\mathcal{J}). \tag{30}$$

From Lemma C.4, $\Re(\lambda)$ is a negative number, and from Assumption 4.2, the term $\Re(\lambda)^2 - \Im(\lambda)^2$ in (30) is a negative number if $\lambda \in \widetilde{\text{Sp}(\mathcal{J})}$. Moreover, for $\lambda \in \text{Sp}(\mathcal{J})/\widetilde{\text{Sp}(\mathcal{J})}$, (30) always satisfied. Thus, for some fixed parameter $\beta, \epsilon$ and $\forall \lambda \in \widetilde{\text{Sp}(\mathcal{J})}$, we need the step size $h$ in (30) satisfies

$$h < \min_{\lambda \in \widetilde{\text{Sp}(\mathcal{J})}} \frac{2\sqrt{\epsilon}(1-\beta)}{(1+\beta)} \frac{|\Re(\lambda)|}{(\Im(\lambda)^2 - \Re(\lambda)^2)}, \tag{31}$$

and this finishes the proof of Theorem 4.3. $\qquad\square$

## C.3. Proof of Theorem 4.4.

**Additional Notations.** Denote the Jacobian matrix of any matrix $A$ by $\mathrm{Jac}(A)$. Denote the spectral radius of any matrix $A$ by $\varrho(A)$.

Define $\boldsymbol{z}_n = (\tilde{\boldsymbol{m}}_n, \tilde{\boldsymbol{v}}_n, \boldsymbol{x}_n, \hat{\boldsymbol{m}}_n, \hat{\boldsymbol{v}}_n, \boldsymbol{y}_n)^\top$. We can rewrite Adam-DA as a time-dependent discrete-time dynamical system $\boldsymbol{z}_{n+1} = T(n, \boldsymbol{z}_n)$, which can be split into an autonomous dynamical system $\bar{T}(\boldsymbol{z}_n)$ and a non-autonomous one $\mathcal{R}(n, \boldsymbol{z}_n)$ in the following form

$$\boldsymbol{z}_{n+1} = T(n, \boldsymbol{z}_n) = \bar{T}(\boldsymbol{z}_n) + \mathcal{R}(n, \boldsymbol{z}_n), \qquad \text{(Non-Autonomous System)}$$

where

$$T(n, \boldsymbol{z}_n) = \begin{bmatrix} \beta\tilde{\boldsymbol{m}}_n + (1-\beta)\nabla_x f(\boldsymbol{x}_n, \boldsymbol{y}_n) \\[2mm] \rho\tilde{\boldsymbol{v}}_n + (1-\rho)\left(\nabla_x f(\boldsymbol{x}_n, \boldsymbol{y}_n)\right)^2 \\[2mm] \boldsymbol{x}_n - h\dfrac{\tilde{\boldsymbol{m}}_{n+1}/(1-\beta^{n+1})}{\sqrt{\tilde{\boldsymbol{v}}_{n+1}/(1-\rho^{n+1})}+\epsilon} \\[2mm] \beta\hat{\boldsymbol{m}}_n + (1-\beta)\nabla_y f(\boldsymbol{x}_n, \boldsymbol{y}_n) \\[2mm] \rho\hat{\boldsymbol{v}}_n + (1-\rho)\left(\nabla_y f(\boldsymbol{x}_n, \boldsymbol{y}_n)\right)^2 \\[2mm] \boldsymbol{y}_n + h\dfrac{\hat{\boldsymbol{m}}_{n+1}/(1-\beta^{n+1})}{\sqrt{\hat{\boldsymbol{v}}_{n+1}/(1-\rho^{n+1})}+\epsilon} \end{bmatrix}, \quad \bar{T}(\boldsymbol{z}_n) = \begin{bmatrix} \beta\tilde{\boldsymbol{m}}_n + (1-\beta)\nabla_x f(\boldsymbol{x}_n, \boldsymbol{y}_n) \\[2mm] \rho\tilde{\boldsymbol{v}}_n + (1-\rho)\left(\nabla_x f(\boldsymbol{x}_n, \boldsymbol{y}_n)\right)^2 \\[2mm] \boldsymbol{x}_n - h\dfrac{\tilde{\boldsymbol{m}}_{n+1}}{\sqrt{\tilde{\boldsymbol{v}}_{n+1}}+\epsilon} \\[2mm] \beta\hat{\boldsymbol{m}}_n + (1-\beta)\nabla_y f(\boldsymbol{x}_n, \boldsymbol{y}_n) \\[2mm] \rho\hat{\boldsymbol{v}}_n + (1-\rho)\left(\nabla_y f(\boldsymbol{x}_n, \boldsymbol{y}_n)\right)^2 \\[2mm] \boldsymbol{y}_n + h\dfrac{\hat{\boldsymbol{m}}_{n+1}}{\sqrt{\hat{\boldsymbol{v}}_{n+1}}+\epsilon} \end{bmatrix},$$

and

$$\mathcal{R}(n, \boldsymbol{z}_n) = \begin{bmatrix} \boldsymbol{0} \\[2mm] \boldsymbol{0} \\[2mm] \dfrac{h\tilde{\boldsymbol{m}}_{n+1}}{\sqrt{\tilde{\boldsymbol{v}}_{n+1}}+\epsilon} - \dfrac{h\tilde{\boldsymbol{m}}_{n+1}/(1-\beta^{n+1})}{\sqrt{\tilde{\boldsymbol{v}}_{n+1}/(1-\rho^{n+1})}+\epsilon} \\[2mm] \boldsymbol{0} \\[2mm] \boldsymbol{0} \\[2mm] \dfrac{-h\hat{\boldsymbol{m}}_{n+1}}{\sqrt{\hat{\boldsymbol{v}}_{n+1}}+\epsilon} + \dfrac{h\hat{\boldsymbol{m}}_{n+1}/(1-\beta^{n+1})}{\sqrt{\hat{\boldsymbol{v}}_{n+1}/(1-\rho^{n+1})}+\epsilon} \end{bmatrix} = \begin{bmatrix} \boldsymbol{0} \\[2mm] \boldsymbol{0} \\[2mm] \dfrac{h\tilde{\boldsymbol{m}}_{n+1}}{\sqrt{\tilde{\boldsymbol{v}}_{n+1}}+\epsilon} - \dfrac{h\sqrt{1-\rho^{n+1}}}{1-\beta^{n+1}}\dfrac{\tilde{\boldsymbol{m}}_{n+1}}{\sqrt{\tilde{\boldsymbol{v}}_{n+1}}+(1-\rho^{n+1})\epsilon} \\[2mm] \boldsymbol{0} \\[2mm] \boldsymbol{0} \\[2mm] \dfrac{-h\hat{\boldsymbol{m}}_{n+1}}{\sqrt{\hat{\boldsymbol{v}}_{n+1}}+\epsilon} + \dfrac{h\sqrt{1-\rho^{n+1}}}{1-\beta^{n+1}}\dfrac{\hat{\boldsymbol{m}}_{n+1}}{\sqrt{\hat{\boldsymbol{v}}_{n+1}}+(1-\rho^{n+1})\epsilon} \end{bmatrix}.$$

With this split in hand, the proof of local convergence of Adam-DA is transformed into the proof of local convergence of $\bar{T}(\boldsymbol{z}_n)$ and $\mathcal{R}(n, \boldsymbol{z}_n)$. The sketched proof of Theorem 4.4 follows by two steps:

- Step 1: We start by proving that finding the Local Nash Equilibrium $(\boldsymbol{x}^*, \boldsymbol{y}^*)$ is equivalent to finding the fixed point $(\boldsymbol{0}, \boldsymbol{0}, \boldsymbol{x}^*, \boldsymbol{0}, \boldsymbol{0}, \boldsymbol{y}^*)$ of $\bar{T}$ in Lemma C.5. We next compute the characteristic polynomial of the Jacobian matrix of $\bar{T}$ at the fixed point. Then by Lemma C.7, we can select the parameters $h$, $\beta$ and $\epsilon$ to ensure that the spectral radius of the Jacobian matrix of $\bar{T}$ is less than 1. Therefore, we can conclude $\bar{T}$ converges locally near the local Nash equilibrium by Lemma C.6.

- Step 2: We prove that $\|\mathcal{R}(n, \boldsymbol{z}_n)\|$, the perturbation term of Non-Autonomous System, converges locally with an exponential rate by direct algebra computation, i.e., the perturbation term of Non-Autonomous System vanish sufficiently fast with an exponential rate. Intuitively, the exponentially vanishing perturbation hardly influence the local convergence of Non-Autonomous System.

- Step 3: We prove that (Non-Autonomous System) consists of an autonomous system $\bar{T}$ with $\varrho\left(\mathrm{Jac}(\bar{T})\right) < 1$ and an exponentially vanishing perturbation will locally converge with an exponential rate in Lemma C.9.

**Lemma C.5.** $(\boldsymbol{x}^*, \boldsymbol{y}^*)$ *is a local Nash equilibrium if and only if* $\boldsymbol{z}^* = (\boldsymbol{0}, \boldsymbol{0}, \boldsymbol{x}^*, \boldsymbol{0}, \boldsymbol{0}, \boldsymbol{y}^*)^\top$ *is the fixed point of* $T$ *and* $\bar{T}$, *and* $\nabla_x^2 f(\boldsymbol{x}^*, \boldsymbol{y}^*) \succeq \boldsymbol{0}$, $\nabla_y^2 f(\boldsymbol{x}^*, \boldsymbol{y}^*) \preceq \boldsymbol{0}$.

*Proof.* On the one hand, if $\boldsymbol{z}^* = (\boldsymbol{0}, \boldsymbol{0}, \boldsymbol{x}^*, \boldsymbol{0}, \boldsymbol{0}, \boldsymbol{y}^*)$ is the fixed point of $\bar{T}$, consider the following equation

$$\bar{T}\left((\boldsymbol{0}, \boldsymbol{0}, \boldsymbol{x}^*, \boldsymbol{0}, \boldsymbol{0}, \boldsymbol{y}^*)^\top\right) = \begin{bmatrix} (1-\beta)\nabla_x f(\boldsymbol{x}^*, \boldsymbol{y}^*) \\ (1-\rho)\left(\nabla_x f(\boldsymbol{x}^*, \boldsymbol{y}^*)\right)^2 \\ \boldsymbol{x}^* \\ (1-\beta)\nabla_y f(\boldsymbol{x}^*, \boldsymbol{y}^*) \\ (1-\rho)\left(\nabla_y f(\boldsymbol{x}^*, \boldsymbol{y}^*)\right)^2 \\ \boldsymbol{y}^* \end{bmatrix} = \begin{bmatrix} \boldsymbol{0} \\ \boldsymbol{0} \\ \boldsymbol{x}^* \\ \boldsymbol{0} \\ \boldsymbol{0} \\ \boldsymbol{y}^* \end{bmatrix},$$

we can get $\nabla_x f(\boldsymbol{x}^*, \boldsymbol{y}^*) = \boldsymbol{0}$ and $\nabla_y f(\boldsymbol{x}^*, \boldsymbol{y}^*) = \boldsymbol{0}$, i.e., $(\boldsymbol{x}^*, \boldsymbol{y}^*)$ is a local Nash equilibrium.

On the other hand, if $\nabla_x f(\boldsymbol{x}^*, \boldsymbol{y}^*) = \boldsymbol{0}$ and $\nabla_y f(\boldsymbol{x}^*, \boldsymbol{y}^*) = \boldsymbol{0}$, we have

$$\bar{T}\left((\tilde{\boldsymbol{m}}^*, \tilde{\boldsymbol{v}}^*, \boldsymbol{x}^*, \hat{\boldsymbol{m}}^*, \hat{\boldsymbol{v}}^*, \boldsymbol{y}^*)^\top\right) = \begin{bmatrix} \beta\tilde{\boldsymbol{m}}^* \\ \rho\tilde{\boldsymbol{v}}^* \\ \boldsymbol{x}^* - h\frac{\tilde{\boldsymbol{m}}^*}{\sqrt{\tilde{\boldsymbol{v}}^*}+\epsilon} \\ \beta\hat{\boldsymbol{m}}^* \\ \rho\hat{\boldsymbol{v}}^* \\ \boldsymbol{y}^* + h\frac{\hat{\boldsymbol{m}}^*}{\sqrt{\hat{\boldsymbol{v}}^*}+\epsilon} \end{bmatrix} = \begin{bmatrix} \tilde{\boldsymbol{m}}^* \\ \tilde{\boldsymbol{v}}^* \\ \boldsymbol{x}^* \\ \hat{\boldsymbol{m}}^* \\ \hat{\boldsymbol{v}}^* \\ \boldsymbol{y}^* \end{bmatrix},$$

solve this fixed point equation, we can get $\tilde{\boldsymbol{m}}^* = \boldsymbol{0}$, $\tilde{\boldsymbol{v}}^* = \boldsymbol{0}$, $\hat{\boldsymbol{m}}^* = \boldsymbol{0}$ and $\hat{\boldsymbol{v}}^* = \boldsymbol{0}$, which means $(\boldsymbol{0}, \boldsymbol{0}, \boldsymbol{x}^*, \boldsymbol{0}, \boldsymbol{0}, \boldsymbol{y}^*)$ is the fixed point of $\bar{T}$. We can implement the similar argument for $T$. $\qquad\square$

**Lemma C.6.** *[Corollary 4.35 in (Elaydi, 2005) and Theorem II.1 in (Bock & Weiß, 2021)] Consider* $\bar{T} : M \to M$ *with a fixed point* $w^*$ *and* $\bar{T}$ *continuously differentiable in an open disk* $B_\delta(w^*) \subset M$ *with radius* $\delta$. *Assume*

$$\varrho(Jac(\bar{T}_{w^*})) < 1,$$

*then there exists* $0 < \delta_0 < \delta$ *and* $0 \le c < 1$ *such that for all* $w_0$ *with* $\|w_0 - w^*\| < \delta_0$ *and for all* $t \in \mathbb{N}$,

$$\|w(t; w_0) - w^*\| \le c^t \|w_0 - w^*\|.$$

**Lemma C.7.** *[Theorem 6.8(b) in (Henrici, 1974)] For a 2rd-order polynomial* $p(\lambda) = \lambda^2 + a\lambda + b$, *where* $a, b \in \mathbb{C}$, *its roots all lie within the open unit disk of the complex plane if and only if*

$$|b| < 1$$

*and*

$$|a - b\bar{a}| < 1 - |b|^2,$$

*where* $\bar{a}$ *is the complex conjugate of* $a$.

Lemma C.8, which describes how to construct an equivalent norm of matrix $J$ satisfying the contraction property when $\varrho(J) < 1$, is the key to proving Lemma C.9.

**Lemma C.8** (Equivalent norm construction (Elaydi, 2005)). *Suppose that the matrix $J$ satisfies $\varrho(J) < 1$. Then there exists a positive definite matrix $P$ defined by*

$$P = \sum_{k=0}^{\infty} (J^{\top})^k J^k,$$

*such that the induced norm*

$$\|\boldsymbol{x}\|_P = \sqrt{\boldsymbol{x}^{\top} P \boldsymbol{x}}$$

*satisfies*

$$\|J\boldsymbol{x}\|_P \le \gamma \|\boldsymbol{x}\|_P, \quad \forall \boldsymbol{x} \in \mathbb{R}^d$$

*for some $\gamma \in (0, 1)$. In other words, $J$ is a contraction in the $\|\cdot\|_P$-norm.*

*Proof.* The following proof comes from standard techniques in matrix calculation, e.g., (Elaydi, 2005). We include it here for the completeness of the proof. The key of the proof is to verify $P = \sum_{k=0}^{\infty} (J^{\top})^k J^k$ is well defined, i.e., $P = \sum_{k=0}^{\infty} (J^{\top})^k J^k$, converges in component-wise. The proof of $P = \sum_{k=0}^{\infty} (J^{\top})^k J^k$ converging in component-wise can be decomposed into the following 5 steps:

**Step 1. Make a Jordan decomposition for $J$.** Over complex field $\mathbb{C}$, there exists an invertible matrix $V$ such that

$$J = V \operatorname{diag}(J_1, \ldots, J_t) V^{-1},$$

where each $J_i$ with size $r_i$ is a Jordan block of the form

$$J_i = \lambda_i \mathcal{I}_{r_i} + N_i,$$

with $N_i$ strictly upper triangular and $N_i^{r_i} = \mathbf{0}$. Let

$$m := \max_i r_i$$

be the size of the largest Jordan block. Then

$$\|J^k\| \le \|V\| \|V^{-1}\| \cdot \max_i \|J_i^k\|.$$

**Step 2. Make binomial expansion of each single Jordan block.** Fix one block $J_i = \lambda_i \mathcal{I}_{r_i} + N_i$. By the binomial expansion,

$$J_i^k = \sum_{s=0}^{r_i-1} \binom{k}{s} \lambda_i^{k-s} N_i^s, \qquad k \ge 0,$$

since $N_i^{r_i} = 0$. Hence

$$\|J_i^k\| \le \sum_{s=0}^{r_i-1} \binom{k}{s} |\lambda_i|^{k-s} \|N_i\|^s.$$

**Step 3. Estimate the combinatorial numbers.** First, $\binom{k}{s} \le k^s/s!$ for $0 \le s \le k$. Second, choose $\mu$ with $\rho(J) < \mu < 1$. Then $|\lambda_i| \le \rho(J) < \mu$, so

$$|\lambda_i|^{k-s} \le \mu^{k-s} = \mu^k \mu^{-s}.$$

Therefore,

$$\|J_i^k\| \le \mu^k \sum_{s=0}^{r_i-1} \frac{k^s}{s!} (\mu^{-1}\|N_i\|)^s.$$

**Step 4. Extract the polynomial factors.** Since $s \le r_i - 1$, for $k \ge 1$ we have $k^s \le k^{r_i-1}$. Hence

$$\|J_i^k\| \le \mu^k k^{r_i-1} \sum_{s=0}^{r_i-1} \frac{(\mu^{-1}\|N_i\|)^s}{s!}.$$

Define the constant

$$C_i := \sum_{s=0}^{r_i-1} \frac{(\mu^{-1}\|N_i\|)^s}{s!}.$$

Thus,

$$\|J_i^k\| \leq C_i \, k^{r_i-1} \, \mu^k.$$

**Step 5. Combine all blocks.** Taking the maximum over all Jordan blocks yields (Recall that $m = \max_i r_i$.)

$$\|J^k\| \leq \|V\|\|V^{-1}\| \cdot \max_i \|J_i^k\| \leq \left(\|V\|\|V^{-1}\| \max_i C_i\right) k^{m-1} \mu^k = C_{\max} \, k^{m-1} \, \mu^k,$$

where we define

$$C_{\max} = \|V\|\|V^{-1}\| \max_i C_i.$$

Therefore, we have

$$\sum_{k=0}^{\infty} \|(J^\top)^k J^k\| \leq \sum_{k=0}^{\infty} \|J^k\|^2 \leq \sum_{k=0}^{\infty} C_{\max}^2 k^{2(m-1)} \mu^{2k} < \infty,$$

i.e., $P = \sum_{k=0}^{\infty}(J^\top)^k J^k$ converges absolutely, thus $P$ converges in component-wise, which means $P$ is well defined. We have proved that $P = \sum_{k=0}^{\infty}(J^\top)^k J^k$ is well defined. Next, we procced to complete the remaining proof of Lemma C.8.

Note that for all $\boldsymbol{x} \in \mathbb{R}^d$,

$$\boldsymbol{x}^\top P \boldsymbol{x} = \sum_{k=0}^{\infty} \boldsymbol{x}^\top (J^\top)^k J^k \boldsymbol{x} = \sum_{k=0}^{\infty} \|J^k x\|^2,$$

we can easily get $P$ is positive definite.

Besides, we can verify

$$J^\top P J = J^\top \left(\sum_{k=0}^{\infty}(J^\top)^k J^k\right) J = \sum_{k=0}^{\infty}(J^\top)^{k+1} J^{k+1} = \sum_{k=1}^{\infty}(J^\top)^k J^k.$$

Then we have

$$P - J^\top P J = \sum_{k=0}^{\infty}(J^\top)^k J^k - \sum_{k=1}^{\infty}(J^\top)^k J^k = (J^\top)^0 J^0 = \mathcal{I}. \tag{32}$$

From (32), for all $\mathbb{R}^d$, we have

$$\boldsymbol{x}^\top J^\top P J \boldsymbol{x} = \boldsymbol{x}^\top (P - \mathcal{I})\boldsymbol{x} = \boldsymbol{x}^\top P \boldsymbol{x} - \|\boldsymbol{x}\|^2.$$

Suppose $\lambda_{\min}$ is the smallest eigenvalue of $P^{-1}$, then we have

$$\begin{aligned}
\boldsymbol{x}^\top J^\top P J \boldsymbol{x} &= \boldsymbol{x}^\top P \boldsymbol{x} - \|\boldsymbol{x}\|^2 \\
&= \boldsymbol{x}^\top P \boldsymbol{x} - \boldsymbol{x}^\top \mathcal{I} \boldsymbol{x} \\
&= \boldsymbol{x}^\top P \boldsymbol{x} - \boldsymbol{x}^\top P^{\frac{1}{2}} P^{-1} P^{\frac{1}{2}} \boldsymbol{x} \\
&\leq \boldsymbol{x}^\top P \boldsymbol{x} - \lambda_{\min} \boldsymbol{x}^\top P \boldsymbol{x} \\
&= (1 - \lambda_{\min})\boldsymbol{x}^\top P \boldsymbol{x}.
\end{aligned}$$

Therefore

$$J^\top P J \preceq (1 - \lambda_{\min})P.$$

Since $P$ is positive definite, then $J^\top P J$ is also positive definite, thus $1 - \lambda_{\min} > 0$. Also, $P^{-1}$ is positive definite implies $\lambda_{\min} > 0$, thus $0 < 1 - \lambda_{\min} < 1$.

Then the operator norm induced by the matrix $P$, $\|\cdot\|_P$ will satisfy (Denote $\gamma^2 = 1 - \lambda_{\min} \in (0,1)$)

$$\|J\|_P^2 := \sup_{\boldsymbol{x} \neq 0} \frac{\|J\boldsymbol{x}\|_P}{\|\boldsymbol{x}\|_P} = \sup_{\boldsymbol{x} \neq 0} \frac{\boldsymbol{x}^\top J^\top P J \boldsymbol{x}}{\boldsymbol{x}^\top P \boldsymbol{x}} \leq 1 - \lambda_{\min} = \gamma^2,$$

i.e.,

$$\|J\boldsymbol{x}\|_P \leq \gamma \|\boldsymbol{x}\|_P, \quad \forall \boldsymbol{x} \in \mathbb{R}^d.$$

$\square$

**Lemma C.9.** *Consider (Non-Autonomous System). Let $\boldsymbol{z}^*$ be a fixed point of $\bar{T}$, and suppose:*

(i) *The Jacobian $Jac(\bar{T}(\boldsymbol{z}^*))$ satisfies $\varrho\left(Jac(\bar{T}(\boldsymbol{z}^*))\right) < 1$.*

(ii) *The perturbation satisfies $\|\mathcal{R}(n, \boldsymbol{z})\| \leq C'\rho^n \|\boldsymbol{z} - \boldsymbol{z}^*\|$ for some $C' > 0$, $\rho \in (0,1)$, uniformly for $\boldsymbol{z}$ in a neighborhood of $\boldsymbol{z}^*$.*

*Then there exists a neighborhood $U$ of $z^*$ and constant $0 < \tilde{\gamma} < 1$ such that for any $z_0 \in U$, the iterates of $\{\boldsymbol{z}_n\}$ satisfy*

$$\|\boldsymbol{z}_n - \boldsymbol{z}^*\|_2 = \mathcal{O}\left(\tilde{\gamma}^n \|\boldsymbol{z}_1 - \boldsymbol{z}^*\|_2\right).$$

*Proof.* By Lemma C.8, there exists a norm $\|\cdot\|_P$ and $\gamma \in (0,1)$ such that

$$\|\bar{T}(\boldsymbol{z}) - \boldsymbol{z}^*\|_P = \|\bar{T}(\boldsymbol{z}) - \bar{T}(\boldsymbol{z}^*)\|_P = \|\bar{T}(\boldsymbol{z} - \boldsymbol{z}^*)\|_P \leq \gamma \|\boldsymbol{z} - \boldsymbol{z}^*\|_P$$

for $\boldsymbol{z}$ close to $\boldsymbol{z}^*$. Now write the iteration as

$$\boldsymbol{z}_{n+1} - \boldsymbol{z}^* = \bar{T}(\boldsymbol{z}_n) - \bar{T}(\boldsymbol{z}^*) + \mathcal{R}(n, \boldsymbol{z}_n).$$

Taking the $\|\cdot\|_P$ norm,

$$\|\boldsymbol{z}_{n+1} - \boldsymbol{z}^*\|_P \leq \gamma \|\boldsymbol{z}_n - \boldsymbol{z}^*\|_P + \|\mathcal{R}(n, \boldsymbol{z}_n)\|_P.$$

By assumption, $\|R(n, z_n)\|_P \leq C'\rho^n \|z_n - z^*\|_P$ for some $C' > 0$. Thus

$$\|\boldsymbol{z}_{n+1} - \boldsymbol{z}^*\|_P \leq (\gamma + C'\rho^n) \|\boldsymbol{z}_n - \boldsymbol{z}^*\|_P.$$

Since $\rho^n \to 0$, for sufficiently large $n$ we have $\gamma + C'\rho^n < \tilde{\gamma} < 1$. Therefore the sequence contracts at rate $\tilde{\gamma} < 1$, implying local convergence:

$$\|\boldsymbol{z}_n - \boldsymbol{z}^*\|_P \leq \tilde{\gamma}^{n-1} \|\boldsymbol{z}_1 - \boldsymbol{z}^*\|_P.$$

By the Equivalence of norms in finite dimensions, we have there exists $c_1, c_2 > 0$ such that

$$c_1 \|\boldsymbol{z}\|_2 \leq \|\boldsymbol{z}\|_P \leq c_2 \|\boldsymbol{z}\|_2, \quad \forall \boldsymbol{z}.$$

This gives the local convergence with an exponential rate in the Euclidean norm as well, i.e.,

$$\|\boldsymbol{z}_n - \boldsymbol{z}^*\|_2 \leq \frac{c_2}{c_1} \tilde{\gamma}^{n-1} \|\boldsymbol{z}_1 - \boldsymbol{z}^*\|_2.$$

$\square$

**Lemma C.10.** *Suppose that $f(\boldsymbol{x}, \boldsymbol{y})$ satisfies Assumption 4.2. Let $\beta \in (-1, 1)$ and $0 < \rho < 1$. Set $h$, $\epsilon$ and $\beta$ such that*

$$h < \min_{\lambda \in \mathrm{Sp}(\mathcal{J})} \frac{2\sqrt{\epsilon}(1 - \beta^2)|\Re(\lambda)|}{(1 + \beta^2)|\lambda|^2 + 2\beta\left(|\Im(\lambda)|^2 - |\Re(\lambda)|^2\right)}.$$

*Then $\bar{T}$ converges locally with an exponential rate.*

*Proof.* According to Lemma C.6, we complete the proof by proving $\varrho\left(\mathrm{Jac}(\bar{T})\right) < 1$.

By direct computation, we can get the Jacobian matrix of $\bar{T}$ at $\boldsymbol{z}^* = (\boldsymbol{0}, \boldsymbol{0}, \boldsymbol{x}^*, \boldsymbol{0}, \boldsymbol{0}, \boldsymbol{y}^*)$ is

$$\mathcal{M}_{\mathcal{S}} = \begin{bmatrix} \beta\mathcal{I} & \boldsymbol{0} & (1-\beta)\nabla_{\boldsymbol{x}}^2 f(\boldsymbol{x}^*,\boldsymbol{y}^*) & \boldsymbol{0} & \boldsymbol{0} & (1-\beta)\nabla_{xy} f(\boldsymbol{x}^*,\boldsymbol{y}^*) \\ \boldsymbol{0} & \rho\mathcal{I} & \boldsymbol{0} & \boldsymbol{0} & \boldsymbol{0} & \boldsymbol{0} \\ -\frac{h\beta}{\sqrt{\epsilon}}\mathcal{I} & \boldsymbol{0} & \mathcal{I} - \frac{h(1-\beta)}{\sqrt{\epsilon}}\nabla_{\boldsymbol{x}}^2 f(\boldsymbol{x}^*,\boldsymbol{y}^*) & \boldsymbol{0} & \boldsymbol{0} & -\frac{h(1-\beta)}{\sqrt{\epsilon}}\nabla_{xy} f(\boldsymbol{x}^*,\boldsymbol{y}^*) \\ \boldsymbol{0} & \boldsymbol{0} & (1-\beta)\nabla_{yx} f(\boldsymbol{x}^*,\boldsymbol{y}^*) & \beta\mathcal{I} & \boldsymbol{0} & (1-\beta)\nabla_{\boldsymbol{y}}^2 f(\boldsymbol{x}^*,\boldsymbol{y}^*) \\ \boldsymbol{0} & \boldsymbol{0} & \boldsymbol{0} & \boldsymbol{0} & \rho\mathcal{I} & \boldsymbol{0} \\ \boldsymbol{0} & \boldsymbol{0} & \frac{h(1-\beta)}{\sqrt{\epsilon}}\nabla_{yx} f(\boldsymbol{x}^*,\boldsymbol{y}^*) & \frac{h\beta}{\sqrt{\epsilon}}\mathcal{I} & \boldsymbol{0} & \mathcal{I} + \frac{h(1-\beta)}{\sqrt{\epsilon}}\nabla_{\boldsymbol{y}}^2 f(\boldsymbol{x}^*,\boldsymbol{y}^*) \end{bmatrix}.$$

Obviously, $\mathcal{M}_S$ has $d_1 + d_2$ eigenvalues $\rho$. Next, we consider the following matrix

$$\mathcal{M}_{\mathcal{S}}^{(1)} = \begin{bmatrix} \beta\mathcal{I} & (1-\beta)\nabla_{\boldsymbol{x}}^2 f(\boldsymbol{x}^*,\boldsymbol{y}^*) & \boldsymbol{0} & (1-\beta)\nabla_{xy} f(\boldsymbol{x}^*,\boldsymbol{y}^*) \\ -\frac{h\beta}{\sqrt{\epsilon}}\mathcal{I} & \mathcal{I} - \frac{h(1-\beta)}{\sqrt{\epsilon}}\nabla_{\boldsymbol{x}}^2 f(\boldsymbol{x}^*,\boldsymbol{y}^*) & \boldsymbol{0} & -\frac{h(1-\beta)}{\sqrt{\epsilon}}\nabla_{xy} f(\boldsymbol{x}^*,\boldsymbol{y}^*) \\ \boldsymbol{0} & (1-\beta)\nabla_{yx} f(\boldsymbol{x}^*,\boldsymbol{y}^*) & \beta\mathcal{I} & (1-\beta)\nabla_{\boldsymbol{y}}^2 f(\boldsymbol{x}^*,\boldsymbol{y}^*) \\ \boldsymbol{0} & \frac{h(1-\beta)}{\sqrt{\epsilon}}\nabla_{yx} f(\boldsymbol{x}^*,\boldsymbol{y}^*) & \frac{h\beta}{\sqrt{\epsilon}}\mathcal{I} & \mathcal{I} + \frac{h(1-\beta)}{\sqrt{\epsilon}}\nabla_{\boldsymbol{y}}^2 f(\boldsymbol{x}^*,\boldsymbol{y}^*) \end{bmatrix}.$$

Since $0 < \rho < 1$, if we want to prove $\varrho\left(\mathrm{Jac}(\bar{T})\right) < 1$, we only need to ensure $\varrho\left(\mathcal{M}_{\mathcal{S}}^{(1)}\right) < 1$.

Exchange the 1st and 4th rows, as well as 1st and 4th columns of $\mathcal{M}_{\mathcal{S}}^{(1)}$, we can get

$$\mathcal{M}_{\mathcal{S}}^{(2)} = \begin{bmatrix} \mathcal{I} + \frac{h(1-\beta)}{\sqrt{\epsilon}}\nabla_{\boldsymbol{y}}^2 f(\boldsymbol{x}^*,\boldsymbol{y}^*) & \frac{h(1-\beta)}{\sqrt{\epsilon}}\nabla_{yx} f(\boldsymbol{x}^*,\boldsymbol{y}^*) & \frac{h\beta}{\sqrt{\epsilon}}\mathcal{I} & \boldsymbol{0} \\ -\frac{h(1-\beta)}{\sqrt{\epsilon}}\nabla_{xy} f(\boldsymbol{x}^*,\boldsymbol{y}^*) & \mathcal{I} - \frac{h(1-\beta)}{\sqrt{\epsilon}}\nabla_{\boldsymbol{x}}^2 f(\boldsymbol{x}^*,\boldsymbol{y}^*) & \boldsymbol{0} & -\frac{h\beta}{\sqrt{\epsilon}}\mathcal{I} \\ (1-\beta)\nabla_{\boldsymbol{y}}^2 f(\boldsymbol{x}^*,\boldsymbol{y}^*) & (1-\beta)\nabla_{yx} f(\boldsymbol{x}^*,\boldsymbol{y}^*) & \beta\mathcal{I} & \boldsymbol{0} \\ (1-\beta)\nabla_{xy} f(\boldsymbol{x}^*,\boldsymbol{y}^*) & (1-\beta)\nabla_{\boldsymbol{x}}^2 f(\boldsymbol{x}^*,\boldsymbol{y}^*) & \boldsymbol{0} & \beta\mathcal{I} \end{bmatrix}.$$

Then we only need to ensure $\varrho\left(\mathcal{M}_{\mathcal{S}}^{(2)}\right) < 1$. Calculate its characteristic polynomial:

$$\det(\hat{\mu}\mathcal{I} - \mathcal{M}_{\mathcal{S}}^{(2)})$$
$$= \det\left(\begin{bmatrix} (\hat{\mu}-1)\mathcal{I} - \frac{h(1-\beta)}{\sqrt{\epsilon}}\nabla_{\boldsymbol{y}}^2 f(\boldsymbol{x}^*,\boldsymbol{y}^*) & -\frac{h(1-\beta)}{\sqrt{\epsilon}}\nabla_{yx} f(\boldsymbol{x}^*,\boldsymbol{y}^*) & -\frac{h\beta}{\sqrt{\epsilon}}\mathcal{I} & \boldsymbol{0} \\ \frac{h(1-\beta)}{\sqrt{\epsilon}}\nabla_{xy} f(\boldsymbol{x}^*,\boldsymbol{y}^*) & (\hat{\mu}-1)\mathcal{I} + \frac{h(1-\beta)}{\sqrt{\epsilon}}\nabla_{\boldsymbol{x}}^2 f(\boldsymbol{x}^*,\boldsymbol{y}^*) & \boldsymbol{0} & \frac{h\beta}{\sqrt{\epsilon}}\mathcal{I} \\ -(1-\beta)\nabla_{\boldsymbol{y}}^2 f(\boldsymbol{x}^*,\boldsymbol{y}^*) & -(1-\beta)\nabla_{yx} f(\boldsymbol{x}^*,\boldsymbol{y}^*) & (\hat{\mu}-\beta)\mathcal{I} & \boldsymbol{0} \\ -(1-\beta)\nabla_{xy} f(\boldsymbol{x}^*,\boldsymbol{y}^*) & -(1-\beta)\nabla_{\boldsymbol{x}}^2 f(\boldsymbol{x}^*,\boldsymbol{y}^*) & \boldsymbol{0} & (\hat{\mu}-\beta)\mathcal{I} \end{bmatrix}\right).$$

Let

$$
A = \begin{bmatrix} (\hat{\mu}-1)\mathcal{I} - \frac{h(1-\beta)}{\sqrt{\epsilon}}\nabla_y^2 f(\boldsymbol{x}^*, \boldsymbol{y}^*) & -\frac{h(1-\beta)}{\sqrt{\epsilon}}\nabla_{yx} f(\boldsymbol{x}^*, \boldsymbol{y}^*) \\[2ex] \frac{h(1-\beta)}{\sqrt{\epsilon}}\nabla_{xy} f(\boldsymbol{x}^*, \boldsymbol{y}^*) & (\hat{\mu}-1)\mathcal{I} + \frac{h(1-\beta)}{\sqrt{\epsilon}}\nabla_x^2 f(\boldsymbol{x}^*, \boldsymbol{y}^*) \end{bmatrix}, B = \begin{bmatrix} -\frac{h\beta}{\sqrt{\epsilon}}\mathcal{I} & \mathbf{0} \\[2ex] \mathbf{0} & \frac{h\beta}{\sqrt{\epsilon}}\mathcal{I} \end{bmatrix},
$$

and

$$
C = \begin{bmatrix} -(1-\beta)\nabla_y^2 f(\boldsymbol{x}^*, \boldsymbol{y}^*) & -(1-\beta)\nabla_{yx} f(\boldsymbol{x}^*, \boldsymbol{y}^*) \\[2ex] -(1-\beta)\nabla_{xy} f(\boldsymbol{x}^*, \boldsymbol{y}^*) & -(1-\beta)\nabla_x^2 f(\boldsymbol{x}^*, \boldsymbol{y}^*) \end{bmatrix}, \quad D = \begin{bmatrix} (\hat{\mu}-\beta)\mathcal{I} & \mathbf{0} \\[2ex] \mathbf{0} & (\hat{\mu}-\beta)\mathcal{I} \end{bmatrix}.
$$

Without loss of generality, we assume $\hat{\mu} \neq \beta$. (If $\hat{\mu} = \beta$, we can easily verify $\beta$ is the unique eigenvalue of $\mathcal{M}_{\mathcal{S}}^{(2)}$. We only need to set $\beta \in (-1,1)$ to ensure $\varrho\left(\mathcal{M}_{\mathcal{S}}^{(2)}\right) < 1$.)

We state a fact that if $D$ is invertible, then $\det\left(\begin{bmatrix} A & B \\ C & D \end{bmatrix}\right) = \det(D)\det(A - BD^{-1}C)$.

According to this fact, we can get

$\det(\hat{\mu}\mathcal{I} - \mathcal{M}_{\mathcal{S}}^{(2)})$

$$
= (\hat{\mu}-\beta)^{d_1+d_2} \det\left(\begin{bmatrix} (\hat{\mu}-1)\mathcal{I} - \frac{h(1-\beta)}{\sqrt{\epsilon}}\nabla_y^2 f(\boldsymbol{x}^*, \boldsymbol{y}^*) & -\frac{h(1-\beta)}{\sqrt{\epsilon}}\nabla_{yx} f(\boldsymbol{x}^*, \boldsymbol{y}^*) \\[2ex] \frac{h(1-\beta)}{\sqrt{\epsilon}}\nabla_{xy} f(\boldsymbol{x}^*, \boldsymbol{y}^*) & (\hat{\mu}-1)\mathcal{I} + \frac{h(1-\beta)}{\sqrt{\epsilon}}\nabla_x^2 f(\boldsymbol{x}^*, \boldsymbol{y}^*) \end{bmatrix}\right.
$$

$$
\left. - \begin{bmatrix} -\frac{h\beta}{\sqrt{\epsilon}}\mathcal{I} & \mathbf{0} \\[2ex] \mathbf{0} & \frac{h\beta}{\sqrt{\epsilon}}\mathcal{I} \end{bmatrix} \begin{bmatrix} \frac{1}{\hat{\mu}-\beta}\mathcal{I} & \mathbf{0} \\[2ex] \mathbf{0} & \frac{1}{\hat{\mu}-\beta}\mathcal{I} \end{bmatrix} \begin{bmatrix} -(1-\beta)\nabla_y^2 f(\boldsymbol{x}^*, \boldsymbol{y}^*) & -(1-\beta)\nabla_{yx}^2 f(\boldsymbol{x}^*, \boldsymbol{y}^*) \\[2ex] -(1-\beta)\nabla_{xy}^2 f(\boldsymbol{x}^*, \boldsymbol{y}^*) & -(1-\beta)\nabla_x^2 f(\boldsymbol{x}^*, \boldsymbol{y}^*) \end{bmatrix}\right)
$$

$$
= (\hat{\mu}-\beta)^{d_1+d_2} \det\left(\begin{bmatrix} (\hat{\mu}-1)\mathcal{I} - \frac{h(1-\beta)\hat{\mu}}{\sqrt{\epsilon}(\hat{\mu}-\beta)}\nabla_y^2 f(\boldsymbol{x}^*, \boldsymbol{y}^*) & -\frac{h(1-\beta)\hat{\mu}}{\sqrt{\epsilon}(\hat{\mu}-\beta)}\nabla_{yx} f(\boldsymbol{x}^*, \boldsymbol{y}^*) \\[2ex] \frac{h(1-\beta)\hat{\mu}}{\sqrt{\epsilon}(\hat{\mu}-\beta)}\nabla_{xy} f(\boldsymbol{x}^*, \boldsymbol{y}^*) & (\hat{\mu}-1)\mathcal{I} + \frac{h(1-\beta)\hat{\mu}}{\sqrt{\epsilon}(\hat{\mu}-\beta)}\nabla_x^2 f(\boldsymbol{x}^*, \boldsymbol{y}^*) \end{bmatrix}\right)
$$

$$
= (\hat{\mu}-\beta)^{d_1+d_2} \det\left(\begin{bmatrix} \hat{\mu}\mathcal{I} & \mathbf{0} \\[2ex] \mathbf{0} & \hat{\mu}\mathcal{I} \end{bmatrix} - \left(\begin{bmatrix} \mathcal{I} & \mathbf{0} \\[2ex] \mathbf{0} & \mathcal{I} \end{bmatrix} - \frac{h(1-\beta)\hat{\mu}}{\sqrt{\epsilon}(\hat{\mu}-\beta)}\begin{bmatrix} -\nabla_y^2 f(\boldsymbol{x}^*, \boldsymbol{y}^*) & -\nabla_{yx} f(\boldsymbol{x}^*, \boldsymbol{y}^*) \\[2ex] \nabla_{xy} f(\boldsymbol{x}^*, \boldsymbol{y}^*) & \nabla_x^2 f(\boldsymbol{x}^*, \boldsymbol{y}^*) \end{bmatrix}\right)\right).
$$

Let $\det(\hat{\mu}\mathcal{I} - \mathcal{M}_{\mathcal{S}}^{(2)}) = 0$. Obviously, $\beta \in (-1,1)$ have ensured that the $d_1 + d_2$ roots $\hat{\mu} = \beta$ lie in the unit disk, we only need to consider

$$
\det\left(\begin{bmatrix} \hat{\mu}\mathcal{I} & \mathbf{0} \\[2ex] \mathbf{0} & \hat{\mu}\mathcal{I} \end{bmatrix} - \left(\begin{bmatrix} \mathcal{I} & \mathbf{0} \\[2ex] \mathbf{0} & \mathcal{I} \end{bmatrix} - \frac{h(1-\beta)\hat{\mu}}{\sqrt{\epsilon}(\hat{\mu}-\beta)}\begin{bmatrix} -\nabla_y^2 f(\boldsymbol{x}^*, \boldsymbol{y}^*) & -\nabla_{yx} f(\boldsymbol{x}^*, \boldsymbol{y}^*) \\[2ex] \nabla_{xy} f(\boldsymbol{x}^*, \boldsymbol{y}^*) & \nabla_x^2 f(\boldsymbol{x}^*, \boldsymbol{y}^*) \end{bmatrix}\right)\right) = \mathbf{0},
$$

which means $\hat{\mu}$ is the eigenvalue of the following matrix

$$N := \begin{bmatrix} \mathcal{I} & \mathbf{0} \\ \mathbf{0} & \mathcal{I} \end{bmatrix} - \frac{h(1-\beta)\hat{\mu}}{\sqrt{\epsilon}(\hat{\mu}-\beta)} \begin{bmatrix} -\nabla_y^2 f(\boldsymbol{x}^*, \boldsymbol{y}^*) & -\nabla_{yx} f(\boldsymbol{x}^*, \boldsymbol{y}^*) \\ \nabla_{xy} f(\boldsymbol{x}^*, \boldsymbol{y}^*) & \nabla_x^2 f(\boldsymbol{x}^*, \boldsymbol{y}^*) \end{bmatrix}.$$

Note that

$$\mathrm{Sp}\left( \begin{bmatrix} -\nabla_y^2 f(\boldsymbol{x}^*, \boldsymbol{y}^*) & -\nabla_{yx} f(\boldsymbol{x}^*, \boldsymbol{y}^*) \\ \nabla_{xy} f(\boldsymbol{x}^*, \boldsymbol{y}^*) & \nabla_x^2 f(\boldsymbol{x}^*, \boldsymbol{y}^*) \end{bmatrix} \right) = \mathrm{Sp}\left( \begin{bmatrix} \nabla_x^2 f(\boldsymbol{x}^*, \boldsymbol{y}^*) & \nabla_{xy} f(\boldsymbol{x}^*, \boldsymbol{y}^*) \\ -\nabla_{yx} f(\boldsymbol{x}^*, \boldsymbol{y}^*) & -\nabla_y^2 f(\boldsymbol{x}^*, \boldsymbol{y}^*) \end{bmatrix} \right)$$

$$= -\mathrm{Sp}\left( \begin{bmatrix} -\nabla_x^2 f(\boldsymbol{x}^*, \boldsymbol{y}^*) & -\nabla_{xy} f(\boldsymbol{x}^*, \boldsymbol{y}^*) \\ \nabla_{yx} f(\boldsymbol{x}^*, \boldsymbol{y}^*) & \nabla_y^2 f(\boldsymbol{x}^*, \boldsymbol{y}^*) \end{bmatrix} \right) = -\mathrm{Sp}\left( \mathcal{J} \right).$$

Let $\{\lambda_i\}_{i=1}^{d_1+d_2}$ be the eigenvalues of the matrix $\mathcal{J}$. Then $\left\{ 1 - \frac{h(1-\beta)\hat{\mu}}{\sqrt{\epsilon}(\hat{\mu}-\beta)}(-\lambda_i) \right\}_{i=1}^{d_1+d_2}$ are the eigenvalues of $N$. Then we can get

$$\hat{\mu} = 1 - \frac{h(1-\beta)\hat{\mu}}{\sqrt{\epsilon}(\hat{\mu}-\beta)}(-\lambda_i), \quad i = 1, 2, \cdots, d_1 + d_2. \tag{33}$$

Solving the equation (33), we can get all the eigenvalues of $\mathcal{M}_{\mathcal{S}}^{(2)}$.

Rewriting (33), we can get

$$\hat{\mu}^2 - \left( \beta + 1 + \frac{h(1-\beta)}{\sqrt{\epsilon}}\lambda_i \right)\hat{\mu} + \beta = 0. \tag{34}$$

Applying Lemma C.7 for (34) with

$$a = -\left( \beta + 1 - \frac{h(1-\beta)}{\sqrt{\epsilon}}\lambda_i \right), \quad b = \beta,$$

and solving the resulting inequalities, we can get

$$h < \min_i \frac{-2\sqrt{\epsilon}(1-\beta^2)\Re(\lambda_i)}{(1+\beta^2)|\lambda_i|^2 + 2\beta\left(|\Im(\lambda_i)|^2 - |\Re(\lambda_i)|^2\right)} \tag{35}$$

$$= \min_{\lambda \in \mathrm{Sp}(\mathcal{J})} \frac{2\sqrt{\epsilon}(1-\beta^2)|\Re(\lambda)|}{(1+\beta^2)|\lambda|^2 + 2\beta\left(|\Im(\lambda)|^2 - |\Re(\lambda)|^2\right)}, \tag{36}$$

where we use the fact that Assumption 4.2 implies $\Re(\lambda_i) < 0$ in the equality.

This means all roots of (34) lie within the open unit disk of the complex plane, which means the spectral radius of $\bar{T}$ is less than 1, i.e., $\varrho(\mathrm{Jac}(\bar{T})) < 1$. $\qquad \square$

**Lemma C.11.** *Assume $f(x)$ is $C^2$. Then the non-autonomous system $\mathcal{R}(n, \boldsymbol{z}_n)$ converges locally with an exponential rate, i.e., there exists $0 < Q < 1$ such that*

$$\|\mathcal{R}(n, \boldsymbol{z}_n)\| = \mathcal{O}(Q^n \|\boldsymbol{z}_n - \boldsymbol{z}^*\|).$$

*Proof.* For the non-autonomous system, we have

$$\|\mathcal{R}(n, \boldsymbol{z}_n)\|^2 = h^2 \underbrace{\left\| \frac{\tilde{\boldsymbol{m}}_{n+1}}{\sqrt{\tilde{\boldsymbol{v}}_{n+1}} + \epsilon} - \frac{\sqrt{1-\rho^{n+1}}}{1-\beta^{n+1}} \frac{\tilde{\boldsymbol{m}}_{n+1}}{\sqrt{\tilde{\boldsymbol{v}}_{n+1} + (1-\rho^{n+1})\epsilon}} \right\|^2}_{\mathrm{Term1}}$$

$$+ h^2 \underbrace{\left\| \frac{\hat{\boldsymbol{m}}_{n+1}}{\sqrt{\hat{\boldsymbol{v}}_{n+1}} + \epsilon} - \frac{\sqrt{1-\rho^{n+1}}}{1-\beta^{n+1}} \frac{\hat{\boldsymbol{m}}_{n+1}}{\sqrt{\hat{\boldsymbol{v}}_{n+1} + (1-\rho^{n+1})\epsilon}} \right\|^2}_{\mathrm{Term2}}.$$

For Term 1, we have

Term1

$$= h^2 \left\| \frac{\tilde{\boldsymbol{m}}_{n+1}}{\sqrt{\tilde{\boldsymbol{v}}_{n+1}} + \epsilon} - \frac{\sqrt{1-\rho^{n+1}}}{1-\beta^{n+1}} \frac{\tilde{\boldsymbol{m}}_{n+1}}{\sqrt{\tilde{\boldsymbol{v}}_{n+1}} + \epsilon} + \frac{\sqrt{1-\rho^{n+1}}}{1-\beta^{n+1}} \left( \frac{\tilde{\boldsymbol{m}}_{n+1}}{\sqrt{\tilde{\boldsymbol{v}}_{n+1}} + \epsilon} - \frac{\tilde{\boldsymbol{m}}_{n+1}}{\sqrt{\tilde{\boldsymbol{v}}_{n+1} + (1-\rho^{n+1})\epsilon}} \right) \right\|^2$$

$$\leq 2h^2 \left\| \frac{\tilde{\boldsymbol{m}}_{n+1}}{\sqrt{\tilde{\boldsymbol{v}}_{n+1}} + \epsilon} - \frac{\sqrt{1-\rho^{n+1}}}{1-\beta^{n+1}} \frac{\tilde{\boldsymbol{m}}_{n+1}}{\sqrt{\tilde{\boldsymbol{v}}_{n+1}} + \epsilon} \right\|^2 + 2h^2 \left\| \frac{\sqrt{1-\rho^{n+1}}}{1-\beta^{n+1}} \left( \frac{\tilde{\boldsymbol{m}}_{n+1}}{\sqrt{\tilde{\boldsymbol{v}}_{n+1}} + \epsilon} - \frac{\tilde{\boldsymbol{m}}_{n+1}}{\sqrt{\tilde{\boldsymbol{v}}_{n+1} + (1-\rho^{n+1})\epsilon}} \right) \right\|^2$$

$$= 2h^2 \left( 1 - \frac{\sqrt{1-\rho^{n+1}}}{1-\beta^{n+1}} \right)^2 \frac{\|\tilde{\boldsymbol{m}}_{n+1}\|^2}{\tilde{\boldsymbol{v}}_{n+1} + \epsilon} + 2h^2 \left( \frac{\sqrt{1-\rho^{n+1}}}{1-\beta^{n+1}} \right)^2 \left\| \frac{\tilde{\boldsymbol{m}}_{n+1}}{\sqrt{\tilde{\boldsymbol{v}}_{n+1}} + \epsilon} - \frac{\tilde{\boldsymbol{m}}_{n+1}}{\sqrt{\tilde{\boldsymbol{v}}_{n+1} + (1-\rho^{n+1})\epsilon}} \right\|^2$$

$$\leq \frac{2h^2\|\tilde{\boldsymbol{m}}_{n+1}\|^2}{\epsilon} \left( \frac{\sqrt{1-\rho^{n+1}} - (1-\beta^{n+1})}{1-\beta^{n+1}} \right)^2 + \frac{2h^2\|\tilde{\boldsymbol{m}}_{n+1}\|^2}{(1-\beta)^2} \left( \frac{1}{\sqrt{\tilde{\boldsymbol{v}}_{n+1}} + \epsilon} - \frac{1}{\sqrt{\tilde{\boldsymbol{v}}_{n+1} + (1-\rho^{n+1})\epsilon}} \right)^2,$$

where we use the fact $\|\boldsymbol{u} + \boldsymbol{v}\|^2 \leq 2(\|\boldsymbol{u}\|^2 + \|\boldsymbol{v}\|^2)$ in the first inequality.

Direct computation yields

$$\left( \frac{\sqrt{1-\rho^{n+1}} - (1-\beta^{n+1})}{1-\beta^{n+1}} \right)^2 = \left( \frac{1 - \rho^{n+1} - (1-\beta^{n+1})^2}{(1-\beta^{n+1})\left(\sqrt{1-\rho^{n+1}} + (1-\beta^{n+1})\right)} \right)^2$$

$$\leq \left( \frac{-\rho^{n+1} + 2\beta^{n+1} - (\rho^2)^{n+1}}{(1-\beta)\left(\sqrt{1-\rho} + 1 - \beta\right)} \right)^2$$

$$\leq \left( \frac{|-\rho^{n+1}| + 2|\beta^{n+1}| + |-(\rho^2)^{n+1}|}{(1-\beta)\left(\sqrt{1-\rho} + 1 - \beta\right)} \right)^2$$

$$\leq \left( \frac{4Q^{n+1}}{(1-\beta)\left(\sqrt{1-\rho} + 1 - \beta\right)} \right)^2 = C_1 Q^{2n+2}$$

Here

$$Q = \max\{|\rho|, |\beta|^2, |\rho^2|\}, \quad C_1 = \frac{16}{(1-\beta)^2(\sqrt{1-\rho} + 1 - \beta)^2},$$

and we use the fact

$$|a + b + c|^2 \leq (|a| + |b| + |c|)^2$$

in the second inequality.

Simple algebra operation yields

$$\left(\frac{1}{\sqrt{\tilde{v}_{n+1}+\epsilon}} - \frac{1}{\sqrt{\tilde{v}_{n+1}+(1-\rho^{n+1})\epsilon}}\right)^2$$

$$= \left(\frac{\rho^{n+1}\epsilon}{\sqrt{\tilde{v}_{n+1}+\epsilon}\sqrt{\tilde{v}_{n+1}+(1-\rho^{n+1})\epsilon}\left(\sqrt{\tilde{v}_{n+1}+\epsilon}+\sqrt{\tilde{v}_{n+1}+(1-\rho^{n+1})\epsilon}\right)}\right)^2$$

$$\leq \left(\frac{\rho^{n+1}\epsilon}{\sqrt{\epsilon}\sqrt{(1-\rho)\epsilon}\left(\sqrt{\epsilon}+\sqrt{(1-\rho)\epsilon}\right)}\right)^2$$

$$= \left(\frac{\rho^{n+1}}{\sqrt{\epsilon}\sqrt{(1-\rho)}\left(1+\sqrt{1-\rho}\right)}\right)^2 = C_2\rho^{2n+2},$$

where

$$C_2 = \frac{1}{\epsilon(1-\rho)(1+\sqrt{1-\rho})^2}.$$

Since $f(x)$ is $C^2$, $\|\nabla^2 f(\boldsymbol{x},\boldsymbol{y})\|$ is bounded in any bounded set, which means $\nabla_x f(\boldsymbol{x},\boldsymbol{y})$ is locally Lipschitz in any bounded set, i.e., there exists $L > 0$ such that for all $(\boldsymbol{x}_1,\boldsymbol{y}_1), (\boldsymbol{x}_2,\boldsymbol{y}_2)$ in the neighborhood $B(\boldsymbol{x}^*,\boldsymbol{y}^*)$ of $(\boldsymbol{x}^*,\boldsymbol{y}^*)$,

$$\|\nabla_x f(\boldsymbol{x}_1,\boldsymbol{y}_1) - \nabla_x f(\boldsymbol{x}_2,\boldsymbol{y}_2)\| \leq L\|(\boldsymbol{x}_1,\boldsymbol{y}_1) - (\boldsymbol{x}_2,\boldsymbol{y}_2)\|.$$

Recall that the fixed point $(\tilde{\boldsymbol{m}}^*, \tilde{\boldsymbol{v}}^*, \boldsymbol{x}^*, \hat{\boldsymbol{m}}^*, \hat{\boldsymbol{v}}^*, \boldsymbol{y}^*) = (\boldsymbol{0},\boldsymbol{0},\boldsymbol{x}^*,\boldsymbol{0},\boldsymbol{0},\boldsymbol{y}^*)$.

$$\begin{aligned}
\|\tilde{\boldsymbol{m}}_{n+1}\| &= \|\tilde{\boldsymbol{m}}_{n+1} - \tilde{\boldsymbol{m}}^*\| \\
&\leq \beta\|\tilde{\boldsymbol{m}}_n - \tilde{\boldsymbol{m}}^*\| + (1-\beta)\|\nabla_x f^{(n)}(\boldsymbol{x}_n,\boldsymbol{y}_n) - \nabla_x f^{(n)}(\boldsymbol{x}^*,\boldsymbol{y}^*)\| \\
&\leq \beta\|\tilde{\boldsymbol{m}}_n - \tilde{\boldsymbol{m}}^*\| + (1-\beta)L\|(\boldsymbol{x}_n,\boldsymbol{y}_n) - (\boldsymbol{x}^*,\boldsymbol{y}^*)\|.
\end{aligned} \tag{37}$$

In fact,

$$\beta\|\tilde{\boldsymbol{m}} - \tilde{\boldsymbol{m}}^*\| + (1-\beta)L\|(\tilde{\boldsymbol{x}},\tilde{\boldsymbol{y}}) - (\boldsymbol{x}^*,\boldsymbol{y}^*)\|$$

corresponds to a norm

$$\|(\tilde{\boldsymbol{m}},\tilde{\boldsymbol{x}},\tilde{\boldsymbol{y}})\|_* := \beta\|\tilde{\boldsymbol{m}}\| + (1-\beta)L\|(\tilde{\boldsymbol{x}},\tilde{\boldsymbol{y}})\|. \tag{38}$$

Next, we verify that $\|\cdot\|_*$ is exactly a norm:

- **Positive definiteness.** Obviously, we have

$$\|(\tilde{\boldsymbol{m}},\tilde{\boldsymbol{x}},\tilde{\boldsymbol{y}})\|_* = \beta\|\tilde{\boldsymbol{m}}\| + (1-\beta)L\|(\tilde{\boldsymbol{x}},\tilde{\boldsymbol{y}})\| \geq 0$$

  and $\|(\tilde{\boldsymbol{m}},\tilde{\boldsymbol{x}},\tilde{\boldsymbol{y}})\|_* = \boldsymbol{0}$ iff $(\tilde{\boldsymbol{m}},\tilde{\boldsymbol{x}},\tilde{\boldsymbol{y}}) = (\boldsymbol{0},\boldsymbol{0},\boldsymbol{0})$, since $0 < \beta < 1$ and $L > 0$.

- **Absolute homogeneity.** For all $a$ and all $(\tilde{\boldsymbol{m}},\tilde{\boldsymbol{x}},\tilde{\boldsymbol{y}})$, we have

$$\begin{aligned}
\|a\,(\tilde{\boldsymbol{m}},\tilde{\boldsymbol{x}},\tilde{\boldsymbol{y}})\|_* &= \beta\,\||a|\,\tilde{\boldsymbol{m}}\| + (1-\beta)L\,\||a|\,(\tilde{\boldsymbol{x}},\tilde{\boldsymbol{y}})\| \\
&= |a|\,\beta\,\|\tilde{\boldsymbol{m}}\| + |a|\,(1-\beta)L\,\|(\tilde{\boldsymbol{x}},\tilde{\boldsymbol{y}})\| \\
&= |a|\,(\beta\,\|\tilde{\boldsymbol{m}}\| + (1-\beta)L\,\|(\tilde{\boldsymbol{x}},\tilde{\boldsymbol{y}})\|) \\
&= |a|\,\|(\tilde{\boldsymbol{m}},\tilde{\boldsymbol{x}},\tilde{\boldsymbol{y}})\|_*
\end{aligned}$$

- **Triangle inequality.** For all $(\tilde{\bm{m}}_1, \tilde{\bm{x}}_1, \tilde{\bm{y}}_1)$ and $(\tilde{\bm{m}}_2, \tilde{\bm{x}}_2, \tilde{\bm{y}}_2)$, we have

$$
\begin{aligned}
&\|(\tilde{\bm{m}}_1, \tilde{\bm{x}}_1, \tilde{\bm{y}}_1) + (\tilde{\bm{m}}_2, \tilde{\bm{x}}_2, \tilde{\bm{y}}_2)\|_* \\
&= \|(\tilde{\bm{m}}_1 + \tilde{\bm{m}}_2, \tilde{\bm{x}}_1 + \tilde{\bm{x}}_2, \tilde{\bm{y}}_1 + \tilde{\bm{y}}_2)\|_* \\
&= \beta \|\tilde{\bm{m}}_1 + \tilde{\bm{m}}_2\| + (1 - \beta)L \|(\tilde{\bm{x}}_1 + \tilde{\bm{x}}_2, \tilde{\bm{y}}_1 + \tilde{\bm{y}}_2)\| \\
&\leq \beta \left( \|\tilde{\bm{m}}_1\| + \|\tilde{\bm{m}}_2\| \right) + (1 - \beta)L \left( \|(\tilde{\bm{x}}_1, \tilde{\bm{y}}_1)\| + \|(\tilde{\bm{x}}_2, \tilde{\bm{y}}_2)\| \right) \\
&= \beta \|\tilde{\bm{m}}_1\| + (1 - \beta)L \|(\tilde{\bm{x}}_1, \tilde{\bm{y}}_1)\| + \beta \|\tilde{\bm{m}}_2\| + (1 - \beta)L \|(\tilde{\bm{x}}_2, \tilde{\bm{y}}_2)\| \\
&= \|(\tilde{\bm{m}}_1, \tilde{\bm{x}}_1, \tilde{\bm{y}}_1)\|_* + \|(\tilde{\bm{m}}_2, \tilde{\bm{x}}_2, \tilde{\bm{y}}_2)\|_* .
\end{aligned}
$$

We have verified $\|\cdot\|_*$ is exactly a norm.

By the equivalence of norms, there exists $\tilde{C} > 0$ such that

$$
\|(\tilde{\bm{m}}, \tilde{\bm{x}}, \tilde{\bm{y}})\|_* \leq \tilde{C}\|(\tilde{\bm{m}}, \tilde{\bm{x}}, \tilde{\bm{y}})\|. \tag{39}
$$

Combining (37), (38) and (39), we have

$$
\|\tilde{\bm{m}}^{(n+1)}\| = \|\tilde{\bm{m}}^{(n+1)} - \tilde{\bm{m}}^*\| \leq \tilde{C}\|(\tilde{\bm{m}}_n - \tilde{\bm{m}}^*, \bm{x}_n - \bm{x}^*, \bm{y}_n - \bm{y}^*)\| \leq \tilde{C}\|\bm{z}_n - \bm{z}^*\|,
$$

since $\tilde{\bm{m}}^* = \bm{0}$ according to Lemma C.5.

Putting all the above facts together, we have

$$
\text{Term 1} \leq \frac{2h^2 C_1 \tilde{C}^2 \|\bm{z}_n - \bm{z}^*\|^2}{\epsilon} Q^{2n+2} + \frac{2h^2 C_2 \tilde{C}^2 \|\bm{z}_n - \bm{z}^*\|^2}{(1 - \beta)^2} \rho^{2n+2}
$$

$$
\leq \left( \frac{2h^2 C_1 \tilde{C}^2 \|\bm{z}_n - \bm{z}^*\|^2}{\epsilon} + \frac{2h^2 C_2 \tilde{C}^2 \|\bm{z}_n - \bm{z}^*\|^2}{(1 - \beta)^2} \right) Q^{2n+2}
$$

$$
= D_1 Q^{2n+2} \|\bm{z}_n - \bm{z}^*\|^2,
$$

where

$$
D_1 = \frac{2h^2 C_1 \tilde{C}^2}{\epsilon} + \frac{2h^2 C_2 \tilde{C}^2}{(1 - \beta)^2}.
$$

As for Term 2, by a similar argument of Term 1, we have

$$
\text{Term 2} \leq D_2 Q^{2n+2} \|\bm{z}_n - \bm{z}^*\|^2.
$$

Eventually, we can get $\|\mathcal{R}(n, \bm{z}_n)\| = \mathcal{O}(Q^n \|\bm{z}_n - \bm{z}^*\|)$, where $Q < 1$. $\qquad\square$

Lastly, putting Lemma C.10, Lemma C.11 together with Lemma C.9, we can get Adam-DA converges locally with an exponential rate and this completes the proof of Theorem 4.4.

**C.4. Future Details of Figure 2.**

The details of Figure 2 are:

- For the Adam-DA and Continuous Adam-DA, the test function is defined as

$$\min_x \max_y 0.2x^2 - xy + 0.2y^2,$$

  which is a convex-concave function. We fix $\epsilon = 10^{-3}$.

- For Adam in minimization problem, the test function is defined as

$$\min_{(x,y)} x^2 + y^2$$

  which is a convex function. We fix $\epsilon = 10^{-4}$.

In Figure 7, we present how the range of $h$ changes with $\epsilon$; the x-axis represents $\epsilon$ and the y-axis represents $h$. We can observe that larger $\epsilon$ values expand the convergence range for all three methods, supporting Theorem 4.3 and 4.4.

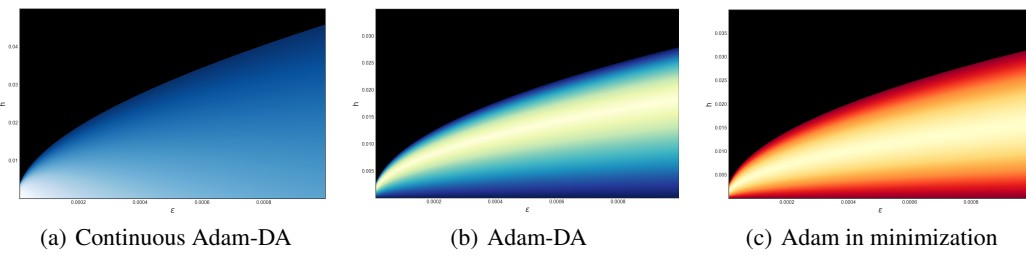

(a) Continuous Adam-DA        (b) Adam-DA        (c) Adam in minimization

*Figure 7. Effect of $\epsilon$.*

# D. Additional Materials for Section 5

## D.1. Additional Experiments

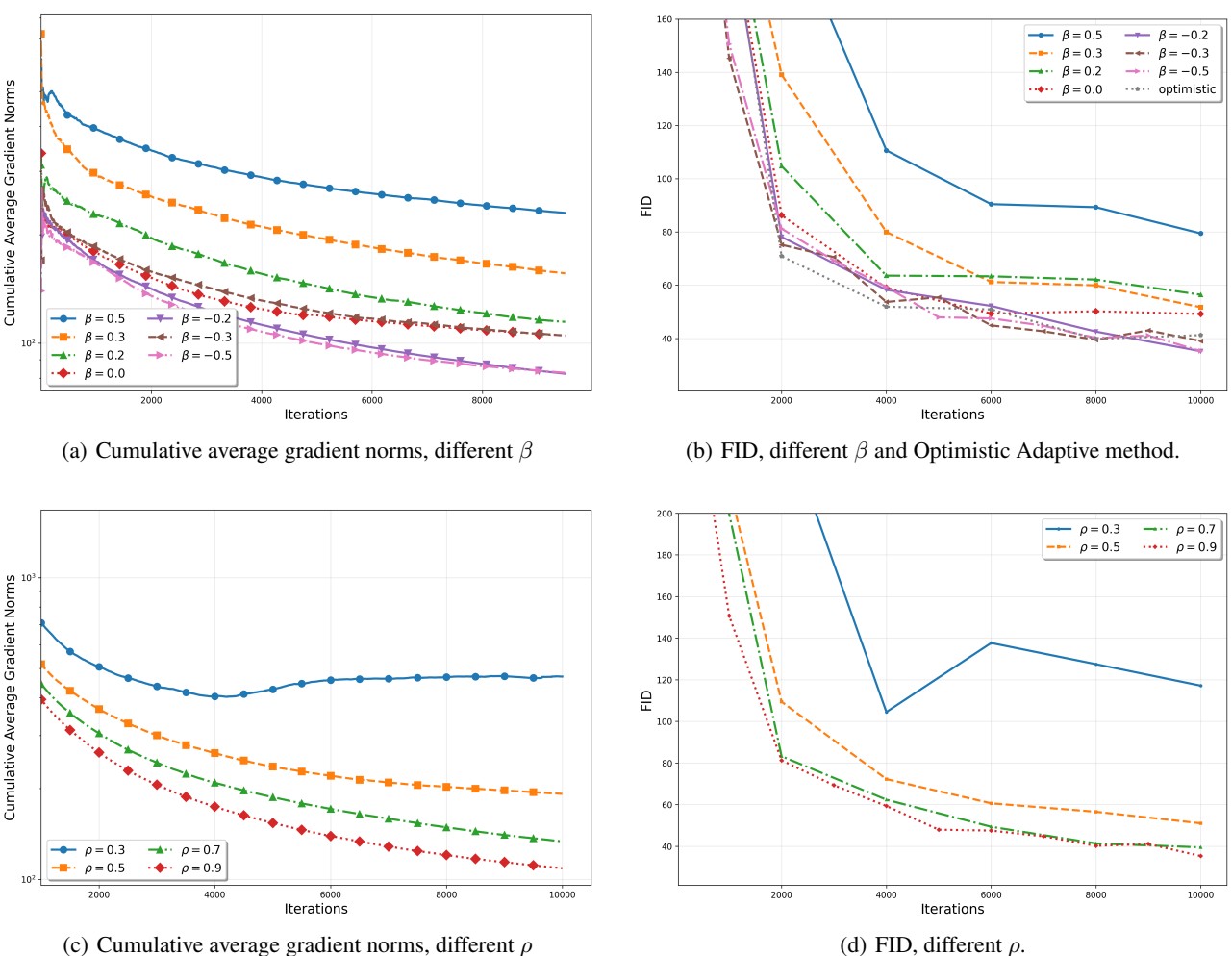

(a) Cumulative average gradient norms, different $\beta$

(b) FID, different $\beta$ and Optimistic Adaptive method.

(c) Cumulative average gradient norms, different $\rho$

(d) FID, different $\rho$.

*Figure 8.* Self-Attention GAN Experiments on CelebA, Evaluated by FID. In Figure 1(a), we present the evolution of the cumulative average gradient norms for $\rho = 0.9$ and different values of $\beta$. In Figure 1(b), we compare the FID scores for these parameter settings with those of the *optimistic* adaptive method proposed by Daskalakis et al. We observe that, in general, smaller $\beta$ leads to lower gradient norms and lower FID, indicating better training performance. Moreover, by comparing the optimistic method with Adam, we find that **although the optimistic method performs better than Adam with $\beta \geq 0$, it does not outperform Adam with negative $\beta$**. In Figures 1(c) and 1(d), we present the evolution of the cumulative average gradient norms and the FID scores for $\beta = -0.5$ and different values of $\rho$. We also observe that larger $\rho$ leads to lower gradient norms and lower FID, indicating better training performance. These results suggest that our findings in Section 5 also hold in this experimental setting.

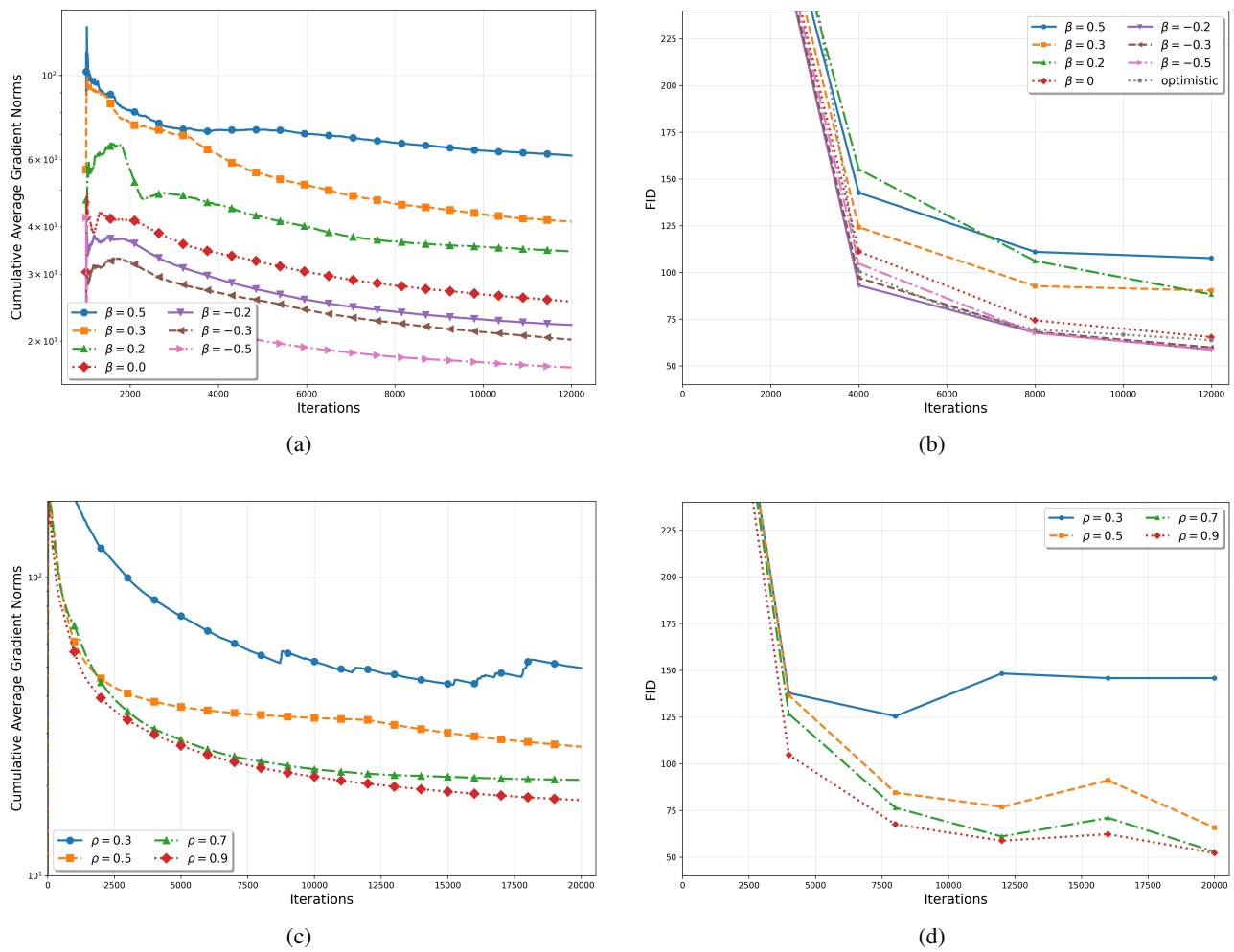

(a)

(b)

(c)

(d)

*Figure 9.* IIn this figure, we reproduce one set of the experimental results from Section 5 of the submission on CNN GANs trained on the CIFAR-10 dataset. We evaluate performance using FID and include a comparison with the *optimistic* adaptive method. The conclusion is the same as that in Figure 8 and is consistent with the results reported in Section 5 of the submission.

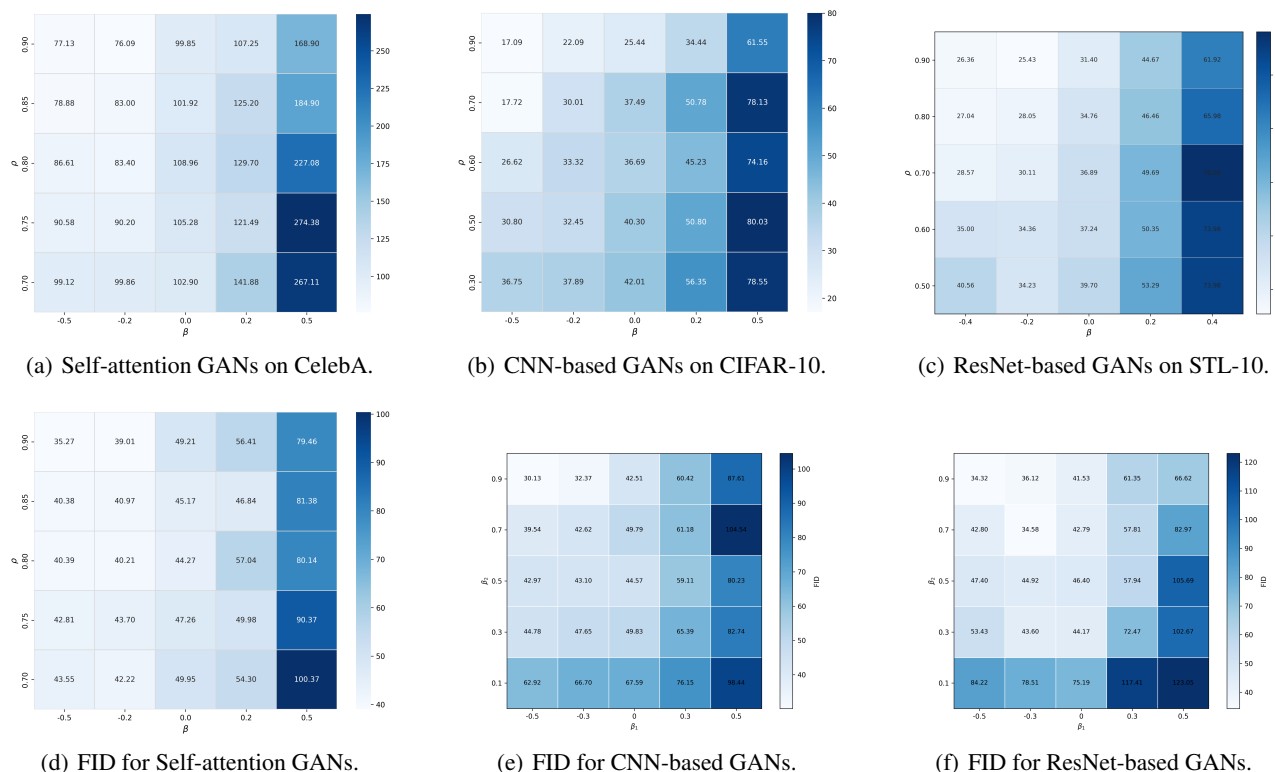

(a) Self-attention GANs on CelebA.  (b) CNN-based GANs on CIFAR-10.  (c) ResNet-based GANs on STL-10.

(d) FID for Self-attention GANs.  (e) FID for CNN-based GANs.  (f) FID for ResNet-based GANs.

*Figure 10.* 2D sweep over $(\beta, \rho)$ jointly. Each figure represents the final cumulative average gradient norms on 25 GANs training. Each figure shows the final cumulative average gradient norms over 25 GAN training runs. We observe that the **upper-left corner of each figure exhibits smaller gradient norms than the lower-right corner**, indicating that smaller $\beta$ and larger $\rho$ guide the optimization trajectories toward flatter regions of the loss landscape. This is consistent with our findings in Section 5. Similarly, upper-left corner of each figure also exhibits smaller FID than the lower-right corner.

### D.2. Inception Scores

We report the final Inception Scores from the experiments in Section 5. Table 2 summarizes the scores across architectures and datasets when training with Adam-DA under different choices of $\beta$. Table 1 reports the corresponding results under different choices of $\rho$. Together, these tables provide the numerical values underlying Figure 4.

*Table 1.* Inception scores under different $\rho$.

| Choice of $\rho$ | 0.9 | 0.7 | 0.5 | 0.3 |
|---|---|---|---|---|
| ResNet, CIFAR-10 | **7.087** ±0.358 | 6.483 ±0.129 | 6.308 ±0.270 | 6.265 ±0.198 |
| ResNet, STL-10 | **7.187** ±0.457 | 6.486 ±0.263 | 6.335 ±0.286 | 5.571 ±0.347 |
| CNN, CIFAR-10 | **7.010** ±0.178 | 6.809 ±0.171 | 6.685 ±0.186 | 6.280 ±0.135 |
| CNN, STL-10 | **7.791** ±0.410 | 7.332 ±0.426 | 6.775 ±0.288 | 6.541 ±0.262 |

*Table 2.* Inception scores under different $\beta$ values.

| Choice of $\beta$ | 0.5 | 0.3 | 0.2 | 0.0 | -0.2 | -0.3 | -0.5 |
|---|---|---|---|---|---|---|---|
| ResNet, CIFAR-10 | 4.160 ±0.122 | 4.698 ±0.158 | 5.217 ±0.232 | 7.022 ±0.282 | 7.020 ±0.255 | **7.087** ±0.358 | 7.002 ±0.313 |
| ResNet, STL-10 | 4.447 ±0.193 | 4.858 ±0.363 | 5.565 ±0.214 | 6.445 ±0.289 | 6.969 ±0.243 | **7.181** ±0.457 | 6.878 ±0.308 |
| CNN, CIFAR-10 | 4.942 ±0.141 | 6.322 ±0.169 | 6.519 ±0.104 | 6.804 ±0.197 | **7.062** ±0.153 | 7.010 ±0.178 | 6.761 ±0.195 |
| CNN, STL-10 | 6.775 ±0.288 | 7.178 ±0.256 | 7.302 ±0.227 | 7.594 ±0.346 | 7.383 ±0.257 | **7.791** ±0.410 | 7.520 ±0.319 |

## D.3. Sample Images

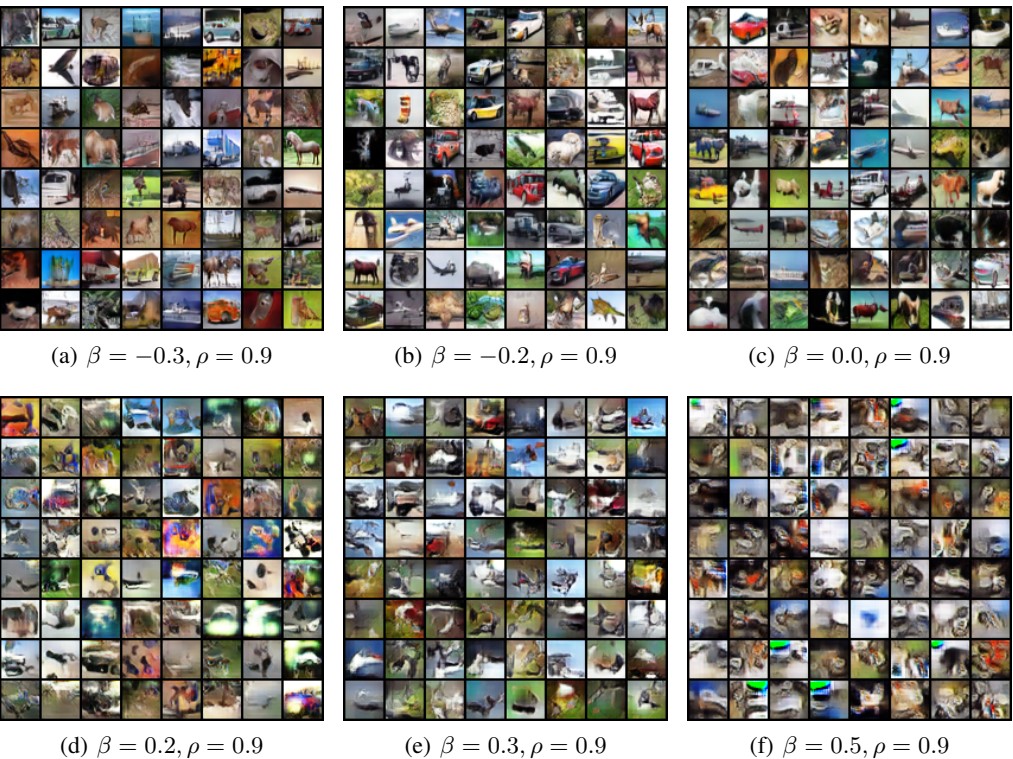

(a) $\beta = -0.3, \rho = 0.9$  (b) $\beta = -0.2, \rho = 0.9$  (c) $\beta = 0.0, \rho = 0.9$

(d) $\beta = 0.2, \rho = 0.9$  (e) $\beta = 0.3, \rho = 0.9$  (f) $\beta = 0.5, \rho = 0.9$

*Figure 11. Sample images for different $\beta$. Architecture: ResNet. Data Set: CIFAR-10.*

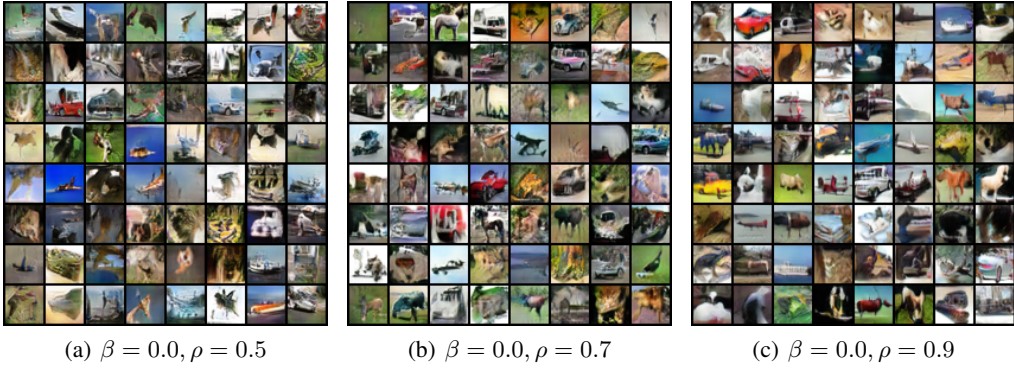

(a) $\beta = 0.0, \rho = 0.5$  (b) $\beta = 0.0, \rho = 0.7$  (c) $\beta = 0.0, \rho = 0.9$

*Figure 12. Sample images for different $\rho$. Architecture: ResNet. Data Set: CIFAR-10.*

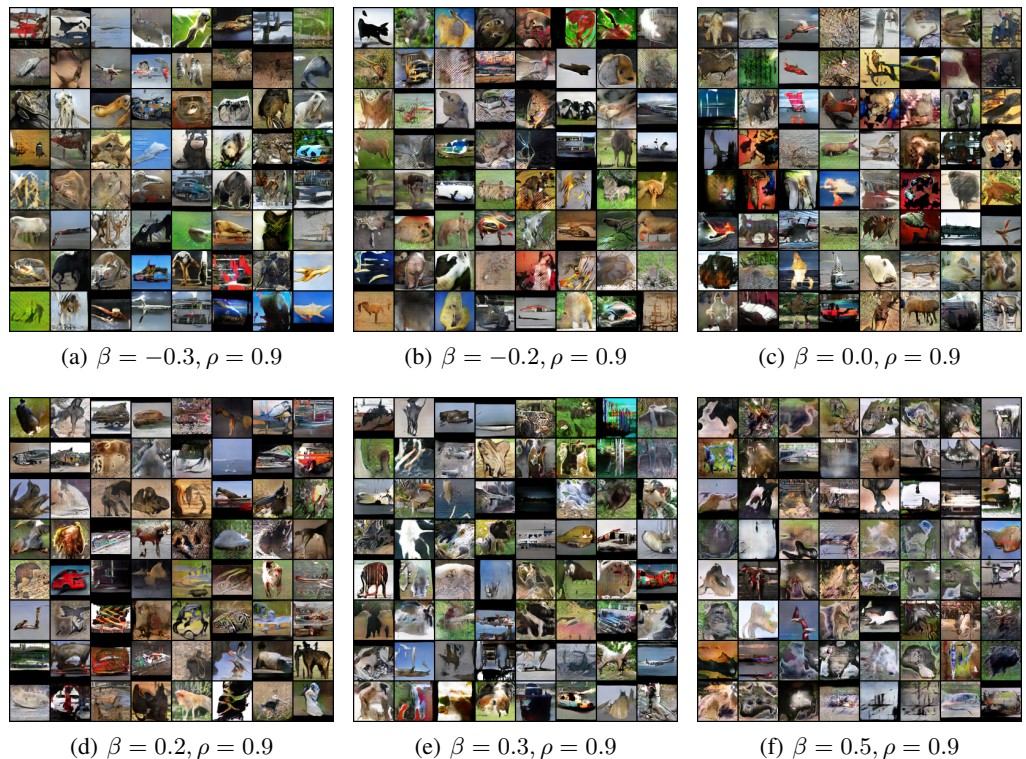

(a) $\beta = -0.3, \rho = 0.9$    (b) $\beta = -0.2, \rho = 0.9$    (c) $\beta = 0.0, \rho = 0.9$

(d) $\beta = 0.2, \rho = 0.9$    (e) $\beta = 0.3, \rho = 0.9$    (f) $\beta = 0.5, \rho = 0.9$

*Figure 13. Sample images for different $\beta$. Architecture: CNN. Data Set: STL-10.*

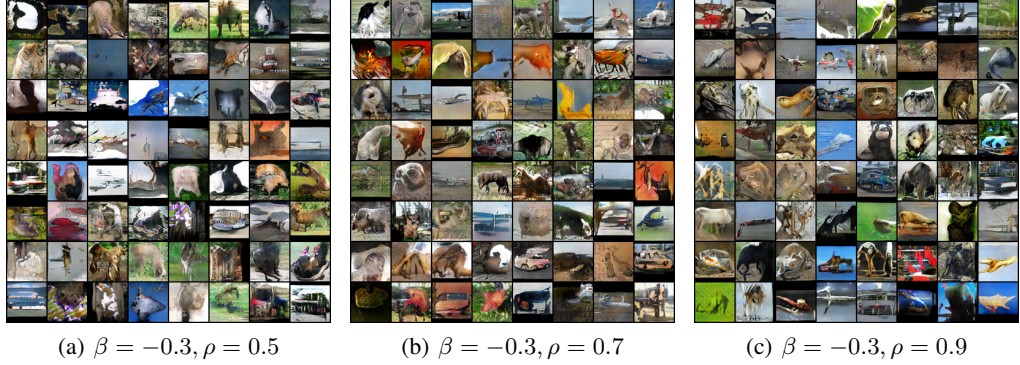

(a) $\beta = -0.3, \rho = 0.5$    (b) $\beta = -0.3, \rho = 0.7$    (c) $\beta = -0.3, \rho = 0.9$

*Figure 14. Sample images for different $\rho$. Architecture: CNN. Data Set: STL-10.*

