# OpenReview forum: "Understanding Dynamics of Adam in Zero-Sum Games: An ODE Approach"
_ICML.cc/2026/Conference — ICML 2026 regular_

### Official Review · Reviewer_YW2L · 2026-03-01

**Soundness:** 4
**Presentation:** 4
**Significance:** 4
**Originality:** 4
**Overall Recommendation:** 5
**Confidence:** 4

**Summary:**

This paper provides a rigorous theoretical framework for understanding the Adam Descent-Ascent (Adam-DA) algorithm in min-max settings. While Adam is the industry standard for training models like GANs, its behavior in competitive games has lacked the same level of theoretical grounding as its performance in minimization. The authors derive Ordinary Differential Equations (ODEs) that act as continuous-time limits of the discrete algorithm, providing a tractable way to analyze its dynamics. Their analysis reveals a "Reversed Momentum Effect," where the traditional roles of first and second-order momentum are inverted compared to standard optimization.

**Compliance With Llm Reviewing Policy:**

Affirmed.

**Final Justification:**

I will keep my positive score. The paper is high quality, and the rebuttal addressed all the concerns.

**Key Questions For Authors:**

Can the authors comment on the stochastic (SDE) case, vs. the deterministic case (ODE). No need for analysis, but to explain what they expect.

**Limitations:**

yes

**Strengths And Weaknesses:**

Strengths
1. The derivation of the high-fidelity ODE (accurate to $\mathcal{O}(h^3)$) is a significant technical contribution, allowing for the analysis of momentum and adaptivity in a unified, continuous-time framework.
2. The paper provides a theoretical justification for why negative momentum or low $\beta$ values, often used by practitioners in GAN training, actually stabilize the system and improve convergence.
3. The authors show that you don't necessarily need to invent a brand-new version of Adam to make it work in many zero-sum games. Instead, you need to tune it differently than you would for a standard minimization task.

Weaknesses

The authors show that for bilinear game continuous Adam-DA and Adam-DA always diverge regardless of the choice of parameters. There is no algorithm for convergence.

---

> ### Author Rebuttal · Authors · 2026-03-30
>
> We thank the reviewer for the careful reading and constructive comments. Please see our itemized responses below:
>
>
> **"Key Questions For Authors" part:**
>
> 1. > Can the authors comment on the stochastic (SDE) case, vs. the deterministic case (ODE). No need for analysis, but to explain what they expect.
>
> We agree with the reviewer that exploring SDE models of Adam in zero-sum games is an interesting direction for future work and could provide further insight into its behavior. In particular, we expect the following aspects to arise in the stochastic setting:
>
> * **Convergence to Stationary Distributions**: From a dynamical systems viewpoint, while the ODE can exhibit exact local asymptotic convergence to a point (as in Theorem 4.3), the continuous injection of noise in the SDE implies that the system will instead converge to a stationary distribution. In particular, it would be interesting to investigate how correlations in the gradient noise between the two players affect the dynamics, as this may reveal new phenomena that do not arise in minimization problems [1].
>
> * **Implicit Regularization**: Our deterministic ODE shows that specific parameter choices (smaller $\beta$, larger $\rho$) induce an effect that biases the trajectory toward flatter regions. In the minimization setting, recent works have shown that stochastic noise can drive optimization trajectories toward wider valleys (flat minima or saddles) [2,3,4]. We expect that an SDE analysis would show that the inherent deterministic IGR of Adam-DA acts synergistically with stochastic diffusion, thereby accelerating the discovery of regions with smaller gradient norms.
>
> ---
>
> **"Weakness" part:**
>
> 2. > The authors show that for bilinear game continuous Adam-DA and Adam-DA always diverge regardless of the choice of parameters. There is no algorithm for convergence.
>
> Bilinear games provide a natural setting for testing the performance of algorithms for zero-sum games, and several algorithms have been shown to fail in this setting [5,6]. We believe that combining Adam with techniques such as optimistic updates may help improve the convergence behavior, and we view this as an interesting direction for future work.
>
> **Reference**
>
> [1] Compagnoni et al., Adaptive Methods through the Lens of SDEs: Theoretical Insights on the Role of Noise. ICLR 2025
>
> [2] Zhu et al., The Anisotropic Noise in Stochastic Gradient Descent: Its Behavior of Escaping from Sharp Minima and Regularization Effects. ICML 2019.
>
> [3] Xie et al., A Diffusion Theory For Deep Learning Dynamics: Stochastic Gradient Descent Exponentially Favors Flat Minima. ICLR 2021
>
> [4] Wu & Su, The Implicit Regularization of Dynamical Stability in Stochastic Gradient Descent. ICML 2023
>
> [5] Bailey et al., Finite Regret and Cycles with Fixed Step-Size via Alternating Gradient Descent-Ascent. COLT 2020
>
> [6] Cheung & Piliouras, Chaos, Extremism and Optimism:
> Volume Analysis of Learning in Games. NeurIPS, 2020

---

> > ### Author Rebuttal · Reviewer_YW2L · 2026-04-01
> >
> > I am happy with the answers and keeping my scores.

---

> > > ### Author Response · Authors · 2026-04-02
> > >
> > > We are pleased that our rebuttal addressed the reviewer’s concern, and we sincerely thank the reviewer for the positive feedback and thoughtful comments.

---

### Official Review · Reviewer_pVBF · 2026-03-12

**Soundness:** 3
**Presentation:** 3
**Significance:** 3
**Originality:** 4
**Overall Recommendation:** 5
**Confidence:** 3

**Summary:**

This paper studies Adam-DA in two-player zero-sum games through a continuous-time ODE approximation and argues that Adam behaves qualitatively differently in games than in minimization. The authors seek to discuss a notable theme on whether the roles of first- and second-moment parameters reverse in adversarial settings. The authors analyze the area via two lenses of local convergence near local Nash equilibria and implicit gradient regularization (IGR).

**Compliance With Llm Reviewing Policy:**

Affirmed.

**Final Justification:**

The requested metrics and baselines were presented indicating strong performance and a valid contribution.

**Key Questions For Authors:**

* Why are there no comparisons with game-specific baselines such as optimistic / extragradient methods, complex momentum, or recent adaptive minimax methods?
* Can the authors report stronger GAN metrics and broader architectures/datasets? Inception Score alone is weak by current standards.

**Limitations:**

Overall, I think this is a good theory paper with an interesting message, but it is not yet fully convincing in its current form. The local analysis is the strongest part and the IGR and practical conclusions are weaker.

**Strengths And Weaknesses:**

The paper addresses a real gap. Adam-style methods are heavily used in GANs and min-max training, while the theory is much less mature than for minimization. The ODE viewpoint is a natural and potentially insightful way to study this. The empirical trends are at least directionally consistent with known practical observations on negative momentum in GANs and with prior game-optimization work showing that game dynamics can favor very different momentum behavior from standard optimization. However, the empirical evaluation is too narrow for the breadth of the claims. It is restricted to GANs, uses Inception Score rather than stronger modern generative metrics, and does not compare against more game-aware optimizers such as optimistic/extragradient families, complex-momentum methods, or recent adaptive minimax baselines. Recent literature includes AdaGDA/AdaMSGDA-type adaptive methods and game-specific momentum methods, which are very relevant here. GAN experiments are restricted (CIFAR-10, STL-10; CNN/ResNet; mainly Inception Score). To support broader claims, I would expect:

* Metrics: FID (at minimum), ideally KID or precision/recall.
* Datasets: e.g., CelebA or LSUN.
* A 2D sweep over ($\beta$, $\rho$) jointly.
* One other architecture family like Transformer-based GANs

---

> ### Author Rebuttal · Authors · 2026-03-30
>
> We thank the reviewer for the careful reading and constructive comments. Please see our itemized responses below:
>
> 1. > Can the authors report stronger GAN metrics and broader architectures/datasets?
>
> We provide additional experimental results in response to the reviewer’s comments in the "Weaknesses" and "Key Questions for Authors" sections, which are available at the following ***[ANONYMOUS LINK](https://www.dropbox.com/scl/fi/eledi6fde8t6z0z688r6t/Rebuttal2.pdf?rlkey=t5qnnzww36ui4ux9sicfa3vov&st=533ussrg&dl=0)***.
>
> In particular, we provide:
>
> * Experimental results on the **Self-Attention** architecture [1,2] trained on the **CelebA** dataset and evaluated using **FID**. The results are presented in Figure 1. The self-attention module is complementary to convolution and helps model long-range, multi-level dependencies across image regions. We therefore use it as a supplementary experimental setting in addition to the CNN/ResNet architectures considered in the submission.
>
> * A reproduction of one group of experiments from Section 5, with the evaluation metric changed from IS to **FID**. The results are presented in Figure 2.
>
> * A **joint 2D sweep** over $(\beta,\rho)$ on the Self-Attention, CNN, and ResNet architectures on different data sets. The results are provided in Figure 3.
>
> * A comparison between the **optimistic adaptive method** of Mokhtari et al. (2020) and Adam with different parameters. The results are shown in Figure 1(b) and Figure 2(b).
>
> These experimental results are consistent with the results reported in Section 5 of the submission. We highlight several findings from these experiments:
>
> *(1)*. Figure 1 shows that our results in Section 5 also holds for the self-attention architecture trained on CelebA. In particular, smaller $\beta$ and larger $\rho$ lead the trajectories to explore flatter regions of the GAN loss landscape in the corresponding zero-sum game. The joint 2D sweep in Figure 3 also supports this claim.
>
> *(2)*. From Figures 1 and 2, by comparing the evolution of gradient norms and FID, we observe that smaller gradient norms generally correspond to lower FID, which indicates better training performance.
>
> *(3)*. In Figures 1(b) and 2(b), we compare Adam algorithms with the optimistic method, which is primarily designed for solving zero-sum games. We future combine the adaptivity into the optimistic method, which results in the algorithms used by [3] in their experiments. We find that the optimistic method outperforms Adam with $\beta=0$, which is the standard choice in GAN training. We use the same step size of $10^{-4}$ for all algorithms, which is a widely used choice in GAN training.
>  This observation is consistent with the optimistic optimization literature. However, we further find that using a negative $\beta$ yields better performance than the optimistic method in our experiments, which support that small momentum can improve the performance of Adam in zero-sum games.
>
> ---
>
> 2. > Why are there no comparisons with game-specific baselines?
>
> Please refer to our point *(3)* in Question 1, where we future provide experiments that compare Adam with optimistic methods. We would also like to respectfully clarify that the goal of the GAN experiments in the current paper is to support our thesis on how Adam’s parameters affect the flatness of the optimization trajectory in a real-world setting. They are not intended to provide evidence that Adam outperforms other algorithms. Indeed, although several game-specific algorithms have been proposed for zero-sum games, Adam remains one of the most widely used algorithms in practice, for example in GAN training and adversarial training. It is precisely this practical importance of Adam in zero-sum games that motivates us to develop a theoretical understanding of its dynamics in this setting. We hope this clarification helps better position the empirical part of the paper.
>
> **Reference**
>
> [1] Zhang et al., Self-Attention Generative Adversarial Networks. ICML 2019
>
> [2] Zhao et al., Improved Transformer for High-Resolution GANs. NeurIPS 2021
>
> [3] Daskalakis et al., Training GANs with Optimism. ICLR 2019
>
> [4] Mokhtari et al., A Unified Analysis of Extra-gradient and Optimistic Gradient Methods for Saddle Point Problems: Proximal Point Approach. AISTATS 2020

---

> > ### Author Rebuttal · Reviewer_pVBF · 2026-04-02
> >
> > The authors have resolved all my concerns. I have raised my score accordingly. Thank you.

---

> > > ### Author Response · Authors · 2026-04-02
> > >
> > > We are pleased that our rebuttal addressed the reviewer’s concern, and we sincerely thank the reviewer for the positive feedback and thoughtful comments.

---

### Official Review · Reviewer_tYsF · 2026-03-15

**Soundness:** 3
**Presentation:** 4
**Significance:** 3
**Originality:** 3
**Overall Recommendation:** 5
**Confidence:** 4

**Summary:**

The paper studies Adam-DA, the descent-ascent variant of Adam used in GAN training and other zero-sum game formulations. The main tool is a family of ODEs (Continuous Adam-DA) obtained via backward error analysis that tracks the discrete-time algorithm with O(h³) local error, improving on the O(h²) approximation given by SignGDA-flow from prior work. Using these ODEs, the paper establishes two sets of results. First, quantitative local convergence theorems for both the ODE and the discrete-time algorithm, showing that smaller first-order momentum β widens the convergence range in zero-sum games, which is the opposite of Adam's behavior in minimization. A corollary shows Adam-DA always diverges in bilinear games regardless of parameter choices. Second, a qualitative "thesis" on implicit gradient regularization (IGR): smaller β and larger ρ steer trajectories toward flatter regions of the loss landscape in zero-sum games, again reversing the roles established in the minimization setting. GAN experiments on CIFAR-10 and STL-10 support the IGR thesis.

**Compliance With Llm Reviewing Policy:**

Affirmed.

**Key Questions For Authors:**

1. Gidel et al. (2019) establish theoretically, for gradient descent-ascent with alternating updates and negative momentum, convergence in bilinear games. Corollary 4.6 in the current paper states that Adam-DA always diverges in bilinear games regardless of the choice of beta (including negative values). The present paper's analysis is for simultaneous updates. Do the authors' results extend to alternating Adam-DA? If alternating Adam-DA with negative beta can converge in bilinear games (analogously to Gidel et al.'s result for GDA), then Corollary 4.6 covers only a subset of practically relevant cases, and the contrast between the two results should be discussed explicitly. This also bears on how the paper frames Corollary 4.5: the claim that smaller beta is beneficial in zero-sum games echoes Gidel et al. directly, but for GDA-type methods rather than Adam. Is the contribution here primarily establishing this for Adam, or is there a qualitatively new message?
2. Related to the above: the practical GAN experiments in Section 5 use what update rule, simultaneous or alternating? Since Gidel et al. showed that the update rule matters substantially for convergence behavior, it would help to be explicit about this in the experimental setup.
3. The IGR thesis is supported by an informal dominance argument (Jacobian cross terms >> diagonal terms) and by experiments. Would it be possible provide a more precise formal statement, even in a simplified quadratic or bilinear setting, that relates the sign and magnitude of K(β, ρ) to the evolution of ‖\nabla f‖_1 in a controlled way? What are the main challenges of arriving at a rigorous result? A cleaner theoretical result here, even a special case, would significantly strengthen the section.
4. Figure 4 shows that lower gradient norms correlate with higher Inception Scores. This is interpreted as evidence for the IGR thesis. But lower gradient norms could also just indicate that training has stabilized or that the model has converged, independent of any flatness argument. Can the authors argue more directly that the trajectories are in a geometrically flatter region, rather than just that norms are lower?

**Limitations:**

The paper honestly acknowledges the restriction to deterministic analysis and the gap to the stochastic regime used in practice. The societal impact statement is adequate for a theory paper. No ethical concerns.

**Strengths And Weaknesses:**

Soundness
The theoretical framework is carefully built. The O(h^3) backward error analysis (Theorem 3.1) follows established methodology from the numerical analysis literature and is a genuine improvement over the O(h^3) baseline. Proposition 4.1, which shows that the Jacobian of Continuous Adam-DA is a quadratic polynomial in the Jacobian J, is a useful insight that makes the local convergence analysis tractable. Theorem 4.4 (discrete-time local convergence) is proved in detail: the strategy of splitting Adam-DA into an autonomous part T-bar plus an exponentially vanishing non-autonomous perturbation R(n,z) is a good idea when it comes to separating the transient effects of Adam’s initial bias correction.
The IGR section (Section 5) is weaker. The argument that smaller beta and larger rho drive the trajectories toward flatter regions is presented as a "thesis" motivated by an ODE dominance argument, but no formal theorem is proved. The key step relies on assuming cross terms of the Jacobian are stronger than the diagonal terms, f along the trajectory, which is posited but not verified. Still, I find that stating this hypothesis (or thesis) is useful and can be valuable to the community. The gap between the ODE-level observation and the stochastic GAN experiments is substantial, but the authors include an honest discussion about it.

Presentation
The paper reads well overall. The comparison between zero-sum games and minimization is clearly set up and paid off consistently throughout. The related work appendix is thorough and does a good job of positioning the contributions relative to Rosca et al. (2021), Feng et al. (2025), and Cattaneo et al. (2024). One minor issue: Section 5 would benefit from a brief upfront note that what follows is a thesis supported by ODE-level reasoning and experiments rather than a theorem, so readers calibrate expectations accordingly.

Significance
The paper tackles a genuinely important theoretical gap: Adam-DA is the dominant optimizer for GAN training, but understanding of its game-theoretic behavior lags well behind its minimization theory. The main qualitative finding, that momentum parameters play opposite roles in zero-sum games versus minimization, is both surprising and practically relevant. It provides theoretical grounding for the empirically observed benefits of negative beta in GAN training. That said, the significance is somewhat tempered by the fact that GANs are no longer at the frontier of generative modeling, having been largely displaced by diffusion models. The paper remains mathematically interesting and the results apply to zero-sum games more broadly, but the practical impact is more limited than it would have been a few years ago.

Originality
The paper clearly extends three lines of prior work (Rosca et al. 2021, Feng et al. 2025, Cattaneo et al. 2024) in a non-trivial direction, and the new insights about the role of ρ and ε in the game setting are genuinely novel. The paper discusses the relationship to Gidel et al. (2019) on negative momentum, but this connection deserves a bit more careful treatment; see the questions below.

---

> ### Author Rebuttal · Authors · 2026-03-30
>
> We thank the reviewer for the careful reading and constructive comments. Please see our itemized responses below:
>
> **"Key Questions For Authors" part:**
>
> 1. > Gidel et al. (2019)  ... the contrast between the two results should be discussed explicitly. ... is there a qualitatively new message?
>
> We thank the reviewer for pointing out the missing discussion on alternating updates, we will add the following remark in the revised version after we introduce the Adam-DA algorithm (line 153):
>
> *For simplicity, this paper mainly considers the setting in which players update their strategies simultaneously. It is worth noting that alternating updates can often improve the convergence of game dynamics (Gidel et al., 2019).*
>
> We believe that the ODE approach developed in this paper can be extended to Adam with alternating updates, although this would introduce additional complexity into the equations. In addition, our Corollary 4.5 extends the divergence result of Gidel et al. (2019) to Adam. We believe this extension is not obvious a priori, since several simpler momentum-based methods, such as optimistic methods, can converge in this setting.
>
> ---
>
> 2. > The practical GAN experiments in Section 5 use what update rule, simultaneous or alternating?
>
> Following the tradition in GANs training literature, the experiments employ alternating updates. We will clarify this point in the “Experimental Results” subsection of Section 5 in the revised manuscript.
>
> ---
>
> 3. > The IGR thesis ... Would it be possible provide a more precise formal statement... What are the main challenges of arriving at a rigorous result?
>
> We thank the reviewer for this suggestion. We believe that the main difficulty in establishing formal results on IGR in certain settings lies in the fact that, although the ODEs used to derive IGR can be shown to have a local error of order $\mathcal{O}(h^3)$, the hidden constant may depend in a complicated way on the specific structure of the instance. As a result, the quantitative conclusions suggested by the ODE analysis may not carry over directly to the original algorithms. Nevertheless, we believe this is definitely an interesting direction for future work.
>
> ---
>
> 4. > Can the authors argue more directly that the trajectories are in a geometrically flatter region, rather than just that norms are lower?
>
> Following the implicit gradient regularization literature [1,2], this paper defines flatness through smaller gradient norms. This differs from other works that define flatness through stationary points with small Hessian eigenvalues. The definition used here is particularly suitable for zero-sum games setting considered in the current submission, where many algorithms including Adam do not converge to stationary points. In this setting, the gradient norm based definition of flatness allows to character the geometry property of the whole trajectories of the game dynamics, rather than on the single stationary points.
>
> ---
>
> **"Weakness" part:**
>
> 5. >Presentation:  Section 5 would benefit from a brief upfront note that what follows is a thesis supported by ODE-level reasoning and experiments rather than a theorem, so readers calibrate expectations accordingly.
>
> We thank the reviewer for this suggestion. We will add the following remark in the revised version at Section 5 (line 298):
>
> *In this section, we adopt an empirical approach to study the interaction between the flatness of algorithm trajectories and algorithmic parameters. Our investigation is inspired by the ODEs derived in the previous section.*
>
>
> **Reference:**
>
> [1] Barrett, D. & Dherin, B. Implicit gradient regularization. ICML 2021
>
> [2] Ghosh et al., Implicit regularization in Heavy-ball momentum accelerated stochastic gradient descent.  ICLR 2023

---

> > ### Author Rebuttal · Reviewer_tYsF · 2026-04-03
> >
> > I am happy with the discussion and maintain my "accept" rating.

---

> > > ### Author Response · Authors · 2026-04-04
> > >
> > > We are pleased that our rebuttal addressed the reviewer’s concern, and we sincerely thank the reviewer for the positive feedback and thoughtful comments.

---

### Official Review · Reviewer_jgiH · 2026-03-15

**Soundness:** 3
**Presentation:** 3
**Significance:** 3
**Originality:** 2
**Overall Recommendation:** 4
**Confidence:** 4

**Summary:**

This paper investigates the properties of simultaneous Adam updates in min-max optimization. Namely, the authors examine the (i) local convergence properties of Adam under different values of the hyperparameters of the algorithm and (ii) a potential implicit bias of Adam descent-ascent (Adam-DA) updates towards "flatter" saddle points.

To analyze the local convergence properties of Adam-DA the authors repurpose a continuous-time model for the dynamics of Adam [1] that tracks the discrete-time system of step-size $h$ with a tracking error of $O(h^3)$ (They prove this is the case in Theorem 3.1). Then, through Lyapunov stability analysis using the Jacobian of the dynamical system, the authors argue about the range of the hyperparameter values that make convergence plausible under larger stepsizes.

In Theorem 4.3, the authors demonstrate that smaller first-momentum parameter  $\beta$ allows stability under a larger stepsize. This comes in contrast to minimization which calls for a larger $\beta$.

In Theorem 4.4, the authors show that under some assumption on the spectrum of the Jacobian at the equilibrium and under a range of values of $\beta$ and $\rho$, Adam-DA converges exponentially to the stationary point.

Section 5 is more exploratory. The authors in equations (4) and (5) recognize that the ODE that trackes discrete-time Adam-DA has two competing terms that implicitly minimize the $\ell_1$ norm of the gradients. Under the assumption of strong interaction (roughly, $\| \nabla_{xy} f(x,y) \| \geq \| \nabla_{xy} f(x,y) \|$) the competing terms in (4,5) resolve into a flattening direction when Adam-DA uses a smaller $\beta$ and larger $\rho$. The authors posit that higher cumulative flatness is correlated to higher inception score in GANs.

---

1. Ma, C., Wu, L., et al. A qualitative study of the dynamic
behavior for adaptive gradient algorithms. In Mathematical and scientific machine learning

**Compliance With Llm Reviewing Policy:**

Affirmed.

**Final Justification:**

The paper investigates the implicit bias of Adam in min-max optimization. The investigation is thorough but I am afraid that there is not adequate evidence to argue for the importance of "flatness" in min-max optimization --- at least nowhere close to the extent of the justification in minimization of loss functions for ML. I weakly recommend acceptance, but I believe the correct notion of flatness should be investigated thoroughly:
* is it the flatness of saddle-point of $f(x,y)$ or of $V(x) = max_{y} f(x,y)$,
* is it the average iterate trajectory flatness or the solution flatness?

I believe that the reason the answer is not clear is because the motivation of investigating flatness in min-max optimization needs additional consideration and grounding in modern ML practice (GANs are a very interesting but dated application.)

**Key Questions For Authors:**

* Non-convergence in bilinear games is somewhat expected from previous works. What happens to convergence when there is some dampening effect e.g. under regularization? This is a common scenario under weight decay and can be a bridge between your result for nonlinear dynamics (Theorem 4.3) and dampened/regularized bilinear games.

* What other settings would justify the interest in the properties of saddle-point flatness in machine learning? There is a lot of indications (and even proofs) that flatness of minima does imply generalization; how should we evaluate the flatness of saddle-points though?

* Can you include a plot that studies the correlation of cumulative flatness to inception score? It would only help your claim in my opinion.

**Limitations:**

yes

**Strengths And Weaknesses:**

Strengths:
* The narrative of the paper is clear. The authors use existing established methods and the authors make reasonable assumptions that have been already considered in the literature.
* The authors make a good job in justifying their use of an ODE for discrete-time Adam-DA by explicitly bounding the error for the min-max case.
* The study of implicit bias in min-max optimization is an interesting direction.

Weaknesses:
* The bridge between the bilinear setting and the general nonlinear dynamical system is missing.  The assumption for Theorem 4.4 implies a dampening effect that should have been considered in the bilinear setting. What happens when there is regularization for at least one of the players?
* The connection of flatness of solutions and better inception score of GANs is not examined in detail. I want to believe in the importance of flatness in min-max optimization but I am afraid that the connection is not well-established.

---

> ### Author Rebuttal · Authors · 2026-03-30
>
> We thank the reviewer for the careful reading and constructive comments. Please see our itemized responses below:
>
> **"Key Questions For Authors" part**:
>
> 1. >What happens to convergence ... under regularization?
>
> Our main local convergence results (Theorems 4.3 and 4.4) also apply to settings with regularization, for example, $$f(x,y) = x^{\top}Ay + \epsilon_1 \lVert x \rVert^2 - \epsilon_2 \lVert y \rVert^2,\ 0<\epsilon_1,\epsilon_2 \ll 1$$In this setting, Theorem 4.3 provides a precise characterization of the parameter regime under which Adam converges, in terms of the spectrum of the Jacobian of the payoff. Moreover, this setting is fully consistent with the interaction-dominated regime described in the paper. Therefore, Theorem 4.3 further reveals a trade-off between the step size and momentum for convergence.
>
> ---
>
> 2. >What other settings ... evaluate the flatness of saddle-points?
>
> In this paper, we use the cumulative average gradient norms $$\frac{1}{T}\sum^T_{t=1}(\lVert\nabla_xf\rVert^2_1 + \lVert\nabla_yf\rVert^2_1)\quad(1)$$to evaluate the flatness of the overall min-max optimization trajectory, rather than that of an individual saddle point. Our motivation is twofold.
>
> * First, this choice is inspired by the literature on implicit gradient regularization [1,2], where gradient norms are used as a measure of trajectory flatness in minimization problems. The objective in (1) can be viewed as an extension of this idea to zero-sum games.
> * Second, practical zero-sum game dynamics often do not converge to saddle points. This suggests that studying the flatness of the entire trajectory may be more meaningful than focusing on a single saddle point. We also note that quantities similar to (1) have also been used to study generalization of algorithms [3].
>
> Flatter equilibria are intuitively desirable because small deviations from such equilibrium strategies lead to minor changes in payoffs, which implies a certain degree of robustness. This property may be beneficial in machine learning tasks formulated as zero-sum games, such as adversarial training [5].
>
> ---
>
> 3. >Include a plot that studies the correlation of  flatness to IS.
>
> We thank the reviewer for this insightful suggestion. At the following ***[ANONYMOUS LINK](https://www.dropbox.com/scl/fi/7a4uutiwn6c39t6xh1bq1/Rebuttal1.pdf?rlkey=y28tbx8rdsuektb94h06vfoqd&st=xgkgeag4&dl=0)***, we provide additional figures illustrating the correlation between IS and flatness. Specifically, taking ResNet on CIFAR-10/STL-10 as examples, we have reorganized the experimental results in Figures 3 and 4 and included the evolution of IS and flatness in the same figure to make their relationship clearer. The results suggest that flatter trajectories tend to be associated with higher IS, which show a strong correlation between them. We will revise figures in the paper accordingly to make this point clearer.
>
> ---
>
> **"Weakness" part**:
>
> 4. >What happens when there is regularization for at least one of the players?
>
> Please refer to our response to Question 1.
>
> ---
>
> 5. > The connection of flatness of solutions and better inception score of GANs is not examined in detail.
>
> We provide additional experimental results via the following ***[ANONYMOUS LINK](https://www.dropbox.com/scl/fi/eledi6fde8t6z0z688r6t/Rebuttal2.pdf?rlkey=t5qnnzww36ui4ux9sicfa3vov&st=533ussrg&dl=0)*** to further strengthen our empirical findings, including experiments with new architecture (self-attention GANs), an additional dataset (CelebA), and an additional metric (FID). The new experiments are consistent with our results in Section 5. Please refer to our response to Question 1 from Reviewer pVBF for a detailed explanation of these experiments.
>
> We would also like to clarify that the goal of the GAN experiments in the current paper is to support our thesis on how Adam’s parameters affect the flatness of the optimization trajectory in a real-world setting. They are not intended to provide a comprehensive study of the relationship between flatness and GAN performance, which is beyond the scope of this work. More generally, even for well-studied minimization problems, the relationship between flatness and generalization is still not fully understood [4]. In this sense, we view our experimental results as evidence suggesting an analogous connection in zero-sum games, rather than as a full characterization of this relationship. We hope this clarification better positions the empirical part of the paper.
>
>
> **Reference:**
>
> [1] Barrett, D. & Dherin, B. Implicit gradient regularization. ICML 2021
>
> [2] Ghosh et al., Implicit regularization in Heavy-ball momentum accelerated stochastic gradient descent.  ICLR 2023
>
> [3] Li et al., On Generalization Error Bounds of Noisy Gradient Methods for Non-Convex Learning. ICLR 2020
>
> [4] Petzka et al., Relative Flatness and Generalization. NeurIPS 2021
>
> [5] Balcan et al., Nash Equilibria and Pitfalls of Adversarial Training in Adversarial Robustness Games. AISTATS 2023

---

> > ### Author Rebuttal · Reviewer_jgiH · 2026-04-06
> >
> > I thank the authors for the time they put to address my concerns. I think your study is interesting but I am afraid that this line of research of implicit bias in min-max optimization is putting the cart before the horse. I believe that more effort should be put into thinking about improved generalization and flatness of saddle-points (what interesting settings does it capture from practice? arguably, GANs are dated technology). Another point that is not clear to me is whether we should we be looking into the flatness of $f(x,y)$ or $V(x) = max_y f(x,y)$?
> >
> > Nonetheless, I do appreciate the fact that the authors offer some preliminary indications that their notion of flatness (the average flatness of the trajectory) improves the output quality of images in GANs.

---

> > > ### Author Response · Authors · 2026-04-07
> > >
> > > We sincerely thank the reviewer for their helpful suggestions and for their interest in our work, which have greatly helped us improve the manuscript.
> > >
> > > We believe the reviewer’s suggestion to compare different notions of “flatness” in zero-sum games is particularly important. Although flatness is a widely used heuristic notion in minimization, there has been relatively little discussion of its possible analogue in zero-sum games. We are grateful that the reviewer appreciated our effort to give a suitable notion that may serve as a generalization in the zero-sum setting. We have explained our motivation for using the current measure of flatness in our previous rebuttal to the **reviewer’s second question**.
> > >
> > > We also agree with the reviewer that it is important to explore settings beyond GANs in which flatness in zero-sum games may be useful. More generally, we believe that machine learning tasks involving strong adversarial interactions between multiple players, such as adversarial example games (Bose et al., 2020) and various multi-agent settings, may also benefit from the notion of flatness. We believe that further exploring these directions would be an interesting avenue for future work.

---

### Decision · Program_Chairs · 2026-04-30

**Decision:**

Accept (regular)

**Comment:**

The reviewers agree that the paper makes a significant technical contribution with the introduction of a high-fidelity ODE for Adam-DA, and appreciate the convergence results, qualitative insights (e.g. comparison between the minimization and zero-sum cases), and connections with certain empirical practices (such as the use of negative momentum and its stabilizing properties).

Some concerns were raised about the narrow empirical evaluation, the lack of comparison to other optimizers, and an unconvincing connection between the flatness notion used in the paper and model quality. Authors provide additional experiments and discussion in their rebuttal, and they are encouraged to include as much details as possible in the revision.